# Neuronal parts list and wiring diagram for a visual system

Arie Matsliah[1,16], Szi-chieh Yu[1,16], Krzysztof Kruk[2,3], Doug Bland[1], Austin T. Burke[1], Jay Gager[1], James Hebditch[1], Ben Silverman[1], Kyle Patrick Willie[1], Ryan Willie[1], Marissa Sorek[1,3], Amy R. Sterling[1,3], Emil Kind[4], Dustin Garner[5], Gizem Sancer[6], Mathias F. Wernet[4], Sung Soo Kim[5], Mala Murthy[1✉], H. Sebastian Seung[1,7✉] & The FlyWire Consortium*

A catalogue of neuronal cell types has often been called a 'parts list' of the brain[1], and regarded as a prerequisite for understanding brain function[2,3]. In the optic lobe of *Drosophila*, rules of connectivity between cell types have already proven to be essential for understanding fly vision[4,5]. Here we analyse the fly connectome to complete the list of cell types intrinsic to the optic lobe, as well as the rules governing their connectivity. Most new cell types contain 10 to 100 cells, and integrate information over medium distances in the visual field. Some existing type families (Tm, Li, and LPi)[6–10] at least double in number of types. A new serpentine medulla (Sm) interneuron family contains more types than any other. Three families of cross-neuropil types are revealed. The consistency of types is demonstrated by analysing the distances in high-dimensional feature space, and is further validated by algorithms that select small subsets of discriminative features. We use connectivity to hypothesize about the functional roles of cell types in motion, object and colour vision. Connectivity with 'boundary types' that straddle the optic lobe and central brain is also quantified. We showcase the advantages of connectomic cell typing: complete and unbiased sampling, a rich array of features based on connectivity and reduction of the connectome to a substantially simpler wiring diagram of cell types, with immediate relevance for brain function and development.

Some of the greatest scientific discoveries of the twentieth century concern the neural basis of sensory perception. Hubel and Wiesel's discovery of simple and complex cells in the visual cortex not only entered neuroscience textbooks, but the hypothetical neuronal wiring diagrams in their 1962 paper[11] also inspired convolutional nets[12,13], which eventually ignited the deep-learning revolution in artificial intelligence[14]. It may come as a surprise that directly mapping such wiring diagrams, influential as they may be, has been highly challenging or even impossible in mammalian brains. Progress is being made by visual physiologists[15–17], and the reconstruction of a column of visual cortex from electron microscopy images is also becoming feasible[18,19]. These are tiny slivers of visual systems; scaling up to the full complexity of mammalian vision is still aspirational.

To imagine the future of visual neuroscience, it is helpful to extrapolate from a brain of a more modest size—that of the fly. Especially over the past 15 years, visual neural circuits have been intensively investigated in *Drosophila*[4] with great progress in understanding the perception of motion[5,10], colour[20] and objects[21], as well as the role of vision in complex behaviours like courtship[22]. The release of a neuronal wiring diagram of a *Drosophila* brain[23–25] poses an unprecedented opportunity. The first wiring diagram for a whole brain contains as a corollary the first wiring diagram for an entire visual system, as well as all the wiring connecting the visual system with the rest of the brain.

About 38,500 neurons are intrinsic to the right optic lobe of the reconstructed *Drosophila* brain (Extended Data Fig. 1a). The full wiring diagram for these neurons is too complex to comprehend or even visualize. It is essential to reduce complexity by describing the connectivity between types of cells. For example, the roughly 800 ommatidia in the compound eye send photoreceptor axons to roughly 800 L1 cells in the lamina, which in turn connect with around 800 Mi1 cells. That is a lot of cells and connections, but they can all be described by the simple rules that photoreceptors connect to L1, and L1 connects to Mi1. Some such rules are known[7,26–30], but this knowledge is fragmentary and incomplete.

Here we exhaustively enumerate all cell types intrinsic to the optic lobe, and find all rules of connection between them. We effectively collapse 38,500 intrinsic neurons onto just 227 types, a reduction of more than 150×. The wiring diagram is reduced from a 38,500 × 38,500 matrix to a 227 × 227 matrix, an even greater compression. We additionally provide rules of connectivity between intrinsic types and 500 types of boundary neurons (defined below), which have also been annotated[25].

In our connectomic approach, a cell type is defined as a set of cells with similar patterns of connectivity[9], and such cells are expected to

[1]Princeton Neuroscience Institute, Princeton University, Princeton, NJ, USA. [2]Independent researcher, Kielce, Poland. [3]Eyewire, Boston, MA, USA. [4]Institut für Biologie—Neurobiologie, Freie Universität Berlin, Berlin, Germany. [5]Molecular, Cellular and Developmental Biology, University of California, Santa Barbara, Santa Barbara, CA, USA. [6]Department of Neuroscience, Yale University, New Haven, CT, USA. [7]Computer Science Department, Princeton University, Princeton, NJ, USA. [16]These authors contributed equally: Arie Matsliah, Szi-chieh Yu. *A list of authors and their affiliations appears at the end of the paper. ✉e-mail: mmurthy@princeton.edu; sseung@princeton.edu

share the same function[2]. By the same logic, cell types with similar patterns of connectivity should have similar functions. This logic will be used to generate hypotheses about the functions of newly discovered cell types, as well as the previously known cell types for which functional information has been lacking.

## Class, family and type

Neurons intrinsic to the optic lobe are those with almost all of their synapses inside the optic lobe (Methods), and are the main topic of this study (Extended Data Fig. 1a). Moreover, there are boundary neurons that straddle the optic lobe and the rest of the brain (Extended Data Fig. 1b). Boundary neurons fall into several classes: visual projection neurons (VPNs) project from the optic lobe to the central brain, visual centrifugal neurons (VCNs) do the opposite and heterolateral neurons extend from one optic lobe to the other while making few or no synapses in the central brain. Targets of boundary neurons in the central brain are generally multimodal and/or sensorimotor[24], mixing information coming from the eyes and other sense organs, so we regard the optic lobe proper as the fly's visual system.

The brain of a single *Drosophila* adult female was reconstructed by the FlyWire Consortium[24,31]. We proofread around 38,500 intrinsic neurons in the right optic lobe (counts by type are shown in Extended Data Table 1), as well as 3,900 VPNs, 250 VCNs, 150 heterolateral neurons and 4,700 photoreceptor cells (left optic lobe numbers are shown in the Methods). In total, 77% of the synapses of intrinsic neurons are with other intrinsic neurons, and 23% are with boundary neurons.

We divide optic lobe intrinsic neurons into four broad classes: columnar, local interneuron, cross-neuropil tangential and cross-neuropil amacrine (Fig. 1a–c). Cells of the columnar class (Fig. 1a) have axons oriented parallel to the main axis of the visual columns ('axon' is defined in the Methods). Following a previous study[6], the arbour of a columnar neuron is allowed to be wider than a single column; what matters is the orientation of the axon, not the aspect ratio of the arbour. Photoreceptor cells are columnar but are not intrinsic to the optic lobe, strictly speaking, because they enter from the retina. Nevertheless, they will sometimes be included with intrinsic types in the following analyses.

The optic lobe (Extended Data Fig. 1a,b) contains four main neuropils (lamina, medulla, lobula and lobula plate) and a smaller fifth neuropil— the accessory medulla (synapse counts by type family in each neuropil are shown in Extended Data Table 2 and the number of cells in each optic lobe is shown in Extended Data Table 3). We further distinguish between distal and proximal medulla, regarding them as two separate neuropils[6] (Extended Data Fig. 1c). The border between them is layer 7 of the medulla (M7), which is also known as the serpentine layer[6,32].

A columnar cell spans multiple neuropils (Fig. 1a). Cells of the local interneuron class (Fig. 1b) are defined as being confined to a single neuropil. We also define two classes that cross multiple neuropils but are not columnar. A cross-neuropil tangential cell (Fig. 1c) has an axon that is oriented perpendicular to the main axis of the visual columns as it runs inside a neuropil. A cross-neuropil amacrine cell (Fig. 1c) lacks an axon. Interneurons are typically amacrine, but sometimes have an axon in the tangential orientation.

Each class is divided into families. A family is defined as a set of cells that share the same neuropils (Fig. 1a–c and Methods). For example, the Tm family projects from the distal medulla to the lobula, while the TmY family projects from the distal medulla to both the lobula and lobula plate (Fig. 1a; Tm and TmY pass through the proximal medulla, and also typically receive inputs there).

Each family is divided into cell types. All 227 intrinsic types as well as photoreceptor types are available for 3D interactive viewing at the Fly-Wire Codex (https://codex.flywire.ai). Supplementary Data 1 includes a list of all intrinsic types and their properties. Supplementary Data 2 contains one 'card' for each type, which includes its discriminative logical predicate (see below), basic statistics, diagram showing stratification and other single-cell anatomy, and 3D renderings of all the cells in the type.

Most neurons in the optic lobe are columnar (Fig. 1e (right)), and half of the families are columnar (Fig. 1e (left)). Interneurons constitute just 17% of optic lobe intrinsic neurons, but the majority of cell types (Fig. 1e (middle)). A columnar family (Tm) contains more cells than any other family (Fig. 1f (right)). An interneuron family (Sm) contains more types than any other family (Fig. 1f (left)).

The columnar families (Fig. 1a) are well known[6]. The Sm interneuron family is new (Fig. 1b), and its name is inspired by its stratification in the serpentine medulla (M7). Some of the cross-neuropil families are wholly or almost wholly new (Fig. 1c). Over half of the cell types are new, and many of these are interneuron types.

## Connectomic approach to cell types

For each cell, we define an output feature vector by the number of output synapses onto neurons of cell type $t$, which runs from 1 to $T$. The output feature vector is a row of the cell-to-type connectivity matrix (Methods). For each cell, we similarly define an input feature vector by the number of input synapses received from neurons of cell type $t$. This is a column of the type-to-cell connectivity matrix (Methods). The input and output feature vectors are concatenated to form a $2T$-dimensional feature vector (Fig. 2a). The feature dimensions include only intrinsic types, so $T$ is 227.

A cell type is defined as a set of cells with similar feature vectors[9]. Cells of the same type are near each other in feature space, while cells of different types are far away (Fig. 2b). This was quantified using the weighted Jaccard distance (hereafter, Jaccard distance; Methods).

Our definition of feature vectors requires that some cell types should already exist. An initial set of cell types was defined by human analysts using traditional morphological criteria (Methods). These traditional cell types were used to compute feature vectors, and hierarchical clustering was applied. In many cases, this led to further division into cell types that could not be distinguished by traditional criteria. In other cases, it led to grouping of morphological variants into a single type. After splitting or merging types, the feature vectors were recomputed and the process was continued iteratively.

The final cell types were validated in several ways (Methods). We show that our clustering is self-consistent, in the sense that almost all cells end up in the original cluster if we attempt to reassign each cell's feature vector to the nearest cluster. For more interpretable evaluations, we construct compact connectivity-based discriminators that can predict cell type membership (Extended Data Fig. 2 and Supplementary Data 3). We show that membership can be accurately predicted by a logical conjunction of on average five synaptic partner types. For each interneuron type, we also provide selected pairs of features that can be used to discriminate that type from others in the same neuropil (Extended Data Fig. 3 and Supplementary Data 4).

## Hierarchical clustering of cell types

We defined a connectomic cell type as a set of cells with similar feature vectors based on connectivity. It follows that cells of the same type should share the same function, according to the maxim "Nothing defines the function of a neuron better than its connections"[33]. The same maxim also implies that cell types with similar feature vectors should have similar visual functions. A cell type feature vector can be obtained by summing the feature vectors over all cells in that type, followed by normalization (Methods). Computing the Jaccard distance between all pairs of cell type feature vectors and applying average linkage hierarchical clustering yields a dendrogram of cell types (Methods and Fig. 2c). Thresholding the dendrogram yields a flat clustering (Fig. 2c), which will be interpreted later on.

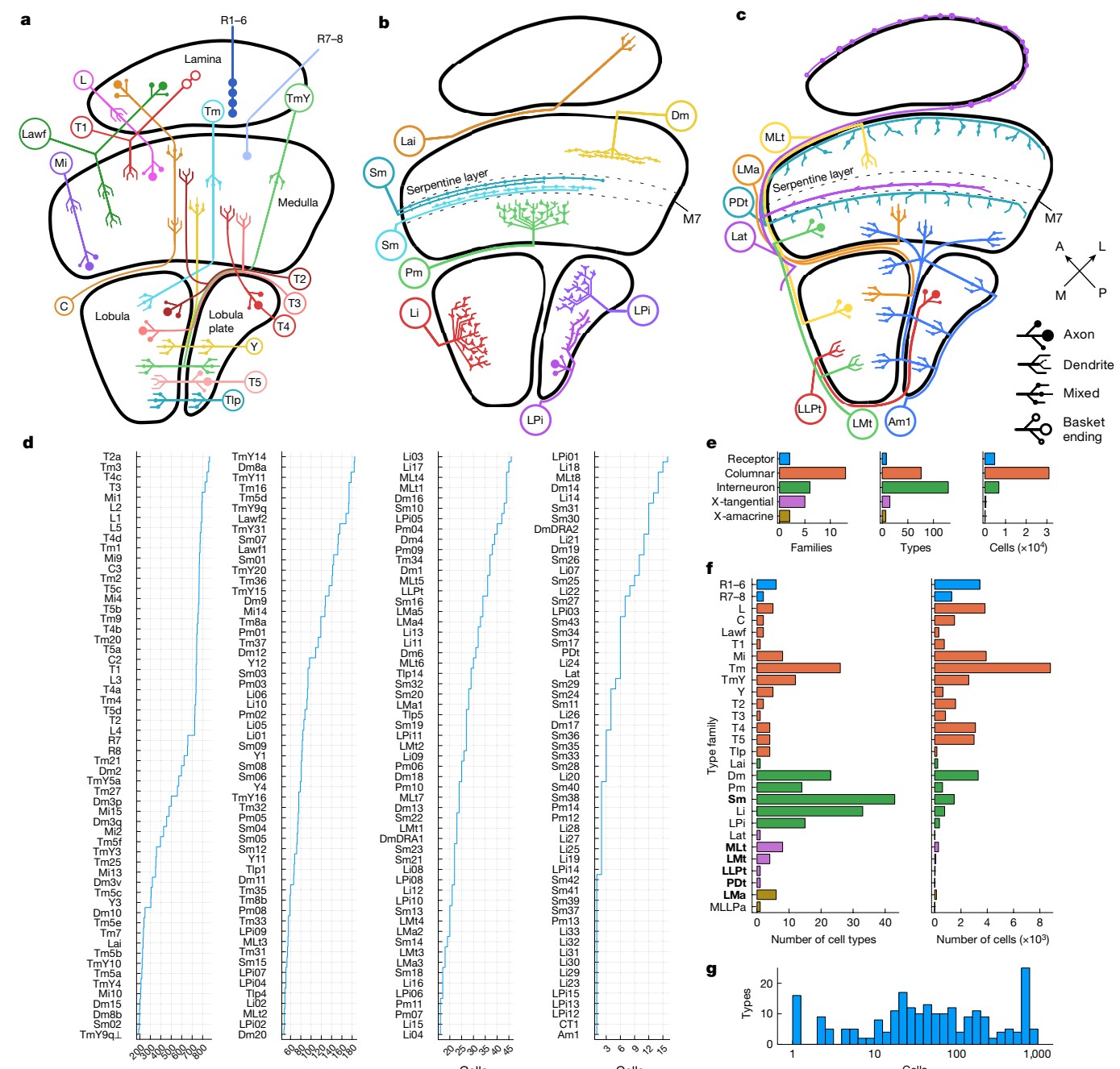

**Fig. 1 | Class, family, type and cell. a**, Families in the columnar class. C, centrifugal; L, lamina monopolar; Lawf, lamina wide-field; Mi, medulla intrinsic; R, receptor; T1–T5, T neuron; Tm, transmedullary; TmY, transmedullary Y; Tlp, translobula plate; Y, Y neuron. **b**, Families in the interneuron class. Serpentine medulla (Sm) is new. Dm, distal medulla; Lai, lamina intrinsic; Li, lobula intrinsic, LPi, lobula plate intrinsic; Pm, proximal medulla. **c**, Families in the cross-neuropil tangential and amacrine classes. For tangential families, axon and dendrite are distinguished graphically. All are new except for Lat and Am1. LLPt, lobula–lobula plate tangential; LMt, lobula–medulla tangential; LMa, lobula medulla amacrine; Lat, lamina tangential; MLt, medulla–lobula tangential; PDt, proximal to distal medulla tangential.

A, anterior; L, lateral; M, medial; P, posterior. **d**, Cell types are ordered by the number of proofread cells in each type, starting with the most numerous types. Additional details are provided in Extended Data Tables 1 and 2. **e**, The numbers of families (left), types (middle) and cells (right) in each class. **f**, The numbers of types (left) and cells (right) in each neuropil-defined family. Bold font indicates families that are entirely new, or almost entirely new. MLLPa, medulla lobula lobula plate amacrine, is a synonym for Am1. **g**, The number of types versus the number of cells in a type. The x axis denotes type size (log-scale), and the y axis shows the number of types with matching size. The peak near 800 consists of the numerous types—those with approximately the same cardinality as the ommatidia of the compound eye.

## Type-to-type connectivity

We define a type-to-type connection matrix in which the $st$ element is the number of synapses from cell type $s$ to cell type $t$ (Methods). The matrix is visualized in Extended Data Fig. 4, and its numerical values can be downloaded (see the 'Data availability' and 'Code availability' sections).

The type-to-type connection matrix can also be visualized as a directed graph. As showing all connections is visually overwhelming, it is important to find ways of displaying meaningful subsets of

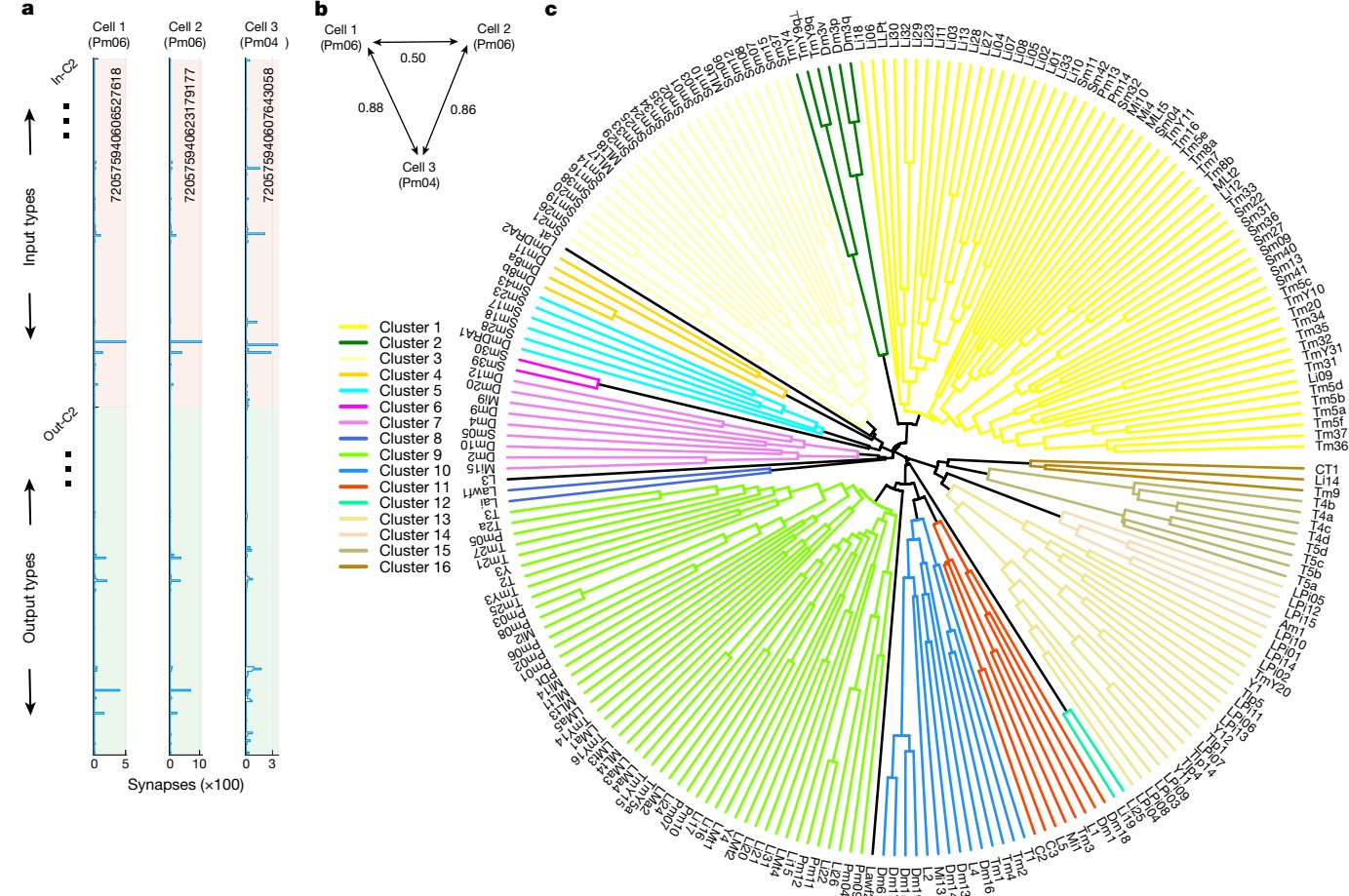

**Fig. 2 | Clustering of cells and cell types based on connectivity. a**, Feature vectors for three example cells. The horizontal axis indicates the synapse numbers that the cell receives from presynaptic types (red region of vertical axis) and sends to postsynaptic types (green region of vertical axis). Cells 1 and 2 (same type) have more similar feature vectors to each other than to cell 3 (different type). The long numbers are the cell IDs in version 783 of the FlyWire connectome. **b**, Cells 1 and 2 (same type) are closer to each other than to cell 3 (a different type), according to the weighted Jaccard distances between the cell feature vectors. Such distances are the main basis for dividing cells into cell types (Methods). **c**, Dendrogram of cell types. Cell types that merge closer to the circumference are more similar to each other. Flat clustering (16 colours) is created by thresholding at 0.9. A few clusters containing single types (Lat, L3 and Lawf2) are uncoloured. To obtain the dendrogram, feature vectors of cells in each type were summed or averaged to yield a feature vector for that cell type, and then cell type feature vectors were hierarchically clustered using average linkage. Jaccard distances run from 0.4 (circumference) to 1 (centre). Clusters containing more than one cell type (legend with coloured lines) are numbered starting at '3 o'clock' on the dendrogram and proceeding counterclockwise.

connections. One that we have found to be helpful is to display the top input and output connections of each type (Figs. 3–7 and Extended Data Figs. 5 and 6). In such a graph, some nodes can have more than one outgoing and/or more than one incoming connection. A few of these nodes show up as 'hubs' with many visible connections. For example, Mi1 is the top input to a large number of postsynaptic types (Fig. 3 and Extended Data Fig. 5).

The nodes of the graph were positioned in 2D space by a graph layout algorithm that tends to place strongly connected types close together (Methods). It turns out that nearby nodes in the 2D graph layout space tend to belong to the clusters that were extracted from the high-dimensional connectivity-based feature vectors (compare the node colourings of Fig. 3 with clusters of Fig. 2c).

We can also normalize the type-to-type connection matrix to be the fraction of synapses from cell type *s* to cell type *t*. Depending on the normalization, this could be the fraction of input to type *t* or fraction of output from type *s* (Methods). Input and output fractions are shown in Supplementary Data 5, and are equivalent to the cell type feature vectors defined earlier. The heat maps of Supplementary Data 5 are important because they show a much more complete set of connections than the wiring diagrams, which are highly selective visualizations.

## Perplexity as a measure of degree of connectivity

The degree of a cell type can be defined as the number of cell types to which it is connected. Weak connections can be excluded from this definition by thresholding the type-to-type connection matrix before computing degree. For a threshold-independent measure, we instead calculate a 'perplexity'[34] for each cell type. The outgoing connection strengths (synapse counts) are normalized as if they were a probability distribution, and out-perplexity is defined as the exponential of the entropy of this distribution. Out-perplexity reduces to out-degree in the special case that the distribution is uniform over the connected partners. In-perplexity is defined analogously.

If intrinsic cell types are ranked by the product of out- and in-perplexity (Extended Data Fig. 7a), then TmY5a is the most connected hub, and various types in the lamina and distal medulla are the least hub-like. Motion-related cell types generally do not have high perplexity. Out-perplexity tends to be greater than in-perplexity (Extended Data Fig. 7a), although they are positively correlated (Extended Data Fig. 7b).

One might expect that 'early' types in visual processing would have divergent connectivity, to distribute photoreceptor signals to many

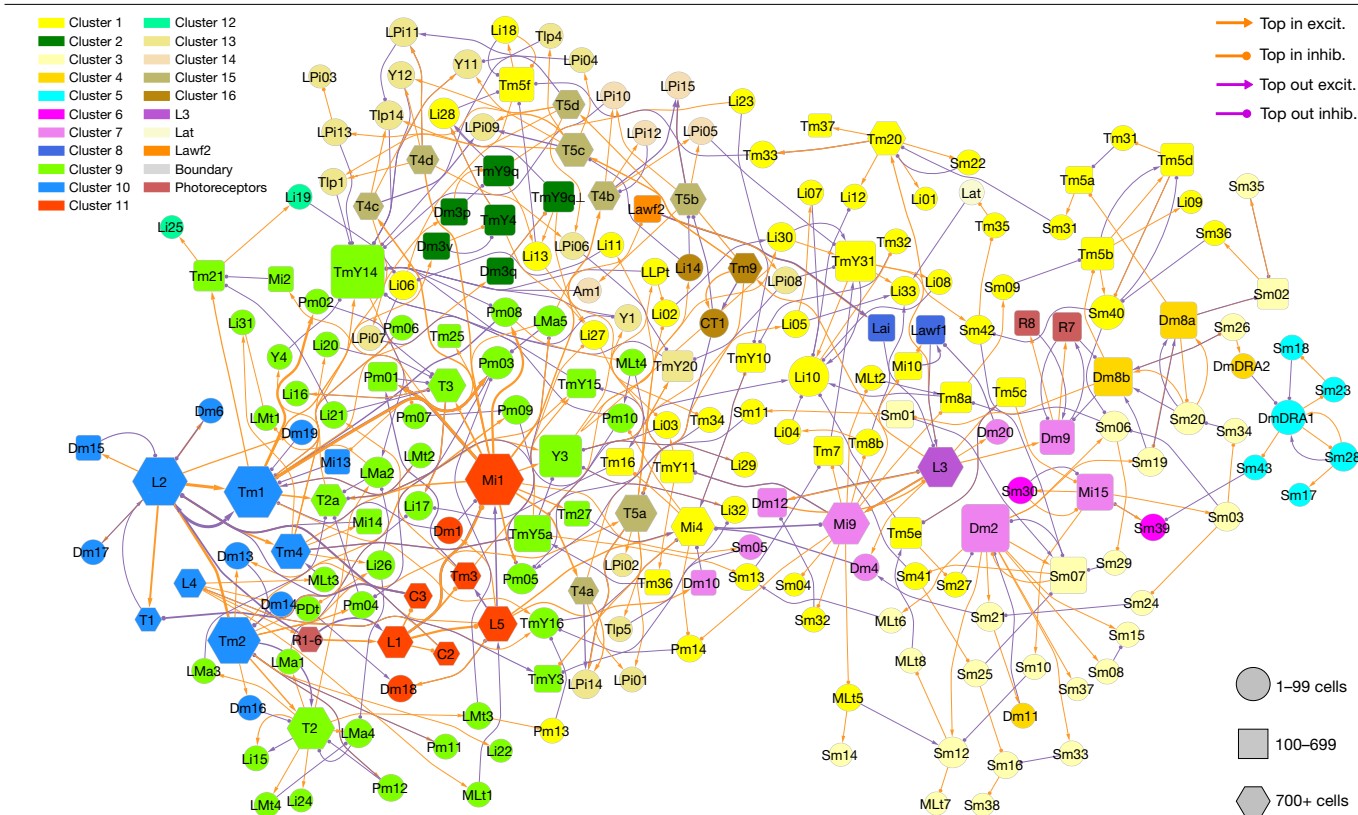

**Fig. 3 | Wiring diagram of cell types—top input and output connections.** Simplified wiring diagram of all cell types intrinsic to the optic lobe and photoreceptors, showing only the top input and output connections of each type. Colours of types (nodes) indicate membership in flat clusters of Fig. 2c. The node size encodes the number of drawn connections, so that hub types look larger. The node shape encodes type numerosity (number of cells). The line colour encodes the relationship (top input versus top output) and the line width is proportional to the number of synapses. The line arrowhead shapes encode excitation (excit.) versus inhibition (inhib.). Further explanation is provided in the Methods.

targets, while 'late' types would have convergent connectivity, summarizing the final results of optic lobe computations for use by the central brain. This idea can be tested by ranking types according to the ratio of out-perplexity to in-perplexity (Extended Data Fig. 8). Indeed, the top of the list includes early types like the inner photoreceptors R7 and R8, L3 and L5, and many Dm and Pm interneuron types, and many Sm types are near the bottom of the list; they can be interpreted as 'late' types given their extensive connectivity with VPNs.

## The 'numerous' cell types

Photoreceptor axons project retinotopically from the eye to the lamina (R1–6) and distal medulla (R7–8). The medulla is divided into columns, which are presumed to be in one-to-one correspondence with ommatidia of the compound eye. Cell types containing >720 cells in our reconstruction (Fig. 1d), as well as photoreceptor types, will be called 'numerous'. The top end (800) of this range is probably the true number of columns in this optic lobe. For each numerous type, the cells appear to be distributed one per column (Supplementary Data 2), and the true number of cells is expected to approximate 800. The observed cell numbers are mostly smaller than 800; some cells are missing from columns, presumably due to under-recovery of cells by proofreading (Methods). The connections between numerous types agree well with a previous reconstruction of seven medulla columns[27] (Methods and Extended Data Fig. 9).

The 28 numerous types have long been known[6]. At the other extreme, 16 types contain only a single cell. Most types (183) lie between the extremes (Fig. 1g and Extended Data Fig. 1d). It is the less numerous types of which our knowledge has been incomplete, and arguably they are where much of the magic of vision happens. As with the

photoreceptors, neural activity in the numerous cell types like L1 and Mi1 mostly encodes information about the image at or near single points in visual space. But perception requires the integration of information from points that can be quite distant from each other, and this is done by the larger neurons that belong to the less numerous types.

For most of the numerous types, visual responses have been observed previously[4], and will be used to interpret the dendrogram of Fig. 2c. We will see that the numerous types that belong to a single cluster have similar functions, which enables us to ascribe a function to each cluster as a whole. In other words, we extrapolate from the functions of the numerous types to yield preliminary clues regarding the functions of the less-numerous types.

These extrapolations are speculative, and are merely starting points for hypothesis generation and experimental research, and the clusters are not set in stone. They were obtained by thresholding a hierarchical clustering (Fig. 2c), and adjusting this threshold will change the number of clusters (Extended Data Fig. 10). Rather than use our clusterings, some readers may prefer to directly consult the weighted Jaccard distances between types (Fig. 2, Source Data), from which the clusterings were derived. Other cautionary notes about the clusters are given in the Methods and Discussion. These caveats notwithstanding, we next proceed to functional interpretation of the clusters in Fig. 2c.

## ON, OFF and luminance channels

Cluster 10 and cluster 11 (Fig. 2c) both receive strong input from photoreceptors R1–6 (Extended Data Fig. 11), and we propose that they are regarded as OFF and ON channels, respectively, carrying information about light decrements (OFF stimuli) and light increments (ON stimuli). Our concept is similar to the well-known ON and OFF motion

pathways[35,36], but differs because our ON and OFF channels are general purpose, feeding into the object and colour subsystems as well as the motion subsystem.

Cluster 10 contains the OFF cells L2, L4, Tm1, Tm2 and Tm4. Cluster 11 contains the ON cells L5, Mi1 and Tm3, and also the OFF cell L1. It makes sense to assign L1 to the ON channel even though it is an OFF cell, because L1 is inhibitory/glutamatergic, so its effects on downstream partners are similar to those of an ON excitatory cell. Note that information about whether synapses are excitatory or inhibitory was not used by our clustering algorithm. Cluster 11 also contains C2 and C3, which are expected to be ON cells because their top inputs are L1 and L5. A companion paper argues that the various Dm interneuron types in cluster 10 and cluster 11 normalize the activities of numerous types in the OFF and ON channels[37].

The ON and OFF motion pathways were traditionally defined by working backwards from the T4 and T5 motion detectors, which respectively compute the directions of moving ON and OFF stimuli[4,5]. The ON motion pathway is directly upstream from T4 and includes Mi1, Mi4, Mi9 and Tm3. The OFF motion pathway is directly upstream from T5 and includes Tm1, Tm2, Tm4 and Tm9. Figure 4 shows that these cell types have other strong targets besides T4/T5, so they do not seem to be solely or chiefly dedicated to motion (see below concerning the lone exception Tm9).

L3 connectivity is sufficiently unique that it stands apart from all of the other cell types as a cluster containing only the single type L3 (Fig. 2c). This is consistent with current thinking that L3 constitutes a separate luminance channel, distinct from ON and OFF channels[38]. L3 is the only L type with a sustained rather than transient response[39], and it encodes luminance rather than contrast[40].

Cluster 7 includes Dm4, Dm9, Dm12, Dm20 and Mi9, which all have L3 as their strongest input. Mi9 is also the strongest output of L3 and, like L3, exhibits a sustained response[41]. We therefore propose that cluster 7 should be lumped with L3 in a hypothetical luminance channel. Mi9 is traditionally grouped in the ON motion pathway, but Mi9 is an input to the object and colour subsystems, not only the motion subsystem. It is less obvious whether the remaining types in cluster 7 (Mi15, Dm2, Dm10 and Sm05) should be grouped in the luminance channel. Indeed, these types break off into a separate clusters when the threshold is adjusted to refine the flat clustering (Extended Data Fig. 10). These types might alternatively be assigned to the colour subsystem as Mi15 and Dm2 are known to receive direct input from inner photoreceptor R8[42].

Lawf2 is a cluster of its own. By targeting cell types (L5, C2 and C3 in Supplementary Data 5) in cluster 11, Lawf2 provides centrifugal feedback to the ON channel (Extended Data Fig. 11). However, the strongest output of Lawf2 is Lai (Fig. 4), which is thought to mediate lateral inhibition in the lamina[43] through pathways such as R1–6→Lai→R1–6 and R1–6→Lai→L3[26]. Lawf2 may therefore modulate lateral interactions mediated by Lai. The strongest input to Lawf2 is OA-AL2b2, which could be octopaminergic or cholinergic[44,45]. If it is octopaminergic, this input could be the source of the previously reported octopaminergic gain modulation of Lawf2 neurons[46]. Lawf2 also receives strong input from cluster 9, which is hypothesized to be an object subsystem later on.

Lai and Lawf1, the two types in cluster 8, have similar targets (L3, T1, R1–6 and L2). Cluster 8 provides centrifugal feedback to the OFF channel (through L2) and to R1–6 (Extended Data Fig. 11). Alternatively, cluster 8 could be interpreted as being part of the luminance channel, as cluster 7 is a strong input and L3 a strong output (Extended Data Fig. 11).

## Motion

The motion-detecting T4 and T5 families belong to cluster 15 (Fig. 2c). Cluster 16 contains CT1 and Tm9, which are well known to be important for motion computation[4,5]. It makes sense to regard Tm9 as dedicated to the motion subsystem rather than part of a general-purpose OFF

channel, as 80% of its output synapses are onto CT1 or T5. Cluster 16 also includes Li14, an interneuron type with T5a as the strongest input, and T5a through T5d as the strongest outputs. T4/T5 neurons synapse onto VPNs that exit the optic lobe and enter the central brain (Fig. 5a and Supplementary Data 5).

Cluster 13 and cluster 14 contain the lobula plate interneuron family, LPi1 through LPi15[6,8]. Over half of these are new (Methods). Some LPi types consist of one or two cells that cover the entire visual field (Fig. 5b). Two LPi types may stratify in the same lobula plate layers, but consist of cells with different sizes (Fig. 5c). Most LPi types are amacrine, but some exhibit axo-dendritic polarization (Fig. 5d). Some types collectively cover only a portion of the visual field (for example, LPiO1 and LPiO3 are ventral only; Supplementary Data 2).

All LPi types receive input from T4/T5 types, so it is clear that cluster 13 and cluster 14 are related to motion vision. All LPi types receive input from T4/T5 cells with a single preferred direction (Fig. 5a and Supplementary Data 5). The only exception is LPi07, which receives inputs from T4/T5 cells with preferred directions *c* and *d* (Supplementary Data 5). LPi types synapse onto other LPi types and onto VPNs (Fig. 5a and Supplementary Data 5).

Cluster 13 also contains columnar neurons from three Y types and all Tlp types. All of these are predicted to be glutamatergic, and are reciprocally connected with T4/T5 of particular preferred directions. The only exception is Tlp5, which receives input only from T4a/T5a. The Y and Tlp types also connect with LPi and columnar VPN types[10]. TmY20 and Am1 also belong to cluster 13, and were previously identified to be motion related[10].

## Objects

Cluster 9 includes the numerous types T2 and T3, which have been implicated in the detection of small objects[47]. Their downstream VPN partners LC11[47] and LC18[48] (Fig. 6) are also activated by small objects. On the basis of this information, we propose that cluster 9 is part of a hypothetical object subsystem (Fig. 6). Cluster 9 (Fig. 2c) includes many other types from columnar families (Mi, TmY, Y), interneuron families (Li and Pm) and cross-neuropil tangential and amacrine families (LMa, LMt, MLt, PDt). Downstream targets include LC, LPLC and LT types (Fig. 6).

Mi1 and Tm1 are the most prominent inputs to the subsystem (Fig. 6), and respectively belong to the ON and OFF channels defined above. They are top inputs to T3, explaining why T3 is ON–OFF[47]. T2 is ON–OFF because its top inputs are L5 and Tm2, which respectively belong to the ON and OFF channels. Note that the Tm1 input to T2 and the L5 input to T2 are second from the top, and therefore do not show up in Fig. 6, which is restricted to the top inputs and outputs.

Several types are nearby T2 and T3 in the cell types dendrogram (Fig. 2c). In particular, T2a, Tm21, Tm25, Tm27, TmY3 and Y3 are fairly numerous and excitatory, so we regard them as candidate object detectors. Despite its name, T2a is more similar to T3 in connectivity than to T2 (Fig. 2c). T2a also receives Mi1 and Tm1 input like T3, and is predicted to be ON–OFF. The top output of T2a is LC17, which is known to be activated by small objects[49] and also receives input from T3.

Cluster 12 contains Li19 and Li25 (Fig. 2c). Cluster 9 is both a strong input to cluster 12 (Extended Data Fig. 11) and a strong output of cluster 12 (Extended Data Fig. 11), largely due to connections between Tm21 and Cluster12. We therefore include cluster 12 as well as cluster 9 in the object subsystem.

## Colour and polarization

The inner photoreceptors R7 and R8 are important for *Drosophila* colour vision because their responses are more narrowly tuned to the wavelength of light than those of the outer photoreceptors R1–6. R7 prefers ultraviolet light, whereas R8 prefers blue or green light[20].

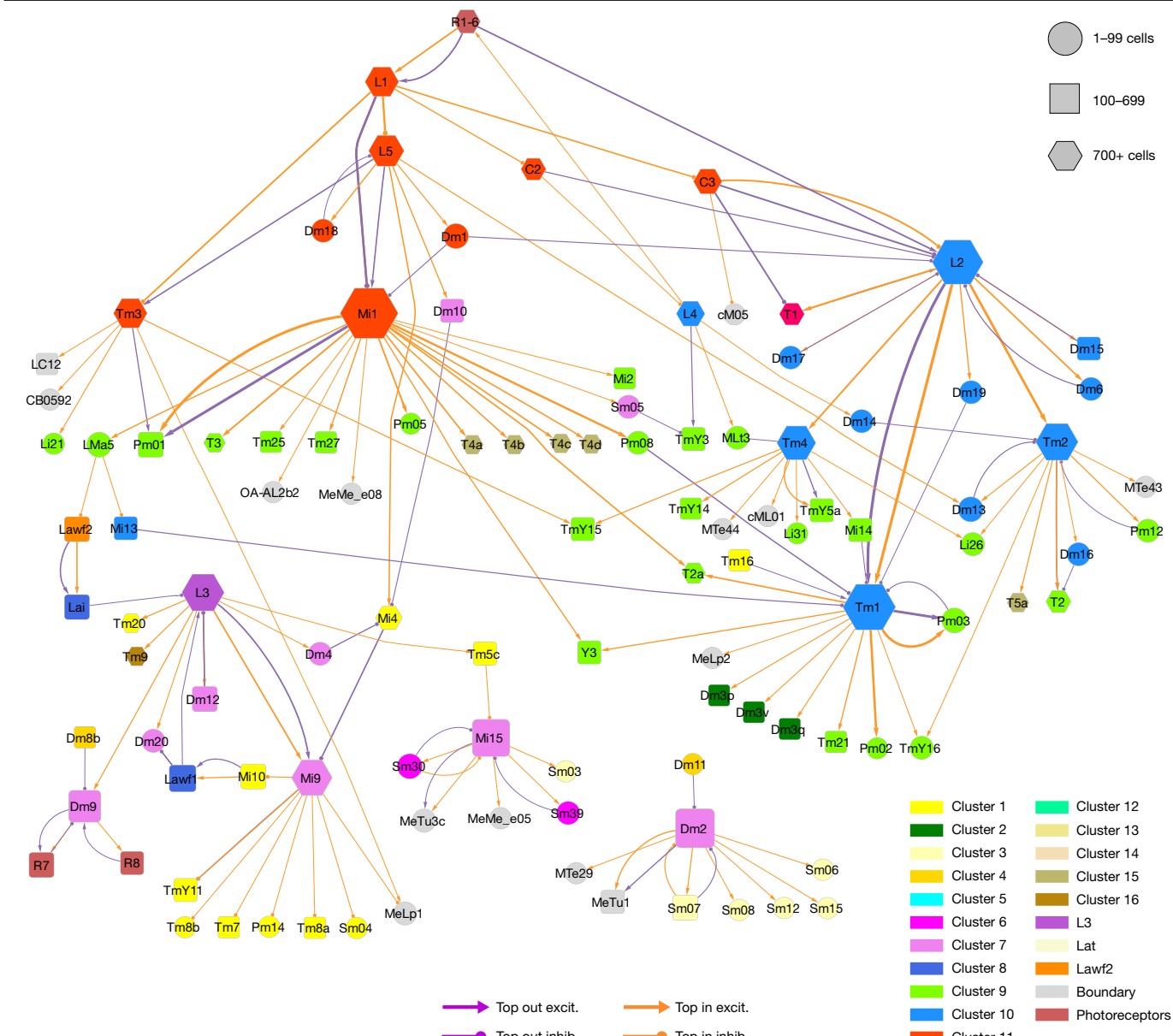

**Fig. 4 | ON, OFF and luminance channels—top inputs and outputs only.** Simplified wiring diagram of ON (cluster 11, red), OFF (cluster 10, blue) and luminance (cluster 7, violet and L3) channels and their primary connections with other subsystems and VPNs. For clarity, only the top input and output connections are shown for each type. Further explanation is provided in Fig. 3 and the Methods.

Cluster 4 contains Dm8a, Dm8b, Dm11 and DmDRA2, which are all inner photoreceptor targets[42]. Cluster 1 contains most of the remaining types so far implicated in colour vision. As originally defined by morphology[6], Tm5 is a potential postsynaptic target of the inner photoreceptors because it stratifies in the distal medulla at the M7 border and also in the M3. These are the medulla layers containing the axon terminals of R7 and R8[7]. We found that Tm5 consists of six cell types (Fig. 7a). Three of our connectomic Tm5 types correspond to canonical Tm5 types that were previously defined by morphology and Ort expression[7,50]. Tm5a and Tm5b receive R7 input, while Tm5c receives R8 input. Moreover, we found three new types, Tm5d, Tm5e and Tm5f, that receive little or no photoreceptor input, although their stratifications are similar to those of the canonical Tm5 types (Fig. 7a).

The correspondences between connectomic and morphological-molecular Tm5 types were established using morphological criteria (Methods). However, the reader should be cautioned that there is considerable variability within a type, so reliably typing individual cells based on morphology alone is difficult or impossible. Connectivity is essential for reliable discriminations.

Tm5a and Tm5b receive R7 and Dm8 input, as expected from previous reports[42,50,51]. Tm5c receives R8 input[42,50], and also strong L3 input (Fig. 7c and Supplementary Data 5). While some synapses from Dm8 to Tm5c do exist[50], this connection seems to be weak.

Tm20 has been implicated in colour vision because it receives R8 input[27,28,42]. It also receives strong L3 input (Fig. 7c). Thus, Tm20 inputs are similar to Tm5c inputs, consistent with the physiological finding that these two types are more similar to each other in their chromatic responses than they are to Tm5a and Tm5b[52].

As Tm5a, Tm5b, Tm5c and Tm20 are known to be related to colour vision, we propose that the rest of cluster 1 is also part of a hypothetical colour subsystem (Fig. 7c). The new Tm5 types (Tm5d, Tm5e and Tm5f) receive few or no synapses directly from photoreceptors, but Tm5d receives indirect R7 input from Tm5b and Dm8a, Tm5e receives indirect R8 input from Tm5c (Fig. 7c), and Tm5f receives indirect R8 input from

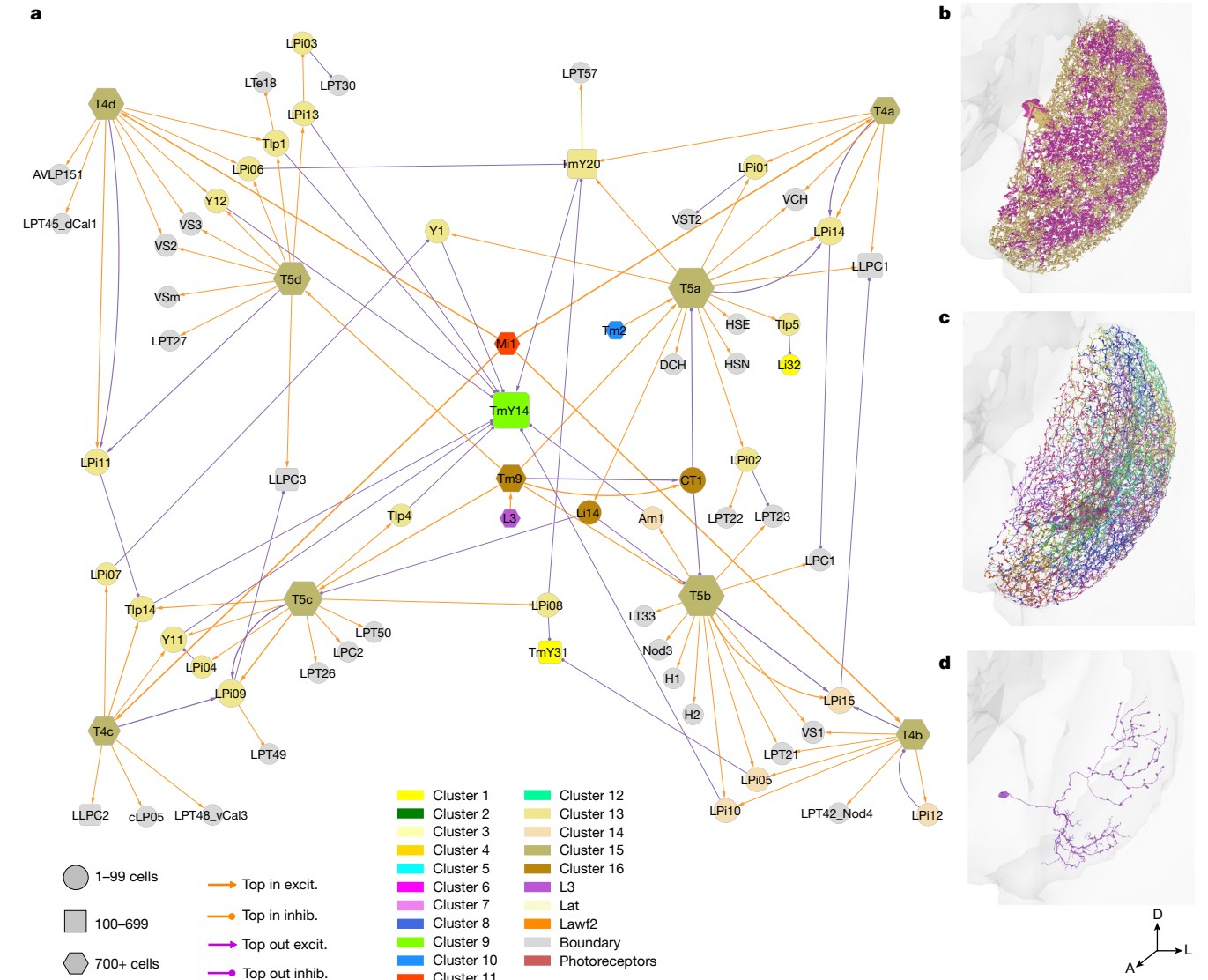

**a**

**b**

**c**

**d**

| | | |
|---|---|---|
| Cluster 1 | Cluster 12 | |
| Cluster 2 | Cluster 13 | |
| Cluster 3 | Cluster 14 | |
| Cluster 4 | Cluster 15 | |
| Cluster 5 | Cluster 16 | |
| Cluster 6 | L3 | |
| Cluster 7 | Lat | |
| Cluster 8 | Lawf2 | |
| Cluster 9 | Boundary | |
| Cluster 10 | Photoreceptors | |
| Cluster 11 | | |

1–99 cells
100–699
700+ cells

Top in excit.
Top in inhib.
Top out excit.
Top out inhib.

D, Dorsal. Scale bar, 30 μm.

**Fig. 5 | Motion subsystem—top inputs and outputs only. a**, Cell types of the motion subsystem (clusters 13 to 16) and their primary connections with other subsystems and VPNs. The motion-detecting T4 types are located at the corners of the square layout, and often share postsynaptic targets with the corresponding T5 types. TmY14 is the top output of many types. For clarity, only the top input and output connections are shown for each type. Further explanation is provided in Fig. 3 and the Methods. **b**, LPi14, also called LPi1-2[10], is a jigsaw pair of full-field cells. **c**, LPi02 stratifies in the same lobula plate layers as LPi14, but the cells are smaller. **d**, LPi08 is an example of an interneuron that is not amacrine. It is polarized, with a bouton-bearing axon that is dorsally located relative to the dendrite. D, Dorsal. Scale bar, 30 μm.

Tm20 (Supplementary Data 5). Tm5d and Tm5e are predicted to be glutamatergic and Tm5f is predicted to be cholinergic.

We have defined Dm8a and Dm8b, which synapse onto Tm5a and Tm5b, respectively (Fig. 7c), and this preference is highly selective (Supplementary Data 5). As with Tm5, splitting Dm8 is straightforward with connectivity but difficult or impossible with morphology. How our two Dm8 types correspond with the two types previously defined by molecular studies (yDm8 and pDm8)[51,53] remains speculative (Methods).

Cluster 1 also includes Tm7, Tm8a and Tm8b (another novel split), Tm16 and wholly new types Tm31 to Tm37. The latter deviate from the classical definition of the Tm family, which is supposed to project from the distal medulla to the lobula[6]. These types mainly stratify in serpentine medulla and lobula, with little or no presence in distal medulla (Fig. 7b). Nevertheless, we decided to lump them into the Tm family. Tm31 to Tm35 each contain relatively few (<100) cells, and are predicted to not be cholinergic. This departs from the norm for existing Tm types,

which are generally more numerous (>100 cells) and predicted to be cholinergic (exceptions are the three glutamatergic Tm5 types). Tm36 and Tm37 contain more than 100 cells each, and are predicted to be cholinergic.

Cluster 1 includes TmY types, Li, Sm and Pm interneuron types, MLt types and LLPt. Cluster 1 also includes Mi4 and Mi10. Mi4 was traditionally regarded as part of the ON motion pathway, but T4 cells are relatively weak outputs. Mi4 has strong partners in the colour and object subsystems (Fig. 7c (yellow and green)). Its strongest output is Mi9, which we have assigned to the luminance channel and is one of the major inputs to the colour subsystem. This diversity of targets shows that Mi4 is a major hub between multiple subsystems, although it has been assigned by the clustering to a single subsystem. Mi10 mediates a feedback loop L3→Mi9→Mi10→Lawf1→L3, so it might seem to belong to the luminance channel, but the clustering has placed it in cluster 1 because it is similar in connectivity to Mi4.

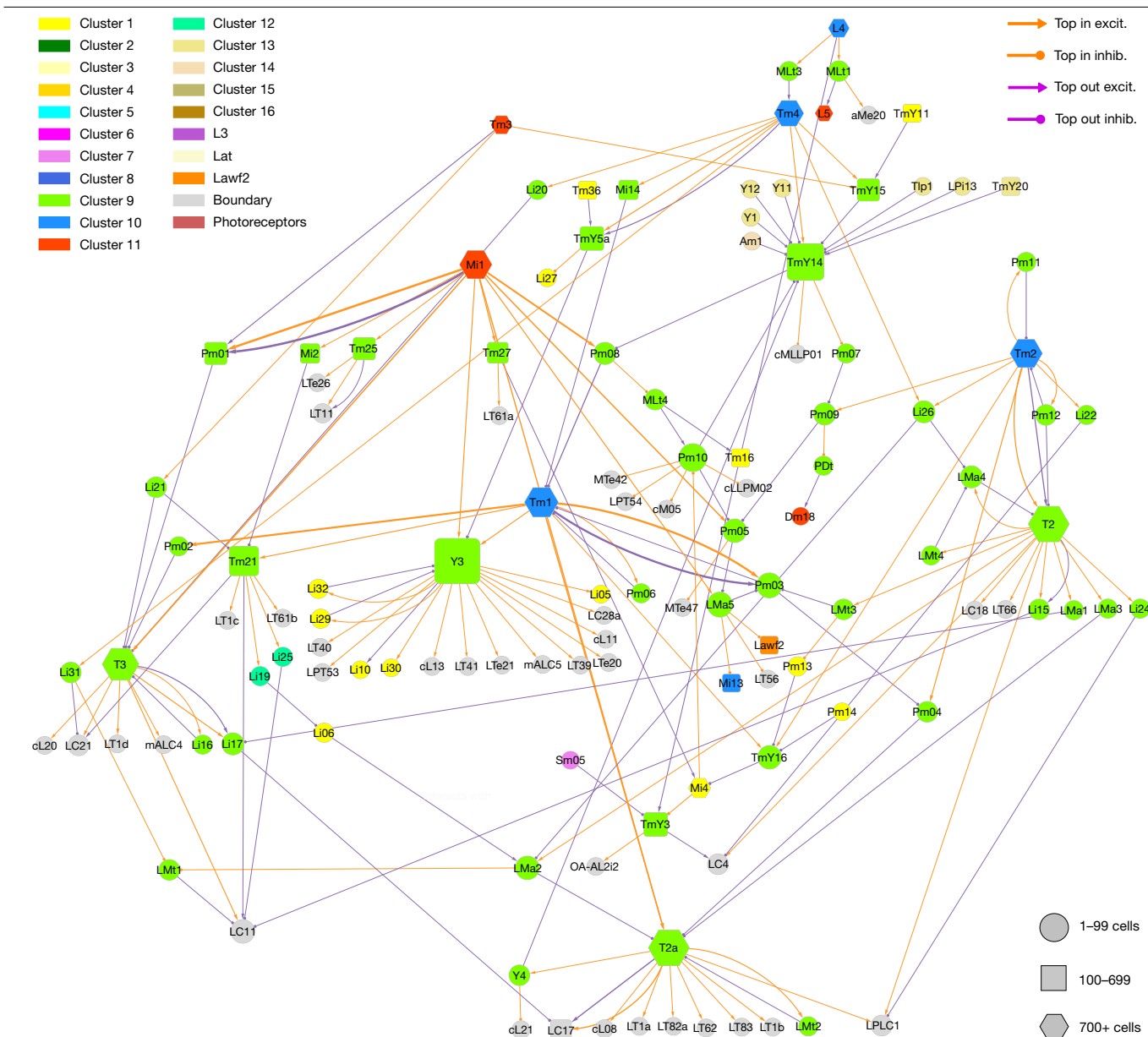

**Fig. 6 | Hypothetical object subsystem.** Cell types of the object subsystem (clusters 9 and 12) and their primary connections with other subsystems and VPNs. T2 and T3 are known to be activated by small objects[47]. For clarity, only the top input and output connections are shown for each type. Further explanation is provided in Fig. 3 and the Methods.

Besides L3, Mi9 is another prominent input to the colour subsystem (Fig. 7c). Both L3 and Mi9 belong to the luminance channel defined above. It makes sense that luminance information should be necessary for colour computations[38].

Cluster 3 consists mainly of a large number of Sm interneuron types (Fig. 2c). It is well-connected with cluster 1 (Extended Data Fig. 11), so we also include it in the hypothetical colour subsystem (Fig. 7c).

Cluster 5 contains DmDRA1, a cell type at the dorsal rim of the medulla that is known to be important for behaviours that depend on skylight polarization[54]. Cluster 4 is therefore regarded as part of the polarization subsystem. It contains several Sm types, most of which are either situated at the dorsal rim or have some specialization there.

## Morphological variation

As mentioned above, connectivity can be essential for distinguishing between types with similar morphologies. Connectivity can also enable one to ignore morphological variations between cells of the same type. For example, TmY14 was originally identified as a cell type intrinsic to the optic lobe[27], but later reclassified as a VPN, because it typically projects to the central brain[55]. In another twist, our optic lobe turns out to contain atypical TmY14 cells that lack the central brain projection (Fig. 8a,b). In cases like this, we double check the proofreading before concluding that this is true biological variation. Even in typical TmY14 cells, the axon has few synapses and minimal impact on connectivity, so TmY14 has reverted to its original status of being intrinsic to the optic lobe (an explanation of the threshold is provided in the Methods). TmY14 ends up as a single type in our connectivity-based clustering, because typical and atypical TmY14 cells have similar connectivity within the optic lobe.

Another interesting example is Tlp4 versus Y11, which have similar connectivity patterns (Fig. 2c and Supplementary Data 5). A major difference is that Tlp4 cells, by definition, have no connectivity in the medulla. However, a few of them do, and look like they do not belong

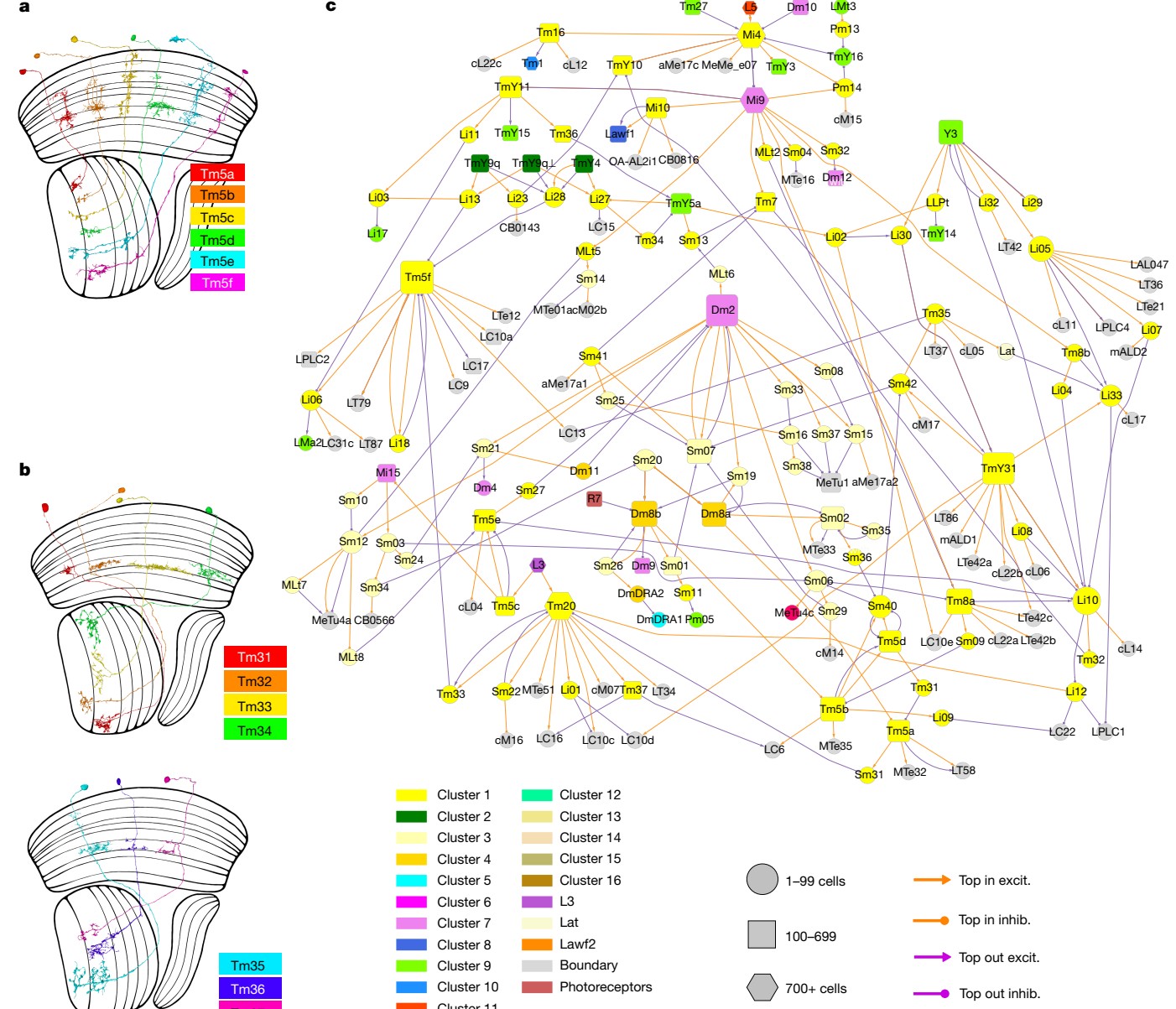

**Fig. 7 | Hypothetical colour subsystem. a**, Tm5a to Tm5c correspond with types that were previously defined by molecular means. Tm5d to Tm5f have similar morphologies, but different connectivity patterns (Supplementary Data 5). **b**, Tm31 to Tm37 are new members of the Tm family that project from the serpentine layer (M7) to the lobula. **c**, Cell types in the colour subsystem (clusters 1, 3 and 4) and their top connections with other subsystems and VPNs. For clarity, only the top input and output connections are shown for each type. Further explanation is provided in Fig. 3 and the Methods.

in Tlp4 (Fig. 8c,d). In the first stage of morphology-based classification, these errant cells were assigned to Y11. But such pseudo-Y11 cells were later reassigned to Tlp4 on the basis of connectivity. Their feature vectors match Tlp4 because their medullary projections make few synapses, and their connectivity in the lobula and lobula plate matches Tlp4.

It is worth mentioning an unusual example in which ignoring morphological variation is correct in one sense, but ultimately turns out to be misleading. Three Li11 cells are annotated in the hemibrain reconstruction[9], and three corresponding cells can be identified in our optic lobe[25]. We group two of these cells in one type (Fig. 8e). The third cell can be paired with a fourth to form a pseudo-Li11 type with a small axonal projection into the central brain (Fig. 8f). Although the axon is visually striking, it has few synapses and therefore little impact on connectivity. Thus, it might be tempting to ignore the axon as a developmental

'accident' and merge Li11 and pseudo-Li11 into a single type. But it turns out that Li11 and pseudo-Li11 are distinct types, owing to their different connectivity in the lobula. For example, Li25 has strong LT61 output, while pseudo-Li11 has strong LT11 input. Pseudo-Li11 also exists in the hemibrain (data not shown), although there it lacks the small projection. So the central brain projection of pseudo-Li11 exhibits variability across individuals, further evidence that it is a developmental accident. We introduce the new names Li25 and Li19 to replace Li11 and pseudo-Li11,

A few cells were dismissed as developmental accidents. This could be done with high confidence when the cells were small and few in number. However, we had difficulty deciding about Li29 because it was a full-field cell in the lobula but it also extended a smaller secondary arbour into the lobula plate (Supplementary Data 2). Originally, we decided that this cell was a developmental accident, and did not include it in our

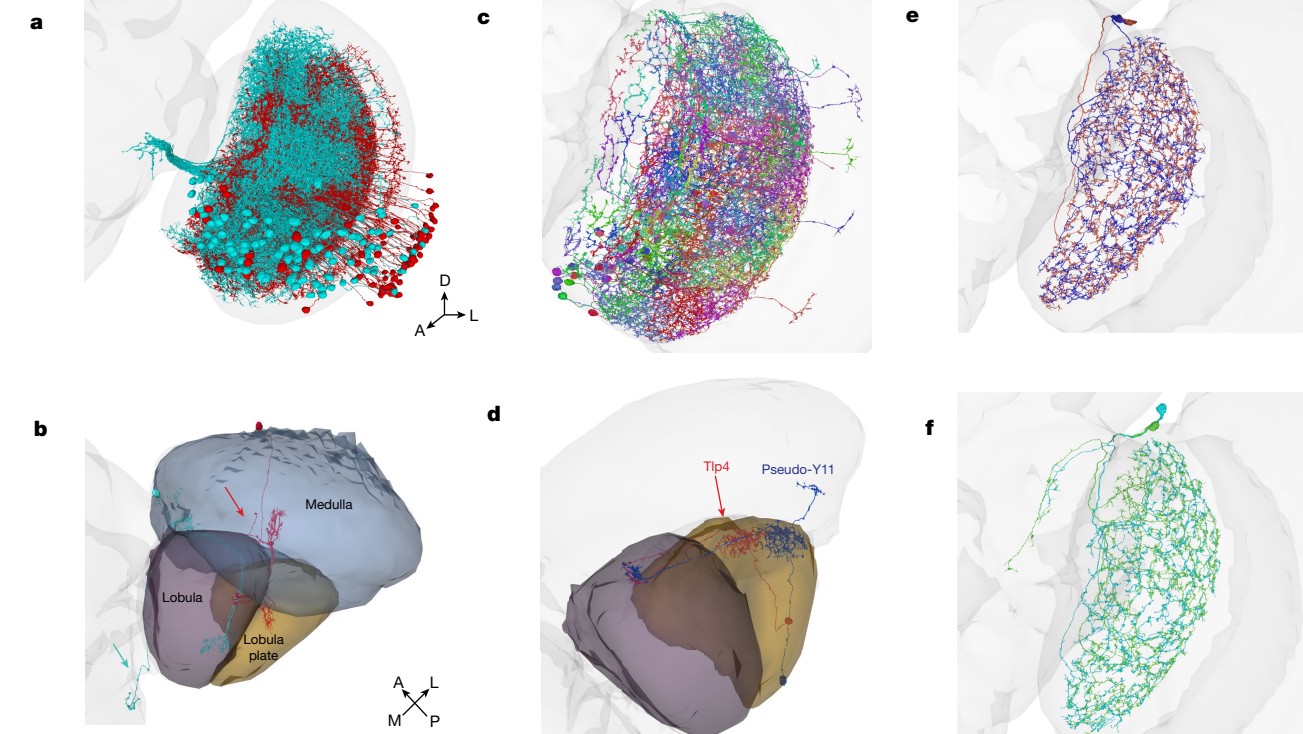

**Fig. 8 | Morphological variation. a**, Typical TmY14 cells (cyan) have axonal projections to the central brain (left). Atypical cells (red) initially project toward the central brain, but their axons turn around and terminate in the medulla. As the axons bear few synapses, typical and atypical cells are approximately the same in connectivity. **b**, Representative typical (cyan) and atypical (red) TmY14 with an axon projecting into the central brain (cyan arrow) and medulla (red arrow), respectively. **c**, Typical Tlp4 cells arborize in the lobula plate and lobula. A few cells (pseudo-Y11) have an additional branch in the medulla (right), and resemble Y11 cells in morphology but have the same connectivity as Tlp4. **d**, Relative to a typical Tlp4 cell (red), a pseudo-Y11 cell (blue) has an additional branch in the medulla. **e**, Li11 does not project into the central brain. **f**, Pseudo-Li11 has an additional arbour projection into the central brain. This arbour makes a few synapses, and might lead to the conclusion that pseudo-Li11 should be categorized as Li11. However, the connectivity between Li11 and pseudo-Li11 is fundamentally different, making them distinct types. Scale bar, 30 μm.

list of types. Later on, we found that this odd-looking cell is repeated in the left optic lobe, and promoted it to a type.

## Spatial coverage

All cell typing efforts must decide whether to split types more finely or merge types more coarsely. We resolved this lumper–splitter dilemma by using spatial coverage as a criterion[2]. As a general rule, the cells of a cell type collectively cover all columns of the optic lobe with a density that is fairly uniform across the visual field. This makes sense for implementing translation-invariant computations, a strategy that is commonly used in convolutional networks and other computer vision algorithms. Uniform spatial coverage is sometimes called 'tiling', although cell type arbours often overlap so much that the analogy to floor tiles is misleading. Spatial coverage is also a property of many cell types in mammalian retina[2,56].

In some types consisting of just one or a few cells, we identified an unconventional jigsaw-style spatial coverage. For example, LPi14, also known as LPi1-2[10], is a pair of full-field cells (Fig. 5b). We refer to them as a jigsaw pair because they jointly cover the visual field in an irregular manner, as if they were cut by a jigsaw. Jigsaw types can also be found in other interneuron families and include Pm14, Li27 and Li28.

Our feature vector (Fig. 2a) includes no explicit information about the spatial coordinates of a cell. Thus, if clustering feature vectors results in cell types with good spatial coverage, that is an independent validation of the clustering. Coverage also solves the lumper–splitter dilemma. Suppose that we attempt to split one type into two candidate types, based on hierarchical clustering. If both candidate types exhibit good coverage, then we accept them as valid. If the cells of both candidate types seem randomly scattered, that means our split is invalid, because it is presumably discriminating between cells based on noise. Chromatic types like Tm5 and Dm8 might seem to be an exception to this rule, but their apparently random locations may turn out to depend systematically on pale and yellow columns (Methods).

The above are easy cases, but there are also edge cases. Suppose that splitting results in two candidate types that neatly cover the dorsal field and the ventral field, respectively, without overlap. We then reject the split, preferring to lump the two candidate types in a single type that exhibits dorsoventral spatial variation in connectivity. On the other hand, if one candidate type covers the dorsal field and the other covers the full field, this is an acceptable split.

With these heuristics, some of our cell types end up with only partial coverage of the visual field (Fig. 9). This is especially common for boundary types. Sm is the intrinsic type family containing the most types with partial coverage. This makes sense, given that Sm cells interact closely with many boundary types arborizing in the serpentine layer. Cell types with partial coverage make sense in the later stages of vision. After the early stages of vision, computer vision also often discards translation invariance and may perform different visual computations in different regions of the visual field.

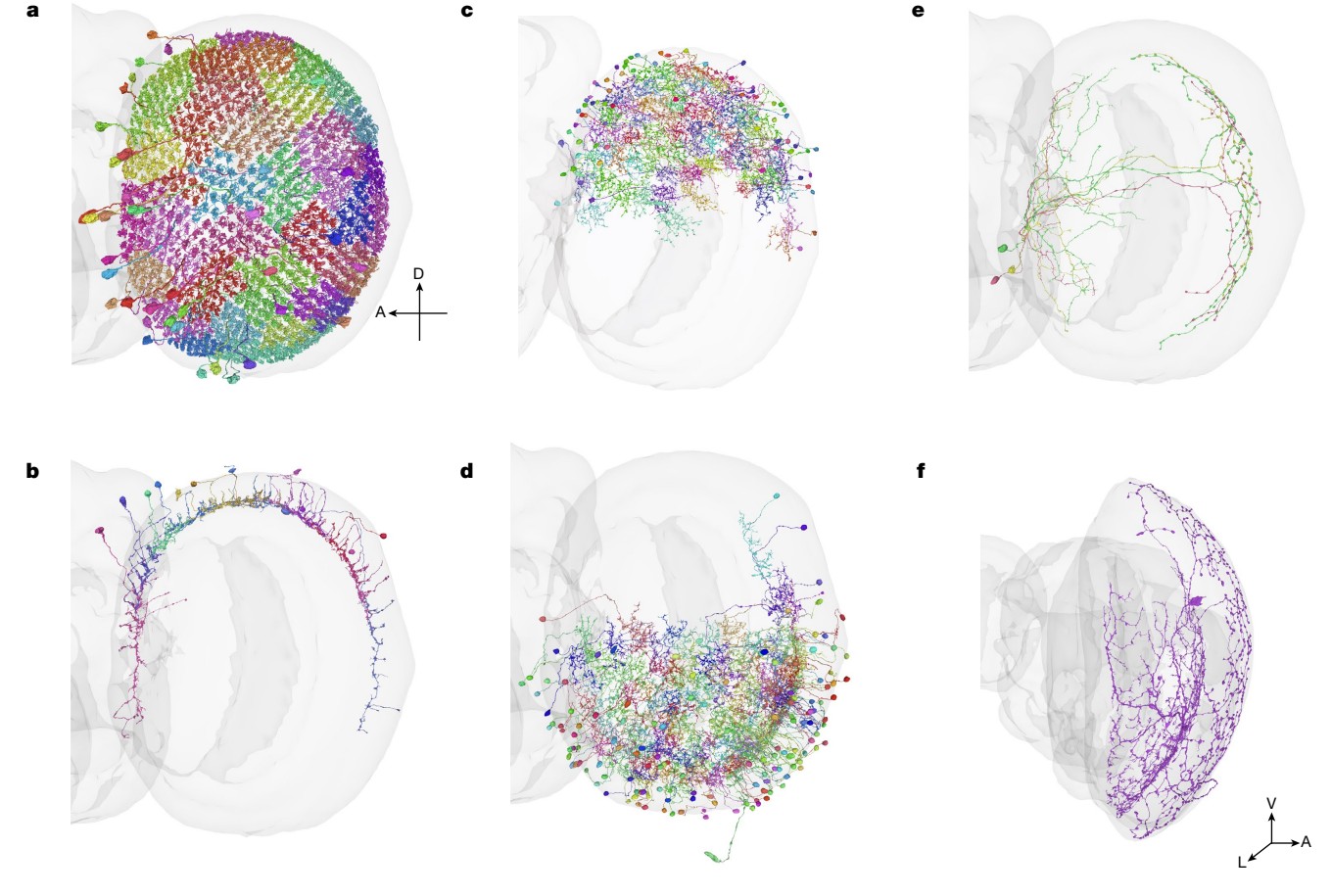

**Fig. 9 | Different kinds of spatial coverage. a**, Dm4 has full spatial coverage, and tiles perfectly with no overlap. **b**, Dm dorsal rim area 2 (DmDRA2) covers the dorsal rim. **c**, Sm05 covers the dorsal hemifield. **d**, Sm01 covers the ventral hemifield. **e**, Sm33 are H-shaped cells that cover the anterior and posterior rim. **f**, Sm39 is a single cell with mixed coverage: dorsal dendritic arbour in M7 and full-field axonal arbour in M1. V, ventral. Scale bar, 50 μm.

## Discussion

The connectomic approach to cell typing has three powers. First, it is not subject to the incomplete and biased sampling that can affect other methods. Second, connectivity turns out to provide a rich set of features for distinguishing between cell types. Third, connectomic cell typing not only yields cell types, but also, importantly, tells us how they are wired to each other.

### Implications for visual function

We clustered cell types with similar connectivity patterns (Fig. 2c), and proposed tentative interpretations of the clusters in terms of visual functions. These interpretations are speculations, but should be useful for generating hypotheses that suggest interesting experiments. Our hypothetical subsystems are devoted to motion, object and colour vision (Figs. 5–7), and are fed by ON, OFF and luminance channels (Fig. 4).

The motion subsystem (clusters 13–16) contains not only the T4 and T5 families but also many interneuron types. Most interneuron types belong to the LPi family, which has been proposed to mediate opponent interactions between cells that are activated by different directions of motion[8]. Such opponency was demonstrated between LPi09 and LPi11, also known as LPi3-4 and LPi4-3[57]. It is likely that LPi types can also mediate spatial normalization, as described in a companion paper[37].

Of the 51 types in the hypothetical object subsystem (clusters 9 and 12), T2 and T3 have been characterized by physiologists as object detectors[47]. Above we hypothesized that a number of other types (T2a, Tm21, Tm25, Tm27, TmY3 and Y3) are object detectors, and these candidates can be tested by future experiments.

The hypothetical colour subsystem (clusters 1,3 and 4) contains 91 types. One can only speculate about the reason for this numeric preponderance. Some insects are known to have sophisticated colour vision capabilities such as colour constancy[58]. The computations required for colour constancy are quite complex, requiring the integration of image information over long ranges[59]. This could potentially be implemented by the large number of Sm and Li interneuron types in the hypothetical colour subsystem, assuming that *Drosophila* turns out to exhibit colour constancy. Alternatively, it is possible that cluster 1 and cluster 3 have additional functions other than colour vision, and should be subdivided more finely (Extended Data Fig. 10). Future experiments will be needed to test these hypotheses.

A companion paper predicts that the six types in cluster 2 (Fig. 2c) should exhibit orientation selectivity[60], and hypothesizes that cluster 2 is a subsystem for form vision. Cluster 2 connects to cluster 1 (Extended Data Fig. 11), suggesting an interaction between form and colour computations.

Although we have carved the optic lobe into distinct subsystems, we are aware that it is simplistic to assign every cell type to just one functional subsystem. This is the result of the 'hard' clustering algorithm that we have used, which always assigns a cell type to a single cluster.

In reality, a cell type could have more than one function, or a cell type might mediate interactions between more than one subsystem. The wiring diagrams show many connections between cell types in different subsystems (Figs. 3–7 and Supplementary Data 5). Assigning such a cell type to a single subsystem is inherently ambiguous.

## Implications for visual development

The detailed wiring diagram for an adult visual system precisely specifies the end goal of visual system development. Single-cell transcriptomics is providing detailed information about the molecules in fly visual neurons[61–63]. Comparison of transcriptomic and connectomic information is already uncovering molecules that are important for the development of the fly visual system[64], and this trend is bound to increase in momentum. Such research could be aided by our low-dimensional discriminators of cell types (Supplementary Data 4 and Extended Data Fig. 3).

## Complete and unbiased

Early studies[6,32] relied on Golgi staining to sample neurons from multiple individuals, a technique that is best suited for identifying the most numerous types. Most of our new types are not as numerous (10 to 100 cells), which may be why they were missed. Furthermore, Golgi studies[6] may have mistaken morphological variants for types, which could explain why many of their types cannot be identified in our optic lobe.

Contemporary light-microscopy anatomy leverages genetic lines, but still does not evade the limitations of incomplete and biased sampling. The story of Tm5 serves as a case in point. A breakthrough in colour vision started by genetically labelling neurons that express the histamine receptor Ort[7]. Researchers reasoned that Ort would be expressed by cells postsynaptic to the chromatic photoreceptors R7 and R8, which are histaminergic. Then, light-microscopy anatomy was used to make fine distinctions between three Tm5 types labelled in the transgenic line[7]. The present connectomic work has revealed six Tm5 types, a finding that was only foreshadowed by previous work on the same EM dataset[42]. The three new Tm5 types were presumably missed by previous studies because they receive little or no direct photoreceptor input (Fig. 7c), and do not express Ort. Nevertheless, they are similar to the old Tm5 types in morphology (Fig. 7a) and connectivity (Fig. 2c), and have been grouped in the hypothetical colour subsystem (Fig. 7c).

The Tm5 example demonstrates that connectomics can find fresh patches in well-trodden ground. More telling is that connectomics can guide us to entirely new landscapes, such as the 43 Sm types in an entirely new type family.

## Distinguishing cell types using connectivity

Features based on connectivity (Fig. 2a) enabled us to discriminate between cell types that stratify in very similar neuropil layers. Stratification constrains connectivity, because neurons cannot connect with each other unless they overlap in the same layers[1]. However, stratification does not completely determine connectivity, because neurons in the same layer may or may not connect with each other. Classical neuroanatomy, whether based on Golgi or genetic staining, relied on stratification because it could be seen with a light microscope. Now that we have electron microscopy data, we can rely on connectivity for cell typing, rather than settle for stratification as a proxy[2].

That being said, the present study used only connectivity at the final stage of cell typing, which was seeded by the morphological types identified during the first and second stages (Methods). It was possible to demonstrate self consistency of the final cell types using connectivity-based features only. We expect that it should be possible to eliminate all dependence on morphological typing, and base the approach on connectivity from start to finish. This challenge is left for future work.

## Spatial organization of connectivity

According to our wiring diagrams (Figs. 3–7 and Extended Data Figs. 4–6), whether two neurons are connected depends on their cell types. Connectivity also depends on the locations of the neurons in the retinotopic maps of the optic lobe. As a trivial example, it is impossible for cells with small arbours to be connected if they are at distant locations. Less trivial dependences of connectivity on location also exist. We expect them to be important for understanding vision, although they turned out to be unnecessary for classifying cell types. To facilitate spatial analyses of connectivity, the FlyWire Codex maps a number of cell types to locations in the hexagonal lattice of columns and ommatidia. In such analyses, it may be helpful to regard cell types and spatial locations as discrete and continuous latent variables[65]. A companion paper demonstrates how to predict visual function by characterizing how connectivity depends on both cell type and spatial location. The cell types of cluster 2 are predicted to exhibit orientation selectivity and related phenomena reminiscent of the primary visual cortex[60].

## Artificial intelligence

This paper began by recounting the story[66] of how wiring diagrams for visual cortex drawn in the 1960s inspired convolutional nets, which eventually sparked the deep learning revolution in artificial intelligence. Convolutional nets have now been applied to reconstruct the fly brain from electron microscopy images[24], making the current study possible. Coming full circle, the fly optic lobe turns out to be as literal an implementation of a convolutional net as one could ever expect from a biological system. The columns of the optic lobe form a hexagonal lattice, rather than the square lattice used in computer vision, but it is a highly regular lattice nonetheless, and the activities of the neurons in each cell type are analogous to a feature map in a convolutional net[67]. Although the connectional architecture of the optic lobe conforms closely to the definition of a convolutional net, the connections do not appear to be learned in the sense of artificial intelligence. No changes in VPN structure[68] and function[69], and only subtle changes in visual behaviour[70] have been detected after rearing flies in darkness, suggesting that visual experience may have little role in *Drosophila* visual development. However, mechanisms based on spontaneous activity in the pupal brain (before visual experience) might have a role[71].

## Implications for mammalian cell types

In the central brain of *Drosophila*, cell types usually consist of just a pair of mirror symmetric neurons[9,25] (Extended Data Fig. 1e), as is also the case for *C. elegans*[72]. By contrast, most optic lobe cell types are represented by many neurons (Fig. 1d and Extended Data Fig. 1d), a situation that is more reminiscent of mammalian brains[3,73]. Could our connectomic approach generalize to mammalian brain structures such as retina and cortex, which are laminated like the optic lobe?

Single-cell transcriptomics, often hailed as the solution to classifying cortical cell types[74], has also been applied to the *Drosophila* optic lobe. One study reported 172 transcriptomic cell types, a figure that includes VPNs as well as intrinsic neurons[62]. Our connectomic study has revealed the existence of a much larger set of types (700+ including boundary types). Encouragingly, many connectomic types can be conclusively matched with transcriptomic types[62]. Failures to match are interesting because they illustrate potential pitfalls of the transcriptomic approach. For example, all eight T4/T5 types look like a single transcriptomic type in adult flies[62], and are only transcriptionally distinct at earlier stages of development. This could be analogous to the fact that adult cortical neurons of the same transcriptomic type can have highly variable morphological properties[75,76]. It will be important to scale up the connectomic approach, and make it as definitive for the cortex as it is now for the fly visual system. A first attempt has already been made in visual cortex[19].

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

**The FlyWire Consortium**

**Krzysztof Kruk**[2,3]**, Ben Silverman**[1]**, Dustin Garner**[5]**, Jay Gager**[1]**, Kyle Patrick Willie**[1]**, Doug Bland**[1]**, Austin T. Burke**[1]**, James Hebditch**[1]**, Ryan Willie**[1]**, Celia David**[1,3]**, Gizem Sancer**[6]**, Jenna Joroff**[8]**, Anne Kristiansen**[3]**, Thomas Stocks**[3]**, Amalia Braun**[9]**, Szi-chieh Yu**[1,16]**, Emil Kind**[4]**, Marion Silies**[10]**, Jaime Skelton**[3]**, Travis R. Aiken**[3]**, Maria Ioannidou**[10]**, Marissa Sorek**[1,3]**, Matt Collie**[11]**, Gerit A. Linneweber**[12]**, Sebastian Molina-Obando**[10]**, Sven Dorkenwald**[1,7]**, Nelsie Panes**[13]**, Allien Mae Gogo**[13]**, Dorfam Rastgarmoghaddam**[10]**, Cathy Pilapil**[13]**, Rey Adrian Candilada**[13]**, Nikitas Serafetinidis**[3]**, Arie Matsliah**[1,16]**, Amy R. Sterling**[1,3]**, Mathias F. Wernet**[4]**, Sung Soo Kim**[5]**, Mala Murthy**[1]**, H. Sebastian Seung**[1,7]**, Wei-Chung Lee**[8]**, Alexander Borst**[9]**, Rachel I. Wilson**[8]**, Philipp Schlegel**[14,15] **& Gregory S. X. E. Jefferis**[14,15]

[8]Harvard Medical School, Boston, MA, USA. [9]Department Circuits–Computation–Models, Max Planck Institute for Biological Intelligence, Planegg, Germany. [10]Johannes-Gutenberg University Mainz, Mainz, Germany. [11]Harvard, Boston, MA, USA. [12]Freie Universität Berlin, Berlin, Germany. [13]SixEleven, Davao City, Philippines. [14]Neurobiology Division, MRC Laboratory of Molecular Biology, Cambridge, UK. [15]Drosophila Connectomics Group, Department of Zoology, University of Cambridge, Cambridge, UK.

## Methods

### Reconstruction accuracy and completeness

The overall quality of our *Drosophila* brain reconstruction has been evaluated elsewhere[24,31] (a summary of the current status is shown in Extended Data Table 3). Here we describe a few additional checks that are specific to the optic lobe. A small percentage of cells have eluded proofreading efforts. The worst cases are some types with visible 'bald spots' in the mid posterior side of the right optic lobe (Supplementary Data 2). In this region, we observed a narrowing and discontinuation of neuronal tracks. Many of these tracks appear to terminate within glial cells, suggesting a potential engulfment of neurons by glia. For most types, under-recovery is hardly visible (Supplementary Data 2).

For a quantitative estimate of under-recovery, we can rely on the 'modular' types[27], defined as cell types that are in one-to-one correspondence with columns. A previous reconstruction of seven medulla columns identified 20 modular types[28]. These largely correspond to the cell types that contain from 720 to 800 cells in our reconstruction (Fig. 1d). The top end (800) of this range is probably the true number of columns in this optic lobe. The lower end of this range is 720, suggesting that under-recovery is 10% at most, and typically less than that.

The inner photoreceptors R7 and R8 are about 650 cells each, and the outer photoreceptors R1–6 total about 3,400 in version 783 of the FlyWire connectome. These numbers are not inconsistent with modularity because photoreceptors are especially challenging to proofread in this dataset and under-recovery is higher than typical.

In the left optic lobe, we have proofread around 38,500 intrinsic neurons, as well as 3,700 VPNs, 250 VCNs, 150 heterolateral neurons and 5,000 photoreceptor cells. Tables comparing precise left/right counts by superclass as well as by type are available for download (see the 'Data availability' section).

Tm21 (also known as Tm6), Dm2, TmY5a, Tm27 and Mi15 are substantially less numerous than 800, so we agree with the seven column reconstruction[28] that they are not modular. On the other hand, some of our types (T2a, Tm3, T4c and T3) contain more than 800 proofread cells (Fig. 1d), which violates the definition of modularity. This partially agrees with the seven column reconstruction[28], which regarded T3 and T2a as modular, and T4 and Tm3 as not modular. T4 is an unusual case, as T4c is above 800 while the other T4 types are below 800. It should be noted that all of the above cell numbers could still creep upward with further proofreading.

A genuine analysis of modularity requires going beyond simple cell counts, and analysing locations to check the idea of one-to-one correspondence. Such an analysis is left for future work. Here we apply the term 'numerous' to those types containing 720 or more cells, as well as photoreceptor types, and do not commit to whether these types are truly modular.

The seven column reconstruction[28] provided a matrix of connections between their modular types. This shows good agreement with our data (Methods and Extended Data Fig. 9), providing a check on the accuracy of our reconstruction in the optic lobe. This validation complements the estimates of reconstruction accuracy in the central brain that are provided in the flagship paper[24].

The major limitation of our reconstruction in the optic lobe concerns the automatically detected synapses[77]. Although accuracy is high overall, outgoing photoreceptor synapses are markedly underdetected. This may be because dark cytoplasm (characteristic of photoreceptors) is not well represented in the example synapse images that were used to train the automated synapse detector. Example images of photoreceptor synapses have been included in the training set of an improved automated synapse detector, but the results were not ready in time for this publication, and will be made available in a future release. The classification of inner photoreceptors as yellow and pale is postponed until the future release. In the present paper, the connectivity from photoreceptors to other cell types in this paper is only qualitative and not quantitative. Furthermore, underdetection of photoreceptor synapses could affect the input fractions of other connections due to normalization.

Another cautionary note is that weaker connections in the type–type connectivity matrix (Extended Data Fig. 4) could be artifactual, due to false positives of automated synapse detection. There are some heuristics for guessing whether a connection is artifactual, short of manually inspecting the original EM images. For example, one might distrust weak connections between cells, that is, those with less than some threshold number of synapses. The choice of the threshold value depends on the context[9]. For example, the flagship paper[24] discarded connections with less than five synapses, a convention followed by the FlyWire Codex. The predicates of the present work apply a threshold of two synapses rather than five. The different thresholds were chosen because the central brain and optic lobes are very different contexts, as we now explain.

In the central brain, most cell types have cardinality 2 (cell and its mirror twin in the opposite hemisphere; Extended Data Fig. 1e). In the hemibrain, the cardinality is typically reduced to one. Therefore, whether there is a connection between cell type A and cell type B must be decided based on only two or three examples of the ordered pair (A, B) in all the connectomic data that is so far available. Given the small sample size, it makes sense to set the threshold to a relatively high value, if false positives are to be avoided.

On the other hand, in the optic lobe, there are often many examples of the ordered pair (A, B), because so many cell types have high cardinality. Therefore, if a connection is consistently found from type A to type B, one can have reasonable confidence even if the average number of synapses in the connection is not so high. That is why we set the threshold to a relatively low value in the optic lobe predicates. In particular, we have found that certain inhibitory types consistently make connections that involve relatively few synapses, and these connections seem real.

Another heuristic is to look for extreme asymmetry in the matrix. If the number of synapses from A to B is much larger than from B to A, the latter connection might be spurious. The reason is that the strong connection from A to B means the contact area between A and B is large, which means more opportunity for false-positive synapses from B to A. False-positive rates for synapses are estimated in the flagship paper[24].

Finally, it may be known from other studies that a connection does not exist. For example, T1 cells lack output synapses[26,78]. Therefore, in our analyses, we typically regarded the few outgoing T1 synapses in our data as false positives and discarded them.

### Morphological cell typing

Our connectomic cell approach to typing is initially seeded with some set of types, to define the feature vectors for cells (Fig. 2a), after which the types are refined by computational methods. For the initial seeding, we relied on the time-honoured approach of morphological cell typing, sometimes assisted by computational tools that analysed connectivity. It is worth noting that 'morphology' is a misnomer, because it refers to shape only, strictly speaking. Orientation and position are actually more fundamental properties because of their influence on stratification in neuropil layers. Thus, 'single-cell anatomy' would be more accurate than morphology, although the latter is the standard term.

**Stage 1: crowdsourced annotation of known types.** Annotations of optic lobe neurons were initially crowdsourced. The first annotators were volunteers from *Drosophila* laboratories. They were later joined by citizen scientists. At this stage, the annotation effort was mainly devoted to labelling cells of known types, especially the most numerous types.
***Drosophila* lab annotators.** E.K. and D.G. proofread and annotated medulla neurons that were upstream of the anterior visual pathway. These included many of the medulla and lamina neurons discussed in this study. The annotated neurons were primarily Dm2, Mi15, R7,

and R8, but also comprised various L, Dm, Mi, Tm, C and Sm cells. Previously known neuron types were identified primarily by morphology and partially by connectivity. Annotators additionally found all Mi1 neurons in both hemispheres to find every medulla column. These Mi1 neurons were used to create a map of medulla layers based on Mi1 stratification[6], which later aided citizen scientists to identify medulla cell types.

**Citizen scientists.** The top 100 players from Eyewire[79] had been invited to proofread in FlyWire[24]. After 3 months of proofreading in the right optic lobe, they were encouraged to also label neurons when they felt confident. Most citizen scientists did a mixture of annotation and proofreading. Sometimes they annotated cells after proofreading, and other times searched for cells of a particular type to proofread.

Citizen scientists were provided with a visual guide to optic lobe cells sourced from the literature[6,80]. FlyWire made available a 3D mesh overlay indicating the four main optic lobe neuropils. Visual identification was primarily based on single-cell anatomy. Initially, labelling of type families (that is, Dm, Tm, Mi and so on) was encouraged, especially for novices. Annotation of specific types (such as Dm3, Tm2) developed over time. The use of canonical names was further enforced by a software tool that enabled easy selection and submission of preformatted type names.

Additional community resources (discussion board/forum, blog, shared Google drive, chat, dedicated email and Twitch livestream) fostered an environment for sharing ideas and information between community members (citizen scientists, community managers and researchers). Community managers answered questions, provided resources such as the visual guide, shared updates, performed troubleshooting and general organization of community activity. Daily stats including number of annotations submitted per individual were shared on the discussion board/forum to provide project progress. Live interaction, demonstrations and communal problem solving occurred during weekly Twitch video livestreams led by a community manager. The environment created by these resources allowed citizen scientists to self-organize in several ways: community driven information sharing, programmatic tools and 'farms'.

**Community-driven information sharing.** Citizen scientists created a comprehensive guide with text and screenshots that expanded on the visual guide. They also found and studied any publicly available scientific literature or resources regarding the optic lobe. They shared findings at discuss.flywire.ai, which as of 10 October 2023 had over 2,500 posts. Community managers interacted with citizen scientists by sharing findings from the scientific literature, consulting *Drosophila* specialists on FlyWire and providing feedback.

**Programmatic Tools.** Programmatic tools were created to help with searching for cells of the same type. One important script traced partners-of-partners, that is, source cell→downstream partners→their upstream partners, or source cell→upstream partners→their downstream partners. This was based on the assumption that cells of the same type will probably synapse with the same target cells, which often turned out to be true. The tool could either look for partners-of-all-partners or partners-of-any-partners. The resulting lists of cells could be very long, and were filtered by excluding cells that had already been identified, or excluding segments with small sizes or low ID numbers (which had probably not yet been proofread). Another tool created from lobula plate tangential cells (for example, HS, VS, H1) aided definition of layers in the lobula plate. This facilitated identification of various cell types, especially T4 and T5.

**Cell farms.** Citizen scientists created farms in FlyWire or Neuroglancer with all the found cells of a given type visible. Farms showed visually where cells still remained to be found. If they found a bald spot, a popular method to find missing cells was to move the 2D plane in that place and add segments to the farm one after another in search of cells of the correct type. Farms also helped with identifying cells near to the edges of neuropils, where neurons are usually deformed. Having a view of all other cells of the same type made it possible to extrapolate to how a cell at the edge should look.

**Stage 2: centralized annotation and discovery of new types.** A team of image analysts at Princeton finished the annotation of the remaining cells in known types, and also discovered new types. Community annotations were initially compared with existing literature to confirm accuracy. Once validated, these cells were used to query various Codex search tools that returned previously unannotated cells exhibiting connectivity similar to that of the cell in the query. The hits from the search query were evaluated by morphology and stratification to confirm match with the target cell type. In some cases in which cell type distinctions were uncertain, predicted neurotransmitters[45] were used for additional guidance. This process enabled us to create a preliminary clustering of all previously known and new types.

## Connectomic cell typing

Eventually morphology became insufficient for further progress. Expert annotators, for example, struggled to classify Tm5 cells into the three known types, not knowing that there would turn out to be six Tm5 types. At this point, we were forced to transition to connectomic cell typing. In retrospect, this transition could have been made much earlier. As mentioned above, connectomic cell typing must be seeded with an initial set of types, but the seeding did not have to be as thorough as it ended up. We leave for future work the challenge of extending the connectomic approach so it can be used from start to finish.

**Stage 3: connectivity-based splitting and merging of types and auto-correction.** We used computational methods to split types that could not be properly split in stage 2. Some candidates for splitting (such as Tm5) were suggested by the image analysts. Some candidates were suspicious because they contained so many cells. Finally, some candidates were scrutinized because their type radii were large. We applied hierarchical clustering with average linkage, and accepted the splits if they did not violate the tiling principle as described in the 'Spatial coverage' section.

We also applied computational methods to merge types that had been improperly split in stage 2. Here the candidates were types with low spatial coverage of the visual field, or types that were suspiciously close in the dendrogram of cell types (Fig. 2c). Merge decisions were made by hierarchical clustering of cells from types that were candidates for merging, and validated if they improved spatial coverage.

Once we arrived at the final list of types, we estimated the 'centre' of each type using the element-wise trimmed mean. Then, for every cell, we computed the nearest type centre by Jaccard distance. For 98% of the cells, the nearest type centre coincided with the assigned type. We sampled some disagreements and reviewed them manually. In the majority of cases, the algorithm was correct, and the human annotators had made errors, usually of inattention. The remaining cases were mostly attributable to proofreading errors. There were also cases in which type centres had been contaminated by human-misassigned cells (see the 'Morphological variation' section), which in turn led to more misassignment by the algorithm. After addressing these issues, we applied the automatic corrections to all but 0.1% of cells, which were rejected using distance thresholds.

## Validation

On the basis of the auto-correction procedure, we estimate that our cell type assignments are between 98% and 99.9% accurate. For another measure of the quality of our cell typing, we computed the 'radius' of each type, defined as the average distance from its cells to its centre. Here we computed the centre by approximately minimizing the sum of Jaccard distances from each cell in the type to the centre (see the 'Computational concepts' section). A large type radius can be a sign that the type contains dissimilar cells, and should be split. For our final

types, the radii vary, but almost all lie below 0.6 (Extended Data Fig. 3a). Lat has an exceptionally high type radius, and deserves to be split (see the 'Cross-neuropil tangential and amacrine' section). The type radii are essentially the same, whether or not boundary types are included in the feature vector (data not shown).

**Discrimination with logical predicates.** Because the feature vector is rather high dimensional, it would be helpful to have simpler insights into what makes a type. One approach is to find a set of simple logical predicates based on connectivity that predict type membership with high accuracy. For a given cell, we define the attribute 'is connected to input type $t$' as meaning that the cell receives at least one connection from some cell of type $t$. Similarly, the attribute 'is connected to output type $t$' means that the cell makes at least one connection onto some cell of type $t$.

An optimal predicate is constructed for each type that consists of 2 tuples: input types and output types. Both tuples are limited to size 5 at most, and they are optimal with respect to the $F$-score of their prediction of the subject type, defined as follows:
- Recall of a predicate for type $T$ is the ratio of true positive predictions (cells matching the predicate) to the total number of true positives (cells of type $T$). It measures the predicate's ability to identify all positive instances of a given type.
- Precision is the ratio of true positive predictions (predictions that are indeed of type T) to the total number of positive predictions made by the logical predicate.
- $F$-score is the harmonic mean of precision and recall—a single metric that combines both precision and recall into one value.

On a high level, the process for computing the predicates is exhaustive—for each type, we look for all possible combinations of input type tuples and output type tuples and compute their precision, recall and $F$-score. A few optimization techniques are used to speed up this computation, by calculating minimum precision and recall thresholds from the current best candidate predicate and pruning many tuples early.

For example, the logical predicate 'is connected to input type Tm9 and output type Am1 and output type LPi15' predicts T5b cells with 99% precision and 99% recall. For all but three of the identified types, we found a logical predicate with 5 or fewer input/output attributes that predicts type membership with an average $F$-score of 0.93, weighted by the number of cells in type (Extended Data Fig. 4 and Supplementary Data 1). Some of the attributes in a predicate are the topmost connected partner types, but this is not necessarily the case. The attributes are distinctive partners, which are not always the most connected partners. The predicate for each type is shown on its card in Supplementary Data 2. For each family, the predicates for all types can be shown together in a single graph containing all of the relevant attributes (Supplementary Data 3).

We experimented with searching for predicates after randomly shuffling a small fraction of types (namely, swapping types for 5% of randomly picked pairs of neurons). We found that precision and recall of the best predicates dropped substantially, suggesting that we are not overfitting. This was expected because the predicates are short.

We also measured the drop in the quality of predicates if excluding boundary types (where the predicates are allowed to contain intrinsic types only). As is the case with the clustering metrics, the impact on predicates is marginal (weighted mean $F$-score drops from 0.93 to 0.92).

**Discrimination with two-dimensional projections.** Another approach to interpretability is to look at low-dimensional projections of the $2T$-dimensional feature vector. For each cell type, we select a small subset of dimensions that suffice to accurately discriminate that type

from other types (Extended Data Fig. 3c). Here we normalize the feature vector so that its elements represent the 'fraction of input synapses received from type $t$' or 'fraction of output synapses sent to type $t$'. In these normalized quantities, the denominator is the total number of all input or output synapses, not just the synapses with other neurons intrinsic to the optic lobe.

For example, we can visualize all cells in the Pm family in the two-dimensional space of C3 input fraction and TmY3 output fraction (Extended Data Fig. 3c). In this space, Pm04 cells are well-separated from other Pm cells, and can be discriminated with 100% accuracy by 'C3 input fraction greater than 0.01 and TmY3 output fraction greater than 0.01'. This conjunction of two features is a more accurate discriminator than either feature by itself.

More generally, a cell type discriminator is based on thresholding a set of input and output fractions, and taking the conjunction of the result. The search for a discriminator finds a set of dimensions, along with threshold values for the dimensions. To simplify the search, we require that the cell type be discriminated only from other types in the same neuropil family, rather than from all other types. Under these conditions, it almost always suffices to use just two dimensions of the normalized feature vector.

Discriminators for all types in all families containing more than one type are provided in Supplementary Data 4. Many although not all discriminations are highly accurate. Both intrinsic and boundary types are included as discriminative features.

## Computational concepts

**Connectivity: cell-to-cell, type-to-cell, cell-to-type and type-to-type.** Define a (weighted) cell-to-cell connectivity matrix $w_{ij}$, as the number of synapses from neuron $i$ to neuron $j$. The weighted out-degree and in-degree of neuron $i$ are:

$$d_i^+ = \sum_j w_{ij} \quad d_i^- = \sum_j w_{ji}$$

The sums are over all neurons in the brain. If neuron $i$ is a cell intrinsic to one optic lobe, the only nonvanishing terms in the sums are due to the intrinsic and boundary neurons for that optic lobe.

Let $A_{it}$ be the 0–1 matrix that assigns neuron $i$ to type $t$. The column and row sums of the assignment matrix satisfy

$$n_t = \sum_i A_{it} \quad 1 = \sum_t A_{it} \tag{2}$$

where $n_t$ is the number of cells assigned to type $t$.

The cell-to-type connectivity matrix $O_{it}$ is the number of output synapses from neuron $i$ to neurons of type $t$,

$$O_{it} = \sum_j w_{ij} A_{jt} \tag{3}$$

For fixed $i$, $O_{it}$ is known as the output feature vector of cell $i$. Similarly, the type-to-cell connectivity matrix $I_{tj}$ is the number of input synapses from neurons of type $t$ onto neuron $j$,

$$I_{tj} = \sum_j A_{it} w_{ij} \tag{4}$$

For fixed $j$, $I_{tj}$ is known as the input feature vector of cell $j$. The $i$th row and $i$th column of these matrices are concatenated to form the full feature vector for cell $i$ (Fig. 2a).

The input and output feature vectors can be normalized by degree to yield input and output fractions of cell $i$, $O_{it}/d_i^+$ and $I_{ti}/d_i^-$. Elements of these matrices are used for the discriminating 2D projections (Extended Data Fig. 3c).

The type-to-type connectivity matrix is the number of synapses from neurons of type $s$ to neurons of type $t$,

$$W_{st} = \sum_{ij} A_{is} w_{ij} A_{jt} \tag{5}$$

The weighted degree of type $t$ is the sum of the weighted degrees of the cells in type $t$,

$$D_t^+ = \sum_i A_{it} d_i^+ \quad D_t^- = \sum_i A_{it} d_i^- \tag{6}$$

The sums are over all neurons in the brain, similar to equation (1). Normalizing by degree yields the output fractions of type $s$, $W_{st}/D_s^+$, where $t$ runs from 1 to $T$. The input fractions of type $t$ are similarly given by $W_{st}/D_t^-$, where $s$ runs from 1 to $T$. Selected output and input fractions of types are shown in Supplementary Data 5.

Alternatively, the feature vectors can be based on connection number rather than synapse number, where a connection is defined as two or more synapses from one neuron to another. Then, weighted degree is replaced by unweighted degree in the above definitions. The threshold of two synapses is intended to suppress noise due to false positives in the automated synapse detection. Synapse number and connection number give similar results, and we use both in our analyses.

We found that it was sufficient for feature dimensions to include only intrinsic types ($T = 227$). Alternatively, feature dimensions can be defined as including both intrinsic and boundary types ($T > 700$), and this yields similar results (data not shown).

For the hierarchical clustering of cell types (Fig. 2c), the feature vector for each cell type is obtained by concatenating the vectors of input and output fractions for that cell type.

## Similarity and distance measures.
The weighted Jaccard similarity between feature vectors $\mathbf{x}$ and $\mathbf{y}$ is defined by

$$J(\mathbf{x}, \mathbf{y}) = \frac{\sum_t \min(x_t, y_t)}{\sum_{t'} \max(x_{t'}, y_{t'})} \tag{7}$$

and the weighted Jaccard distance $d(\mathbf{x},\mathbf{y})$ is defined as one minus the weighted Jaccard similarity. These quantities are bounded between zero and one since our feature vectors are nonnegative. In our cell typing efforts, we have found empirically that Jaccard similarity works better than cosine similarity when feature vectors are sparse.

## Type centres.
Given a set of feature vectors $\mathbf{x}^a$, the centre $\mathbf{c}$ can be defined as the vector minimizing

$$\sum_a d(\mathbf{x}^a, \mathbf{c}) \tag{8}$$

This cost function is convex, as $d$ is a metric satisfying the triangle inequality. Therefore, the cost function has a unique minimum. We used various approximate methods to minimize the cost function.

For auto-correction of type assignments, we used the element-wise trimmed mean. We found empirically that this gave good robustness to noise from false synapse detections. For the type radii, we used a coordinate descent approach, minimizing the cost function with respect to each $c_i$ in turn. The loop included every $i$ for which some $x_i$ was non-zero. This converged within a few iterations of the loop.

## Hierarchical clustering of cell types
The type-to-type connectivity matrix of equation (5) was the starting point for clustering the cell types. For each cell type, the corresponding row and column of the matrix were normalized to become input and output fractions, as described in the text following equation (6), and then concatenated (this is yet another way of computing type centres). Feature vectors included only dimensions corresponding to cell types intrinsic to the optic lobe. Then, average linkage hierarchical clustering was applied to yield a dendrogram (Fig. 2c). The dendrogram was thresholded to produce a flat clustering (Fig. 2c).

The precise memberships in the clusters warrant cautious interpretation, as the clusters are the outcome of just one clustering algorithm (average linkage), and differ if another clustering algorithm is used. Each cluster contains core groups of types that are highly similar to each other, that is, types that merge early during agglomeration (closer to the circumference of the dendrogram). These are more certain to have similar visual functions, and tend to be grouped together by any clustering algorithm. Types that are merged late (closer to the origin of the dendrogram) are less similar, and their cluster membership is more arbitrary. Some degree of arbitrariness is inevitable when one divides the visual system into separate subsystems, because subsystems interact with each other, and types that mediate such interactions are borderline cases.

Each cluster is generally a mixture of types from multiple neuropil families. Sceptics might regard such mixing as arising from the 'noisiness' in the clustering noted above at the largest distances. Indeed, the nearest types, those that merge in the dendrogram farther from the centre (Fig. 2c), tend to be from the same neuropil family. But plenty of dendrogram merges between types of different families happen at intermediate distances rather than the largest distances. Thus, some of the mixing of types from different neuropil families seems genuinely rooted in biology.

## Wiring diagrams
**Reduction.** To make the wiring diagrams readable, we display only the top type-to-type connections, which are defined as follows. For every cell type, the top input cell type and top output cell type are selected by ranking connected partners by the total number of synapses in the connection. If cell types are nearly tied, any runner up within 5% of the winner is also displayed. Figure 3 shows the top connections between all optic lobe intrinsic types. Figures 4–7 each focus on one or a few subsystems, but also include the top input/output connections they participate in with the rest of the network as well as top output connections to boundary types (for example, in Fig. 4, Dm2 is selected because it belongs to cluster 5, luminance channel, but then also other types outside of ON, OFF, and luminance channels are included because either Dm2 is their top input/output type or the other way around). Extended Data Figs. 5 and 6 show the top input and top output connections separately, for improved readability. For the top output connections we also include boundary types (VPNs).

**Colours and shapes.** Nodes, representing cell types, are coloured by clusters. Node size encodes the number of drawn connections, so that types that are top input/output of many other types look larger. Node shapes encode type numerosities (number of cells of that type), from most numerous (hexagon) to least (ellipse) (see the figure legends). The lines indicate connections between cell types. The line colour encodes the relationship (top input or top output) and the line width is proportional to the number of synapses connecting the respective types. The line arrowheads encode neurotransmitter predictions (excitatory/cholinergic or inhibitory/GABAergic/glutamatergic).

**Layout.** We used Cytoscape[81] to draw the wiring diagrams. Organic layout was used for Figs. 3 and 7c, and hierarchical layout was used for the others. The hierarchical layout tries to make arrows point downwards. After Cytoscape automatically generated a diagram, nodes were manually shifted by small displacements to minimize the number of obstructions.

## Intrinsic versus boundary
The optic lobes are divided into five regions (neuropils): lamina of the compound eye (LA); medulla (ME); accessory medulla (AME); lobula

(LO); lobula plate (LOP). All non-photoreceptor cells with synapses in these regions are split into two groups: optic lobe intrinsic neurons and boundary neurons.

Optic lobe intrinsic neurons are almost entirely contained in one of the optic lobes (left or right), more precisely, 95% or more of their synapses are assigned to the five optic lobe regions listed above.

Boundary neurons are those with at least 5% (and less than 95%) of synapses in the optic lobe regions, and are either visual projection, visual centrifugal or heterolateral neurons.

### Axon versus dendrite

In the main text (in the 'Class, family and type' section), we used the term 'axon'. An axon is defined as some portion of the neuron with a high ratio of presynapses to postsynapses. This ratio might be high in an absolute sense. Or the ratio in the axon might only be high relative to the ratio elsewhere in the neuron (the dendrite). In either case, the axon is typically not a pure output element, but has some postsynapses as well as presynapses. For many types it is obvious whether there is an axon, but for a few types we have made judgement calls. Even without examining synapses, the axon can often be recognized from the presence of varicosities, which are presynaptic boutons. The opposite of an axon is a dendrite, which has a high ratio of postsynapses to presynapses.

An amacrine cell is defined as one for which the axon–dendrite distinction does not hold, and presynapses and postsynapses are intermingled in roughly the same ratio throughout. The branches of an amacrine cell are often called dendrites, but the neutral term 'neurite' is perhaps better for avoiding confusion.

### Columnar neurons

Fischbach and Dittrich[6] defined 13 columnar families based on neuropils (Fig. 1a). Families consisting exclusively of 'numerous' (~800 cells) types include L (lamina to medulla), C (medulla to lamina), T1 (distal medulla to lamina), T2 (distal and proximal medulla to lobula), T3 (proximal medulla to lobula), T4 (proximal medulla to lobula plate) and T5 (lobula to lobula plate). We follow the convention of grouping the less numerous Lawf1 (distal medulla to lamina) and Lawf2 (proximal and distal medulla to lamina) types in the same family, despite the differences between their neuropils and connectivity. Although T1 shares the same neuropils with Lawf1, T1 lacks output synapses[26,78], so it is an outlier and deserves to be a separate family. Distal and proximal medulla are regarded as two separate neuropils[6].

**Mi.** Fischbach and Dittrich[6] defined Mi as projecting from distal to proximal medulla. Mi contains both numerous and less numerous types. We identified five (Mi1, 2, 4, 9, 10) of the dozen Mi types originally defined[6], and three (Mi13, 14, 15) types uncovered by EM reconstruction[27]. Mi1, Mi4, and Mi9 are consistent with the classical definition, but Mi13 projects from proximal to distal medulla. Other Mi types are less polarized, and the term "narrow-field amacrine" might be more accurate than "columnar". Nevertheless we will adhere to the convention that they are columnar. Narrow-field amacrine cells are also found in the Sm family, and exist in the mammalian retina[82].

**Tm transmedullary.** As classically defined[6], Tm cells project from the distal medulla to the lobula. Tm1 through Tm26 and Tm28 were defined[6], and Tm27/Tm27Y was reported later[83]. We were able to identify Tm1, 2, 3, 4, 7, 9, 16, 20, 21, 25 and 27. We split Tm5 into six types, and Tm8 into two types. We merged Tm6 and Tm21 into a single type Tm21. We prefer the latter name because the cells more closely match the Tm21 stratification as drawn by Fischbach and Dittrich[6]. Tm1a and Tm4a were defined as morphological variants[6], but we have found that they do not differ in connectivity and are not common, so we have merged them into Tm1 and Tm4, respectively. We merged Tm27Y into Tm27[83]. TmY5 was merged into TmY5a[6,84], the name that has appeared more often in the literature. These morphological distinctions originally arose

because the projection into the lobula plate, the differentiator between Tm and TmY, can vary across cells in a type. We added new types Tm31 to Tm37, which project from the serpentine medulla to the lobula. We moved Tm23 and Tm24 to the Li family. They were originally classified as Tm because their cell bodies are in the distal rind of the medulla, and they send a neurite along the columnar axis of the medulla to reach the lobula[6]. However, they do not form synapses in the medulla, so we regard them as Li neurons despite their soma locations. Overall, around half of the 26 types in the Tm family are new.

**TmY.** TmY cells project from the distal medulla to the lobula and lobula plate. The Y refers to the divergence of branches to the lobula and lobula plate. Previous definitions include TmY1 to TmY13[6]; TmY5a[6,84]; TmY14[27]; TmY15[29]; and TmY16, TmY18 and TmY20[30]. We identified TmY3, TmY4, TmY5a, TmY10, TmY11, TmY14, TmY15, TmY16 and TmY20. We divided TmY9 into two types, as discussed in a companion paper[60]. We added a new type, TmY31.

**Y.** Y cells project from the proximal medulla to the lobula and lobula plate. They are similar to TmY cells, but the latter traverse both the distal and proximal medulla[6]. Previous definitions were Y1 and Y3 to Y6[6]; and Y11 and Y12[10]. We have identified Y1, Y3, Y4, Y11 and Y12 in our reconstruction, and have not found any new Y types. Y1, Y11 and Y12 have the majority of their synapses in the lobula plate, and are assigned to the motion subsystem. Y3 and Y4 have few synapses in the lobula plate, and are assigned to the object subsystem (Fig. 2). Y3 is more numerous (~300 cells) than Y4, and is the only Y type that is predicted cholinergic.

**Tlp.** A Tlp neuron projects from the lobula plate to the lobula. Tlp1 to Tlp5 were defined first[6], and Tlp11 to Tlp14 were defined later on[10]. We have identified Tlp1, Tlp4, Tlp5 and Tlp14. We propose that the names Tlp11, Tlp12 and Tlp13 should be retired[10], as these types can now be unambiguously identified with Tlp5, Tlp1 and Tlp4, respectively.

### Interneurons

A local interneuron is defined as being completely confined to a single neuropil (Fig. 1b). Interneurons make up the majority of types, but a minority of cells (Fig. 1e). Lai is the only lamina interneuron. Dm and Pm interneurons[6] stratify in the distal or proximal medulla, respectively. We have more than doubled the number of Pm types, and slightly increased the number of Dm types. We introduce the Sm family, which is almost completely new and contains more types than any other family (Fig. 1f). Li and LPi interneurons stratify in the lobula or lobula plate, respectively. Interneurons are usually amacrine and presumed inhibitory (GABA or glutamate), but some are tangential or cholinergic. Interneurons are often wide field but some are narrow field.

**Dm.** Dm1 to Dm8[6]; Dm9 and 10[27]; and Dm11 to Dm20[85] were previously defined. We do not observe Dm5 and Dm7, consistent with a previous study[85]. Most types are predicted to secrete glutamate or GABA, but there are also a few cholinergic types (Supplementary Data 1). To Dm3p and Dm3q[61,62,85], we added a third type, Dm3v (Supplementary Data 2). We split Dm8 into Dm8a and Dm8b (see the 'Correspondences with molecular–morphological types' section).

**DmDRA.** The DRA differs from the rest of the retina in its organization of inner photoreceptors. Photoreceptors in non-DRA and DRA differ in their axonal target layers and output cell types[54,86]. Specifically, DRA-R7 connects with DmDRA1, whereas DRA-R8 connects to DmDRA2[54,87]. These distinctive connectivity patterns result in DmDRA1 and DmDRA2 types exhibiting an arched coverage primarily in the M6 layer of the dorsal medulla (Fig. 9b). R7-DRA and R8-DRA are incompletely annotated at present, and this will be rectified in a future release. DmDRA1

receives R7 input, but sits squarely in M7. This could be regarded as an Sm type, but we have chosen not to change the name for historical reasons.

**Pm.** Pm1, 1a and 2[6] were each split into two types. Pm3 and 4 remain as previously defined[85]. We additionally identified six new Pm types, for a total of 14 Pm types, numbered Pm01 to Pm14 in order of increasing average cell volume. The new names can be distinguished from the old ones by the presence of leading zeros. All are predicted GABAergic. Pm1 was split into Pm06 and Pm04, Pm1a into Pm02 and Pm01, and Pm2 into Pm03 and Pm08.

**Sm.** Dm and Pm interneurons are defined[6] to stratify on the distal or proximal side, respectively, of the serpentine layer (M7) of the medulla. Many interneuron types turn out to have significant stratification in the serpentine layer, and these borderline cases constitute a large new Sm family of interneurons, almost all new. They have been named Sm01 to Sm43, mostly in order of increasing average cell volume. The Sm family includes types recently named medulla tangential intrinsic[42]. We avoid using this term indiscriminately because some Sm types are tangential while others are amacrine. Some Sm types spill over from M7 into the distal or proximal medulla, and a few reach from M7 to more distant medulla layers.

Sm stratification in M7 has functional implications. First, Sm types are positioned to communicate with the medulla tangential (Mt) cells and other boundary types that are important conduits of information in and out of the optic lobe (Supplementary Data 5). Second, Sm types are positioned to communicate with the inner photoreceptor terminals, which are in M6 or at the edge of M7. Consequently many Sm types are involved in the processing of chromatic stimuli, and end up being assigned to the colour subsystem.

The Sm family more than doubles the number of medulla interneuron types, relative to the old scheme with only Pm and Dm. The Sm family might be related to the M6-LN class of neuron previously defined[88]. The correspondence is unclear because M6-LN neurons are defined to stratify in M6, while Sm mainly stratifies in M7. But some Sm types stratify at the border between M6 and M7, and therefore could be compatible with the M6-LN description.

**Li.** After two lobula intrinsic types (Li1 and Li2) were initially defined[6], 12 more (Li11 to 20 and mALC1 and mALC2) were identified by the hemibrain reconstruction[9]. Of these, we have confirmed Li2, Li12, Li16, mALC1 and mALC2. We identified 21 additional Li types, but have not been able to make conclusive correspondences with previously identified types. As mentioned earlier, we transfer Tm23 and Tm24[6] from the Tm to the Li family. This amounts to a total of 33 Li types, which have been named Li01 to Li33 in order of increasing average cell volume.

Collisions with Li1 and Li2[6] are avoided by the presence of leading zeros in our new names. The hemibrain names Li11 to Li20 and mALC1 and mALC2[9] have been used by few or no publications, so there is little cost associated with name changes. In any case, we were only able to establish conclusive correspondences for a minority of the hemibrain Li11 to Li20 types, which are detailed in Supplementary Data 1. Hemibrain Li12 is now Li27 (jigsaw pair), and hemibrain Li16 is now Li28 (pair of full-field cells). Hemibrain Li11 was split into Li25 and Li19 (see the 'Morphological variation' section). Hemibrain Li18 was split into three types: (1) Li08 covers the whole visual field. (2) Li04 covers a dorsal region except for the dorsal rim. It is tangentially polarized, with the axon more dorsal than the dendrites. Both axon and dendrite point in the posterior direction, perpendicular to the direction of polarization. The dendrites are more thickly stratified than the axon. (3) Li07 has ventral coverage only. The axons are in one layer, and extend over a larger area than the dendrites, which hook around into another layer and are mostly near the ventral rim.

We considered merging Li04 and Li07, but their connectivity is quite different. Furthermore, in a hierarchical agglomerative clustering, Li07 would merge with Li08 before Li04.

**LPi.** LPi names were originally based on stratification in layers 1 to 4 of the lobula plate, including LPi1-2 and 2-1[10]; LPi3-4 and 4-3[8]; and LPi2b and LPi34-12[10] (we are not counting fragments for which correspondences are not easy to establish). We have added nine new types, for a total of 15 LPi types.

Now that LPi types have multiplied, stratification is no longer sufficient for naming. The naming system could be salvaged by adding letters to distinguish between cells of different sizes. For example, LPi15 and LPi05 could be called LPi2-1f and LPi2-1s, where 'f' means full-field and 's' means small. For simplicity and brevity, we instead chose the names LPi01 to LPi15, in order of increasing average cell volume. Correspondences with old stratification-based names are detailed in Codex.

### Cross-neuropil tangential and amacrine

Most types that span multiple neuropils are columnar. One tangential type that spans multiple neuropils inside the optic lobe was previously described: Lat has a tangential axon that projects from the medulla to the lamina[6]. There is some heterogeneity in the Lat population, as reflected in the large type radius (Extended Data Fig. 3a). We have decided to leave splitting for future work, as Lat has many dense core vesicles that are presently unannotated.

Here we introduce two new families of cross-neuropil types that are tangential (MLt1-8 and LMt1-4), and one that is amacrine (LMa1-5). Along with two new tangential families (PDt, LLPt) that contain only single types, and the known CT1 and Am1 types, that is a total of 21 cross-neuropil types that are non-columnar (Fig. 1c). Each of the new types (except PDt with 6 cells) contains between 10 and 100 cells.

The tangential types connect neuropils within one optic lobe and do not leave the optic lobe. Our usage of the term 'tangential' focuses on axonal orientation only. It should not be misunderstood to imply a wide-field neuron that projects out of the optic lobe, which is the case for the well-known lobula plate tangential cells or lobula tangential cells. The term 'tangential' presupposes that we can identify an axonal arbour for the cell (see the 'Axon versus dendrite' section).

**PDt.** We found one tangential type that projects from proximal to distal medulla (Supplementary Data 2).

**MLt.** ML1 was previously identified[42] as a tangential neuron projecting from the medulla to lobula. We will refer to this type as MLt1, and have discovered more types of the same family, MLt2 to MLt8. Mlt1 and Mlt2 dendrites span both distal and proximal medulla, and Mlt3 dendrites are in the distal medulla, so MLt1 to MLt3 receive L input (Supplementary Data 2 and 5). Mlt4 dendrites are in the proximal medulla (Supplementary Data 2). Mlt5 to Mlt8 have substantial arbour overlap with the serpentine layer M7 (Supplementary Data 2), and are therefore connected with many Sm types to be discussed later on (Supplementary Data 5). Interaction between MLt types is fairly weak, with the exception of MLt7 to MLt5 (Supplementary Data 5). MLt7 and MLt8 are restricted to the dorsal and dorsal rim areas.

**LMt.** We identified four tangential types (LMt1 to LMt4) that project from the lobula to medulla. Their axonal arbours are all in the proximal medulla (Supplementary Data 2), thinly stratified near layer M7, so they have many Pm targets (Supplementary Data 5). Only LMt4 exhibits partial coverage.

**LLPt.** We discovered one tangential type that projected from the lobula to lobula plate, and called it LLPt. This is just a single type, rather than a family.

**LMa.** We discovered four amacrine types that extend over the lobula and medulla. LMa1 to LMa4 are coupled with T2, T2a and T3, and LMa4 and LMa3 synapse onto T4 and T5 (Supplementary Data 5). The LMa family could be said to include CT1, a known amacrine cell that also extends over both the lobula and medulla. However, the new LMa types consist of smaller cells that each cover a fraction of the visual field, whereas CT1 is a wide-field cell.

**MLLPa.** Am1 was defined[10] as a wide-field amacrine cell that extends over the medulla, lobula and lobula plate. We found no other amacrine types like Am1 with such an extended reach.

### Correspondences with molecular–morphological types

**Tm5.** Tm5a, Tm5b and Tm5c were originally defined by single-cell anatomy and Ort expression[7,50]. Tm5a is cholinergic, the majority of the cells extend one dendrite from M6 to M3, and often has a 'hook' at the end of its lobula axon. Tm5b is cholinergic, and most (~80%) cells extend several dendrites from M6 to M3. Tm5c is glutamatergic and extends its dendrites up to the surface of the distal medulla. Three of our types are consistent with these morphological descriptions (Fig. 7a), and receive direct input from inner photoreceptors R7 or R8.

**Dm8.** Molecular studies previously divided Dm8 cells into two types (yDm8 and pDm8), depending on whether or not they express DIPγ[51,53]. Physiological studies demonstrated that yDm8 and pDm8 have differing spectral sensitivities[89]. The main dendrites of yDm8 and pDm8 were found to connect with R7 in yellow and pale columns, respectively. On the basis of its strong coupling with Tm5a, our Dm8a probably has some correspondence with yDm8, which is likewise selectively connected with Tm5a[51,53]. It is not yet clear whether there is a true one-to-one correspondence of yDm8 and pDm8 with Dm8a and Dm8b. It is the case that Dm8a and Dm8b strongly prefer to synapse onto Tm5a and Tm5b, respectively. However, Tm5a and Tm5b are not in one-to-one correspondence with yellow and pale columns. Rather, the main dendritic branch of Tm5a is specific to yellow columns, while the main dendritic branches of Tm5b are found in both yellow and pale columns[50]. Furthermore, Dm8a and Dm8b cells are roughly equal in number, while the yDm8:pDm8 ratio is expected to be substantially greater than one[51,53], like the ratio of yellow to pale columns. Thus, the correspondence of Dm8a and Dm8b with yDm8 and pDm8 is still speculative. The yellow/pale issue should be revisited in the future when accurate photoreceptor synapses become available (see the 'Reconstruction accuracy and completeness' section).

**Additional validation.** HHMI Janelia has released a preprint detailing cell types in the right optic lobe of an adult male *Drosophila* brain[90]. The list of intrinsic cell types is almost identical to ours, apart from naming differences in new types. Since our original submission, we have completed typing of the left optic lobe of our female fly brain reconstruction, and the results match the right optic lobe analysed in the present paper. These replications in another hemisphere of the same brain and in the brain of another individual fly provide additional validation of our findings.

### Reporting summary

Further information on research design is available in the Nature Portfolio Reporting Summary linked to this article.

### Data availability

The present work is based on version 783 of the FlyWire connectome, which incorporates proofreading up to 30 September 2023 (stats are shown in Extended Data Table 3). A static snapshot of the data used in this work is available in a dedicated repository at GitHub (https://github.com/murthylab/visual-system-parts-list). This repository contains the proofread cell IDs, their types, connectivity (broken up by regions), as well as aggregate information such as type summary table, type connectivity table and raw data used to make the figures, including CSV files for each of the wiring diagrams. Most up to date information can be browsed, searched and downloaded at the FlyWire Codex (https://codex.flywire.ai). Codex will also provide access to future releases of the FlyWire connectome, incorporating updated proofreading and annotations. Pre-release annotations can be downloaded directly from the Codex download portal (https://codex.flywire. ai/api/download). Pre-release proofread cells are available through CAVEclient[24,91].

### Code availability

Code for making the figures along with additional data analysis tools are also included/linked in GitHub repositories (https://github.com/ murthylab/visual-system-parts-list and https://github.com/hsseung/ OpticLobe.jl). Most up to date information can be browsed, searched and downloaded at the FlyWire Codex (https://codex.flywire.ai).

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

**Acknowledgements** We thank all those who proofread optic lobe intrinsic neurons[24]; A. Nern and M. Reiser for educating and advising the FlyWire community members who engaged in annotation of visual neurons, and for sharing correspondences with optic lobe cell types identified at HHMI Janelia; the members of the FAFB tracing community for supportive and open sharing of methods and data, especially the FAFB optic lobe working group; Y. Kurmangaliyev for his comments on the manuscript; R. Behnia and K. Zinn for advice about colour; R. Behnia and M. Silies for feedback about visual motion detection pathways; J. Wiggins, G. McGrath and D. Barlieb for computer system administration; M. Husseini for project administration; and J. Maitin-Shepard for Neuroglancer. M.M. and H.S.S. acknowledge support from the National Institutes of Health (NIH) BRAIN Initiative RF1 MH117815, RF1 MH129268 and U24 NS126935, from the Princeton Neuroscience Institute, as well as assistance from Google. D.G. and S.S.K. were supported by the National Eye Institute of the NIH (DP2EY032737), Searle Scholars Program, Sloan Research Fellowship and Klingenstein-Simons Fellowship in Neuroscience. E.K., G.S. and M.F.W. were supported by Deutsche Forschungsgemeinschaft (DFG) grant WE 5761/4-1, SPP 2205, FOR 5289 and AFOSR grant FA9550-19-1-7005.

**Author contributions** D.G., E.K. and G.S. annotated cells under the supervision of S.S.K. and M.F.W. M.S. and A.R.S. recruited, trained and managed citizen scientists with help from E.K. K.K. annotated cells and created computational cell typing tools for use by the community. S.-c.Y. trained and managed D.B., A.T.B., J.G., J.H., B.S., K.P.W. and R.W. to annotate the remaining known cell types and identify and annotate new types. A.M. and H.S.S. created semiautomated cell typing tools. A.M. and H.S.S. carried out the final automated stage of typing. A.M. verified

types with predicates. H.S.S. verified types with 2D projections. S.-c.Y., H.S.S. and M.M. devised type family names. A.M. and H.S.S. defined and characterized subsystems. A.M. drew wiring diagrams. S.S.K., M.F.W., M.M. and H.S.S. identified implications for visual function. A.T.B., J.G., J.H., B.S., K.P.W., R.W., S.-c.Y., A.M. and H.S.S. created figures. K.K., M.S., A.R.S., A.M., S.-c.Y., S.S.K., M.M. and H.S.S. wrote the text. M.M. and H.S.S. supervised the project. Members listed in the FlyWire consortium made at least ten annotations in the optic lobe.

**Competing interests** H.S.S. declares financial interests in Zetta AI. The other authors declare no competing interests.

**Additional information**
**Correspondence and requests for materials** should be addressed to Mala Murthy or H. Sebastian Seung.

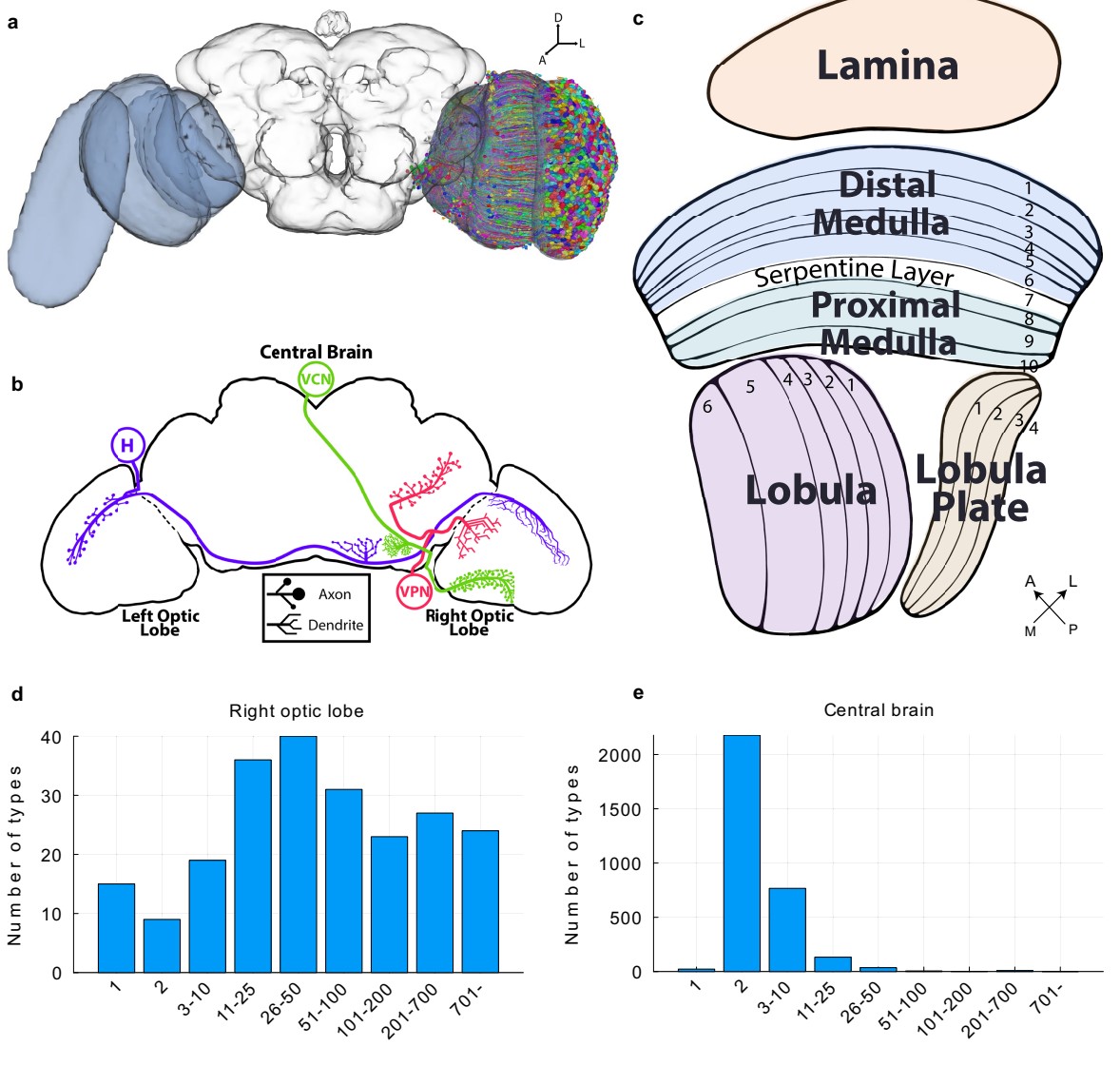

**Extended Data Fig. 1 | Cell counts of types in optic lobe versus central brain. a**, *Drosophila* central brain and flanking optic lobes. Neurons intrinsic to the optic lobes (colours) are the subject of this study (A: Anterior. L: Lateral. D: Dorsal). **b**, Boundary cells straddle the optic lobe and central brain (H: heterolateral, VCN: visual centrifugal neuron: VPN: visual projection neuron). **c**, Optic lobe main neuropils (brain regions) and their layering (A: Anterior. L: Lateral. M: Medial. P: Posterior). **d**, Distribution of number of **optic lobe** types by bucketed unilateral cardinality. Each bar represents types whose cardinality (number of cells) is within the specified range. Most types contain 10+ cells, and a significant portion of types contain hundreds of cells. **e**, Distribution of the number of **central brain** types by bucketed bilateral cardinality. In contrast to the optic lobe, here most types have cardinality 2 (cell and its mirror twin in the opposite hemisphere).

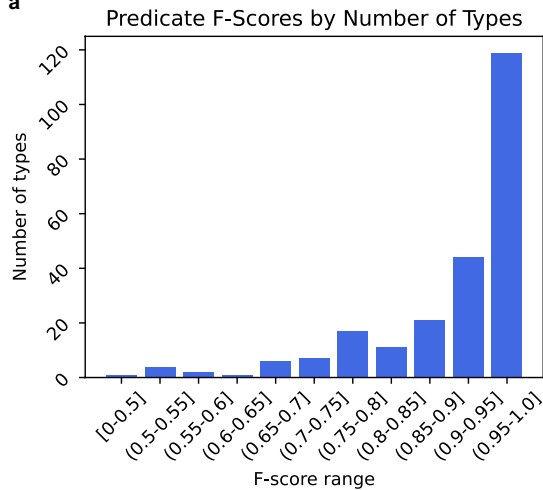

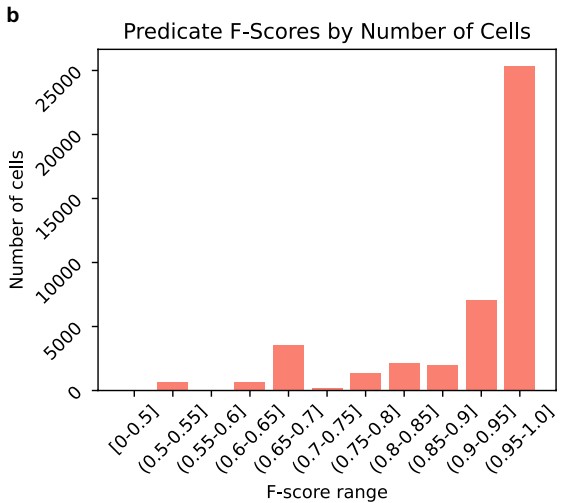

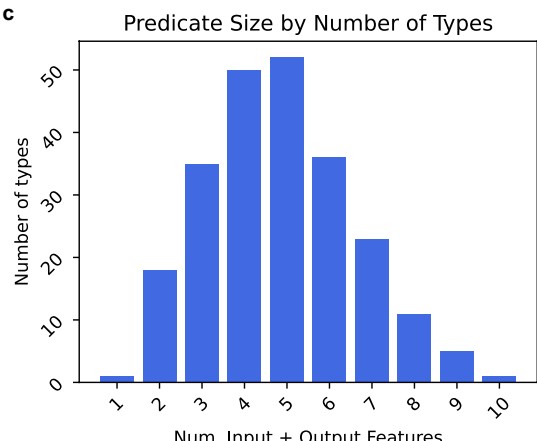

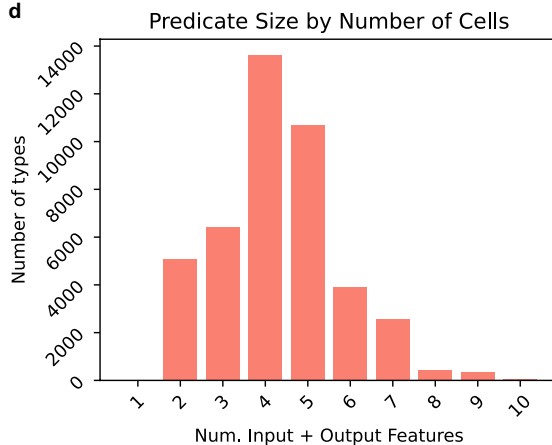

**Extended Data Fig. 2 | Logical connectivity predicate statistics. a**, Number of types by predicate F-score range. **b**, Number of cells by their types' predicate F-score range. **c**, Number of types by predicate size, that is the sum of the number of input features and output features participating in the binary conjunction. **d**, Number of cells by their types' predicate size.

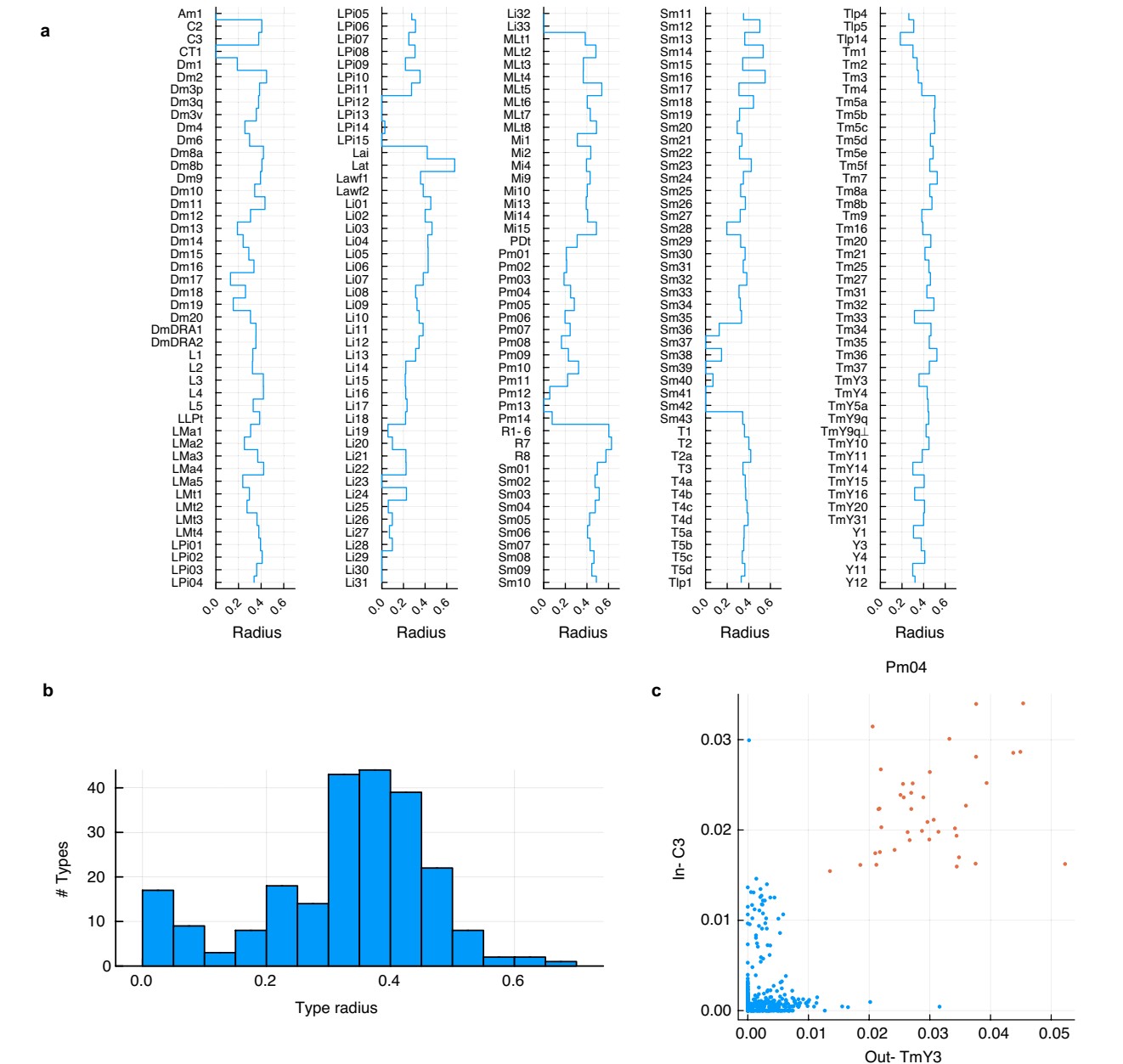

**Extended Data Fig. 3 | Discrimination in high and low dimensions. a**, Radii of types in high-dimensional feature space. **b**, Histogram of type radii in high-dimensional feature space. **c**, Example 2D discriminator for Pm04 cells (red) versus other Pm types (blue). On the X and Y axis are the fraction of their inputs/outputs in C3/TmY3 respectively.

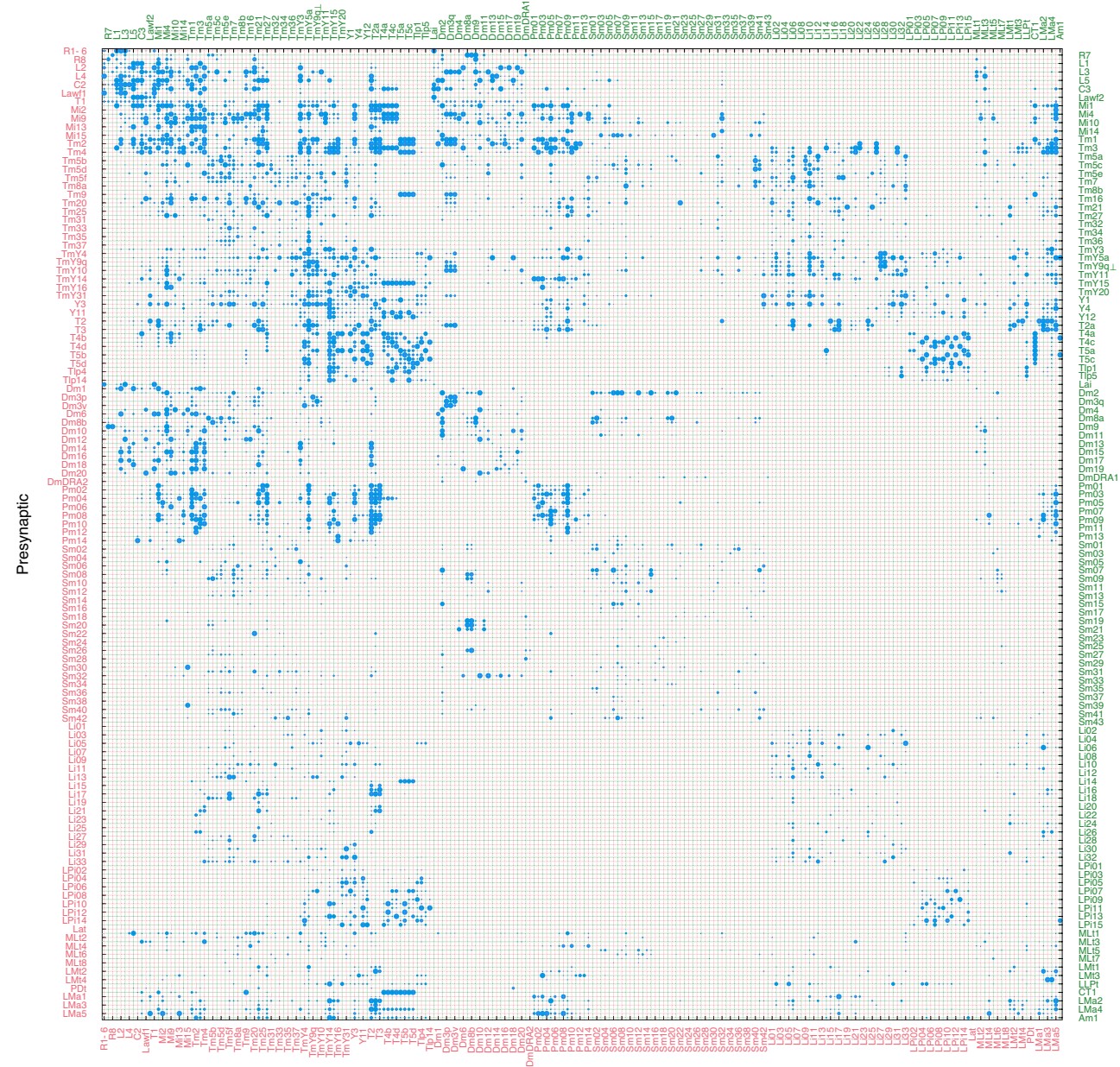

**Extended Data Fig. 4 | Type-to-type connectivity as a matrix.** The number of synapses from one cell type to another is indicated by the area of the corresponding dot. Dot area saturates above 3600 synapses, to make weaker connections visible. For legibility, the type names alternate between left and right edges, and bottom and top edges, and are colour coded to match the lines that are guides to the eye.

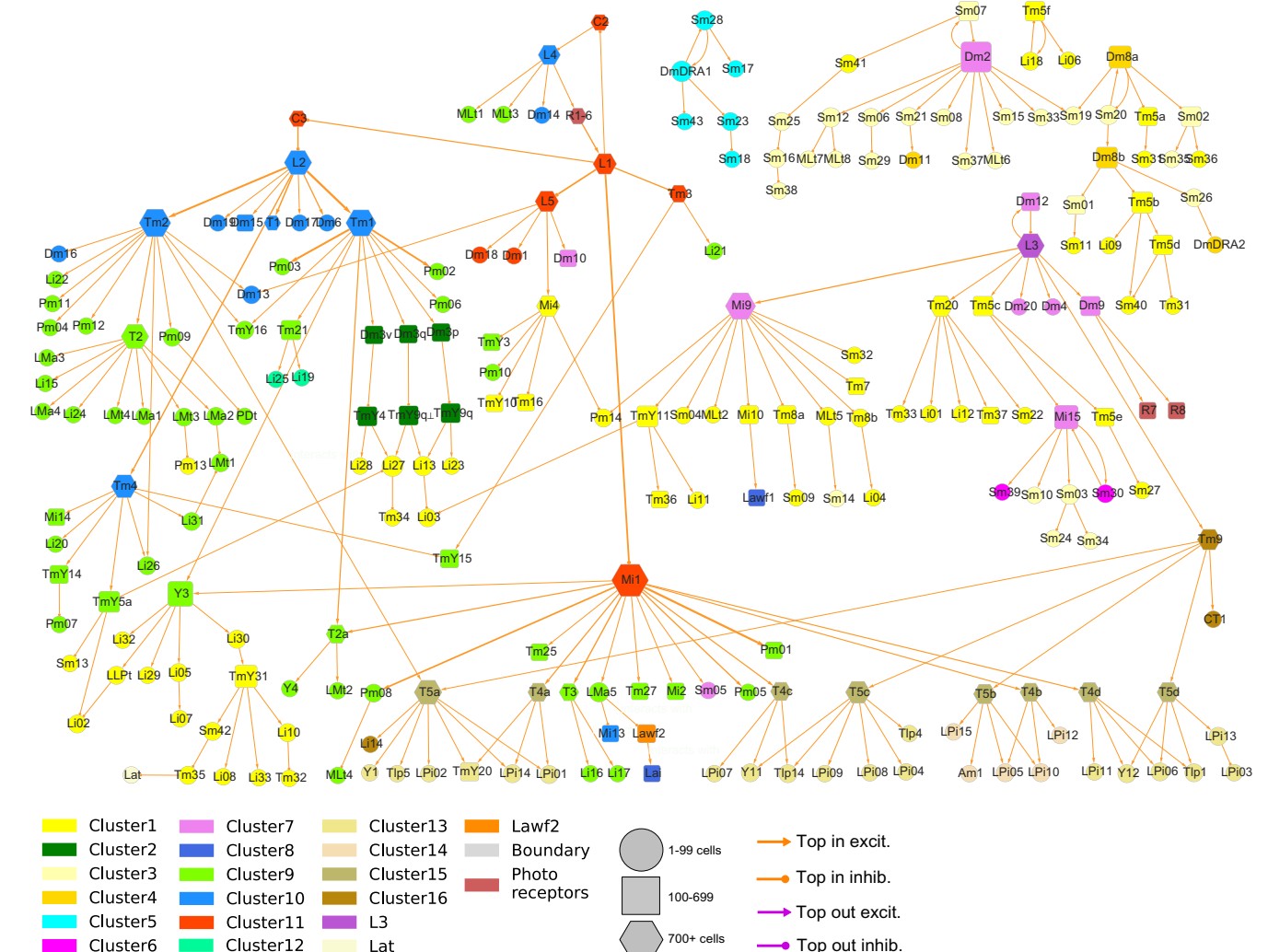

**Extended Data Fig. 5 | Wiring diagram of cell types (top input connections).**
Wiring diagram depicting top inputs for all cell types intrinsic to the optic
lobe, as well as photoreceptors. Node size encodes the number of drawn
connections, highlighting "hub" inputs. Node colour indicates membership in
the subsystems defined in the text. See legend and additional explanation in
Fig. 3 and Methods.

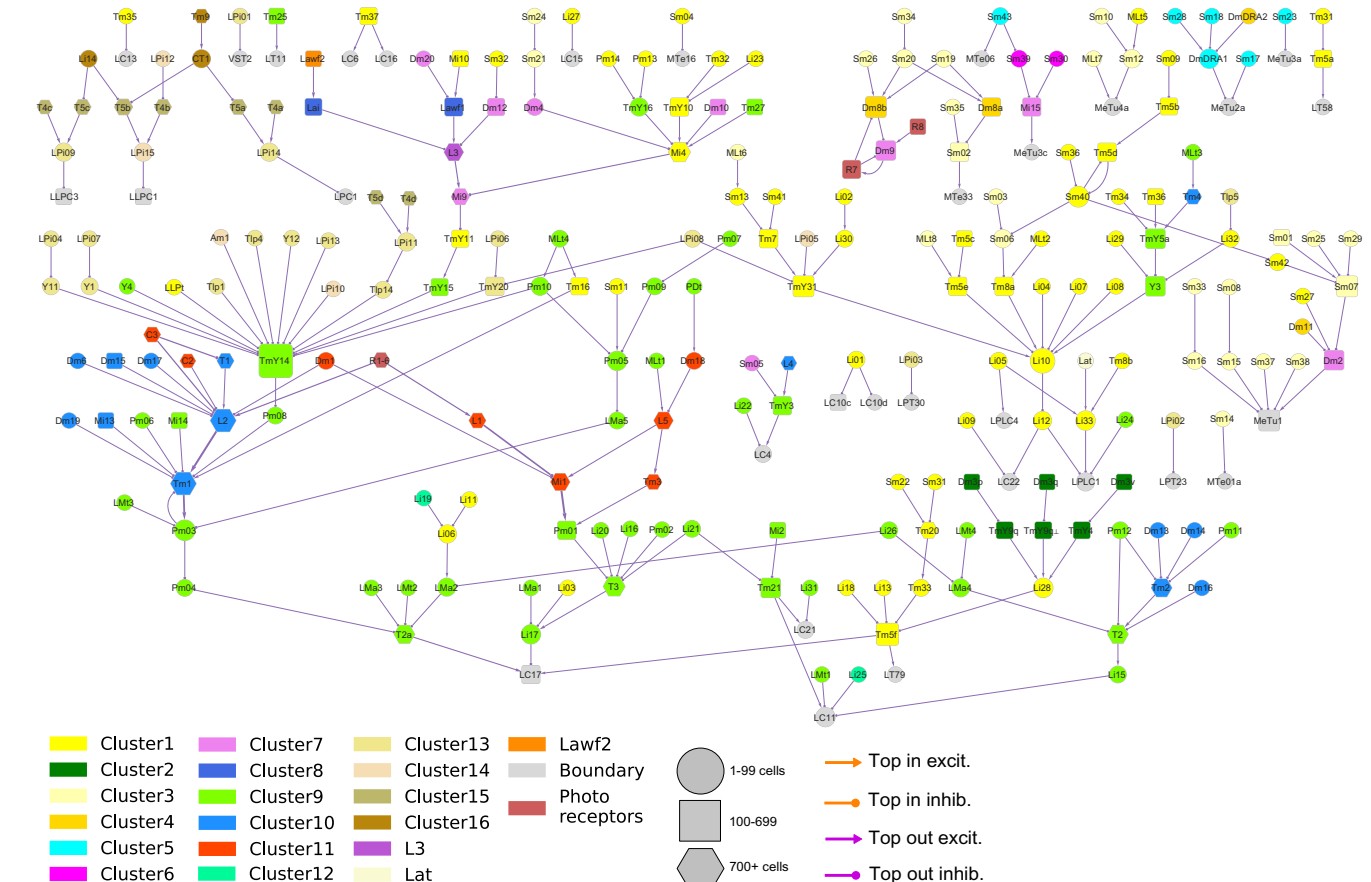

**Extended Data Fig. 6 | Wiring diagram of cell types (top output connections).**
Wiring diagram depicting top outputs for all types intrinsic to the optic lobe. Node size encodes the number of drawn connections, highlighting "hub" outputs. Node colour indicates membership in the subsystems defined in the text. See legend and additional explanation in Fig. 3 and Methods.

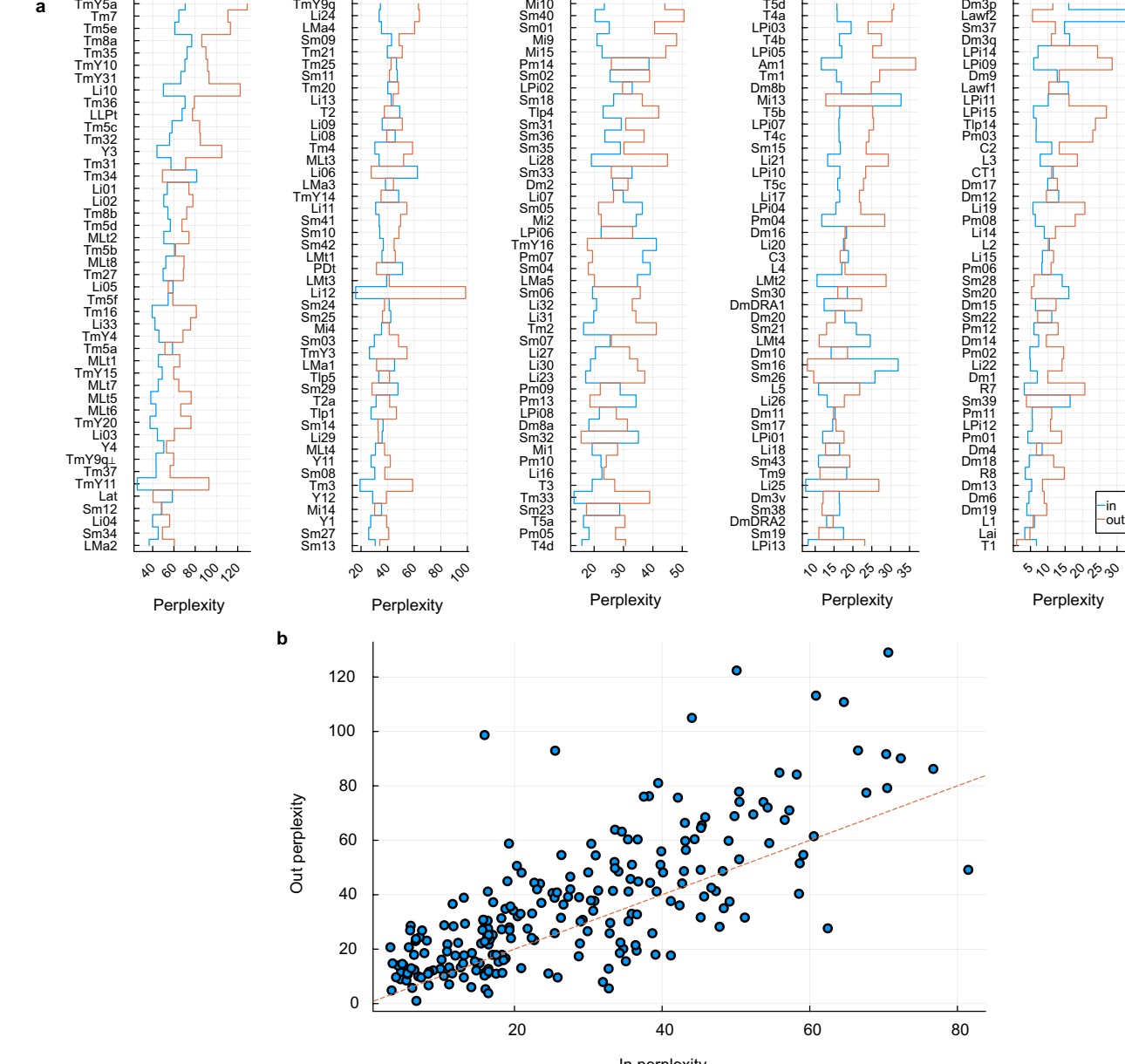

**Extended Data Fig. 7 | Input and output perplexity. a**, Input (blue) and output (red) perplexities. Types are ordered by the product of input and output perplexities. **b**, Output and input perplexity are correlated. Out-perplexity tends to exceed in-perplexity (more points above red line drawn to indicate equality of out and in).

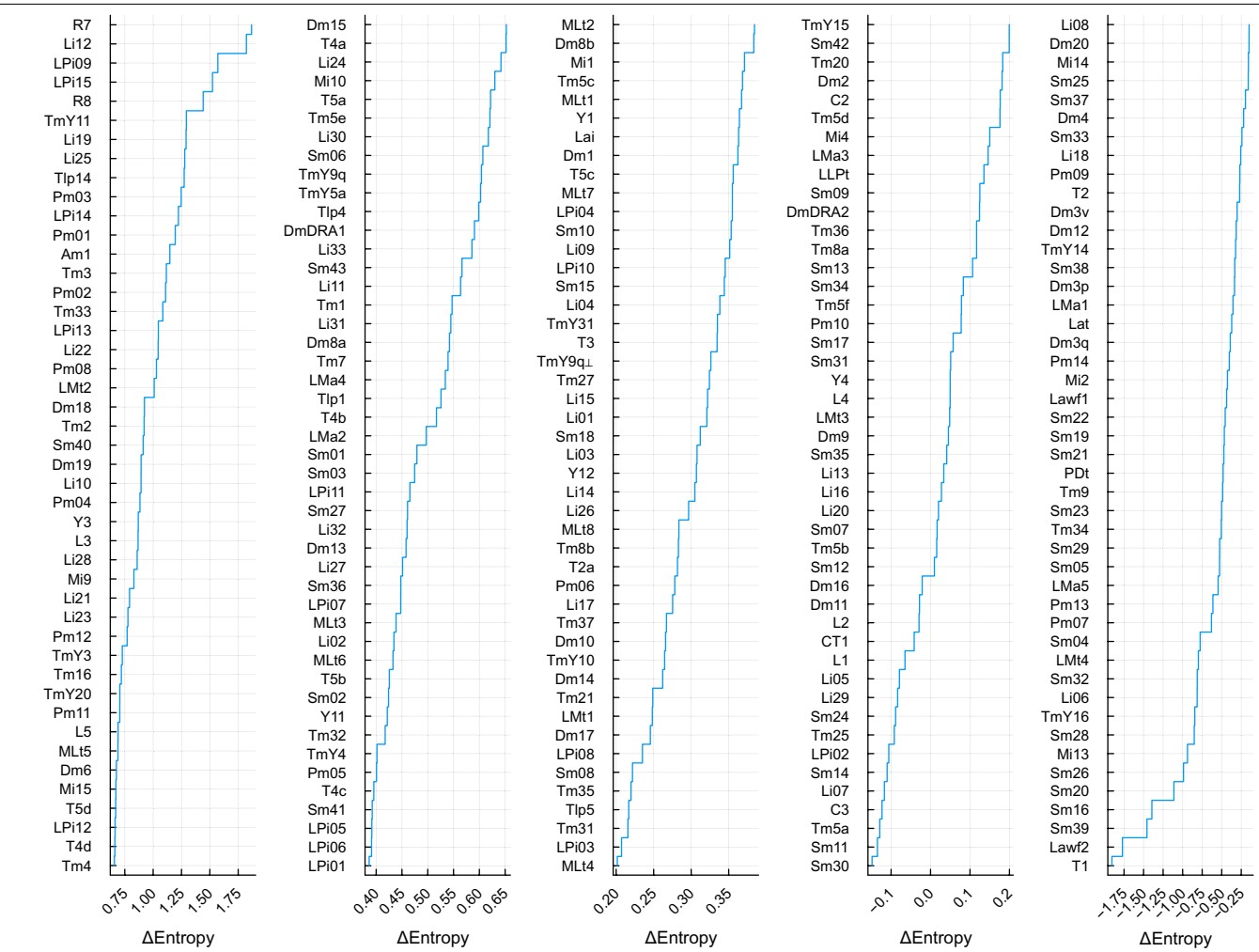

**Extended Data Fig. 8 | Difference between output and input entropies.** The difference between output and input entropies (units of nats) quantifies the degree of divergence or convergence. This difference is equivalent to the logarithm of the ratio of out- and in-perplexities. The connectivity of the top types (top left) is more divergent, as the output entropy is greater than the input entropy. The connectivity of the bottom types (bottom right) is more convergent, as the input entropy is greater than the output entropy.

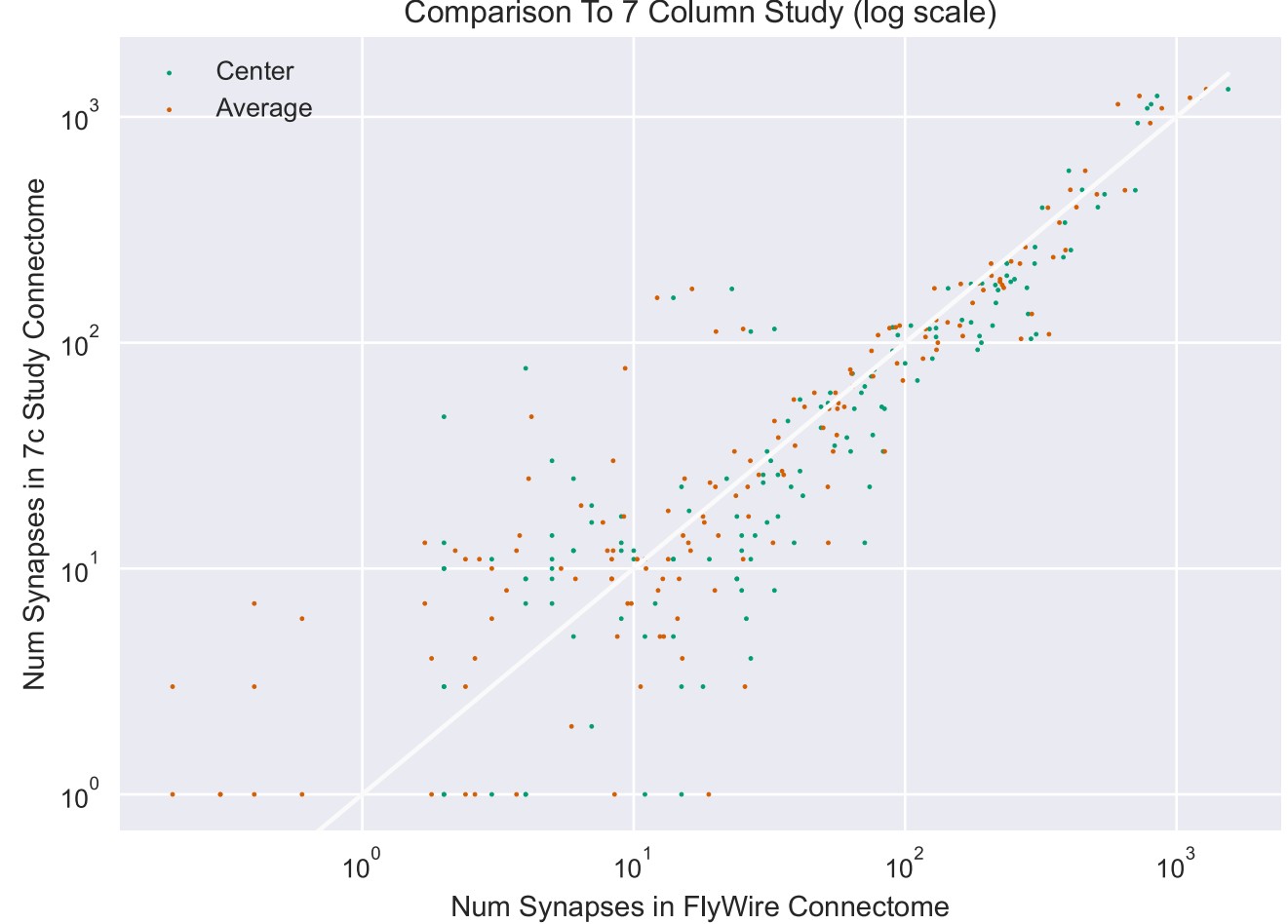

**Extended Data Fig. 9 | Comparison with seven-column reconstruction.** We compared the synapse counts between type pairs to the corresponding synapse counts in the seven-column reconstruction[28]. The types included in the reconstruction are: C2, C3, L1, L2, L3, L4, L5, Mi1, Mi4, Mi9, R7, R8, T1, T2, T2a, T3, Tm1, Tm2, Tm20 and Tm9. For this comparison we used the centre column and its surrounding 6 columns from our dataset (green dots) as well as the average of 100 columns and their surrounding ones (red dots). Each point represents an ordered pair of types, and the number of synapses between them in the FlyWire connectome (X) and the seven-column reconstruction (Y). Correlation coefficients are 0.952 for the centre + 6 columns and 0.954 for the average.

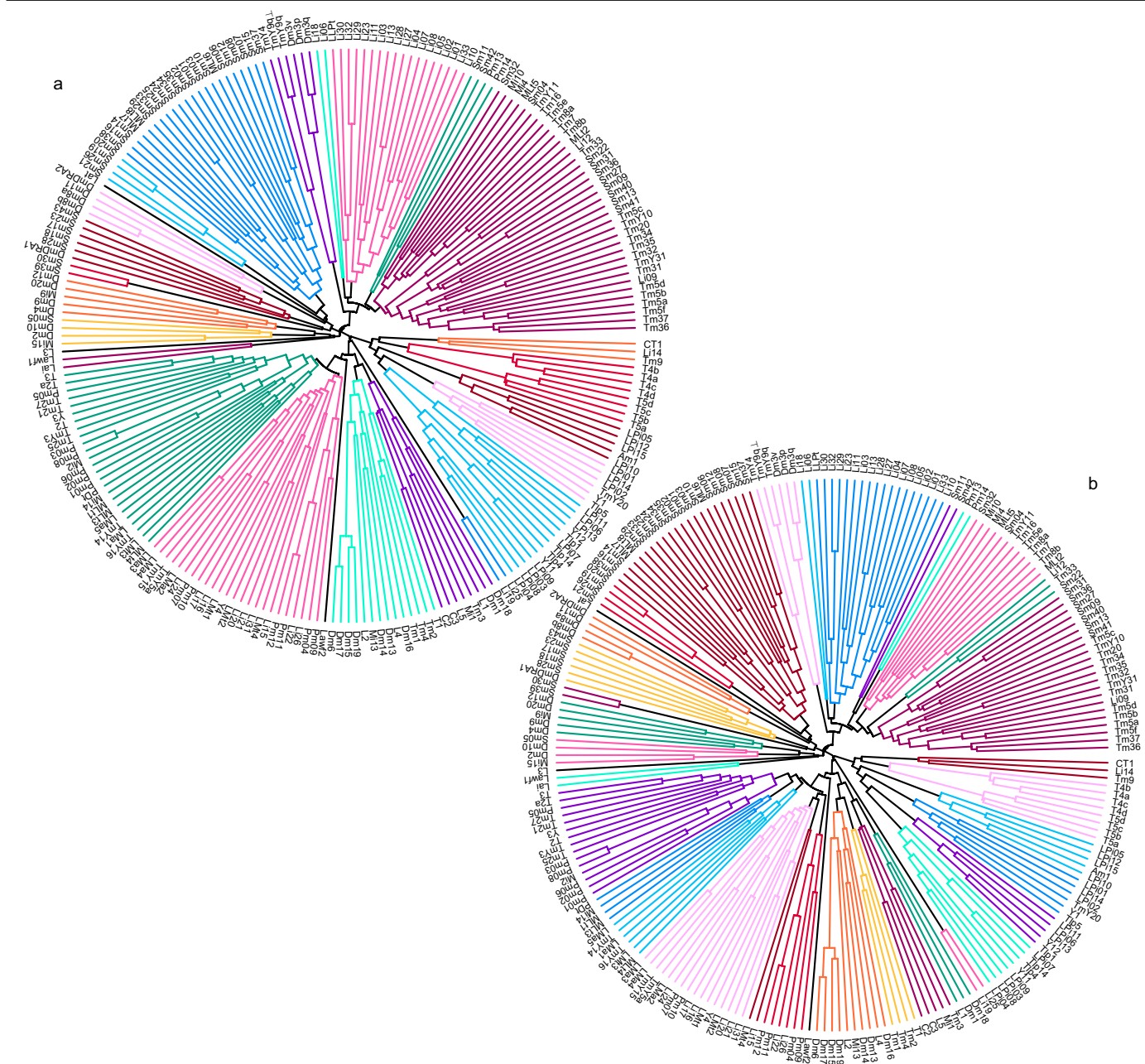

**Extended Data Fig. 10 | Carving the dendrogram to yield finer clusters.** The hierarchical clustering was coloured in Fig. 2c to indicate 19 flat clusters at a threshold of 0.9. (a) Lowering the threshold to 0.885 yields 26 clusters (b) Lowering the threshold further to 0.86 yields 36 clusters. Clusters containing a single cell type are uncoloured (black). R1-6 and L3 are separate clusters in both panels.

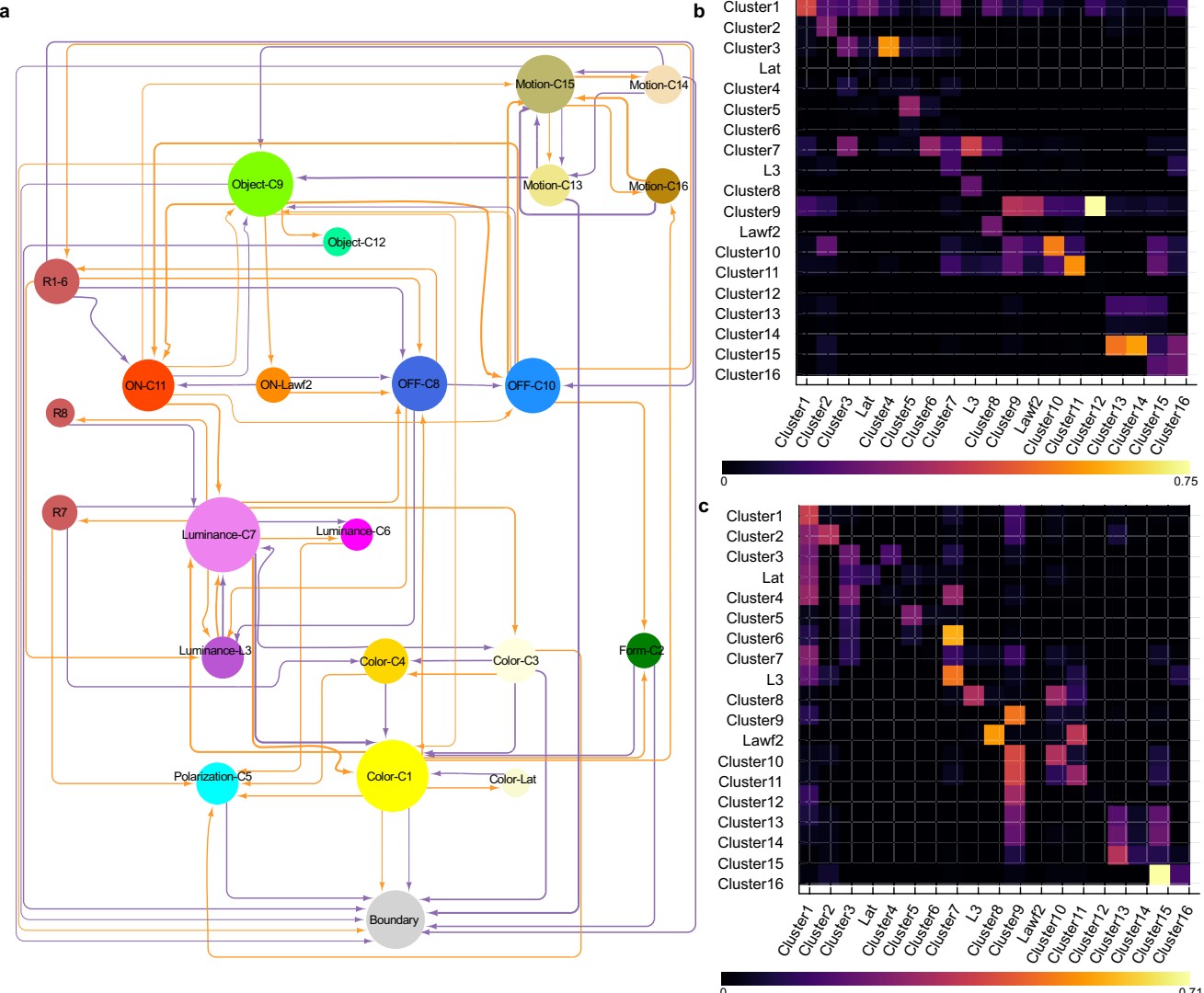

**Extended Data Fig. 11 | Wiring diagram of type clusters (major input and output connections). a**, Wiring diagram depicting major input and output connections between type clusters of Fig. 2c. Node size encodes the number of drawn connections. For each cluster major inputs are drawn as orange inbound edges, and major outputs as purple outbound edges. Major input/output connection is defined as having at least 50% synapses relative to top input/ output connection respectively, excluding loops. **b**, Heatmap is strength of connectivity (fraction of input synapses to post) from pre- to post-synaptic cluster. Heatmap maximum of 0.75. **c**, Strength of connectivity (fraction of output synapses from pre) from pre- to post-synaptic cluster. Heatmap maximum of 0.71.

## Extended Data Table 1 | Type families and their properties

| Family | Affinity | Linkage | Types | Cells | Trans | Neuropils |
|---|---|---|---|---|---|---|
| Centrifugal | Cross Neuropil | Axon Bearing / Columnar | 2 | 1511 | GABA | ME → ME, LA |
| Distal Medulla | Neuropil Intrinsic | Non-columnar | 21 | 3284 | GLUT | ME → ME |
| Distal Medulla Dorsal Rim Area | Neuropil Intrinsic | Non-columnar | 2 | 35 | GLUT | ME → ME |
| Lamina Intrinsic | Neuropil Intrinsic | Non-columnar | 1 | 231 | | LA → LA |
| Lamina Monopolar | Cross Neuropil | Axon Bearing / Columnar | 5 | 3831 | ACH | ME, LA → ME |
| Lamina Tangential | Neuropil Intrinsic | Axon Bearing / Tangential | 1 | 6 | | ME, LO, AME, PLP → LO |
| Lamina Wide Field | Cross Neuropil | Axon Bearing / Columnar | 2 | 320 | ACH | ME → LA |
| Lobula Intrinsic | Neuropil Intrinsic | Non-columnar | 33 | 761 | GABA | LO → LO |
| Lobula Lobula Plate Tangential | Cross Neuropil | Axon Bearing / Tangential | 1 | 36 | GABA | LO, LOP → LO, LOP |
| Lobula Medulla Amacrine | Cross Neuropil | Amacrine | 6 | 134 | GABA | ME, LO → ME, LO |
| Lobula Medulla Tangential | Cross Neuropil | Axon Bearing / Tangential | 4 | 88 | GLUT | LO, ME → LO, ME |
| Lobula Plate Intrinsic | Neuropil Intrinsic | Non-columnar | 15 | 363 | GLUT | LOP → LOP |
| Medulla Intrinsic | Neuropil Intrinsic | Axon Bearing / Columnar | 8 | 3922 | ACH | ME → ME |
| Medulla Lobula Lobula Plate Amacrine | Cross Neuropil | Amacrine | 1 | 1 | GABA | LOP, ME, LO → LOP, LO, ME |
| Medulla Lobula Tangential | Cross Neuropil | Axon Bearing / Tangential | 8 | 295 | ACH | ME, LO → ME, LO |
| Photo Receptors | Neuropil Intrinsic | Axon Bearing / Columnar | 3 | 4751 | | ME, LA → LA, ME |
| Proximal Distal Medulla Tangential | Neuropil Intrinsic | Axon Bearing / Tangential | 1 | 6 | DA | ME → ME |
| Proximal Medulla | Neuropil Intrinsic | Non-columnar | 14 | 599 | GABA | ME → ME |
| Serpentine Medulla | Neuropil Intrinsic | Non-columnar | 43 | 1488 | GABA | ME → ME |
| T Neuron | Cross Neuropil | Axon Bearing / Columnar | 12 | 9252 | ACH | ME, LO, LOP → LOP, LO, ME |
| Translobula Plate | Cross Neuropil | Axon Bearing / Columnar | 4 | 172 | GLUT | LOP → LOP, LO |
| Transmedullary | Cross Neuropil | Axon Bearing / Columnar | 26 | 8855 | ACH | ME, LO → ME, LO |
| Transmedullary Y | Cross Neuropil | Axon Bearing / Columnar | 12 | 2598 | ACH | ME, LOP, LO → LO, ME, LOP |
| Y Neuron | Cross Neuropil | Axon Bearing / Columnar | 5 | 631 | GLUT | LOP, ME, LO → LO, LOP, ME |

Families of optic-lobe intrinsic types. Number of types/cells in each family, predicted neurotransmitter type and primary synapse regions.

**Extended Data Table 2 | Distribution of synapses over neuropils for each type family**

| Family | Affinity | Linkage | Types | Cells | Trans | Neuropils |
|---|---|---|---|---|---|---|
| Centrifugal | Cross Neuropil | Axon Bearing / Columnar | 2 | 1511 | GABA | ME → ME, LA |
| Distal Medulla | Neuropil Intrinsic | Non-columnar | 21 | 3284 | GLUT | ME → ME |
| Distal Medulla Dorsal Rim Area | Neuropil Intrinsic | Non-columnar | 2 | 35 | GLUT | ME → ME |
| Lamina Intrinsic | Neuropil Intrinsic | Non-columnar | 1 | 231 | | LA → LA |
| Lamina Monopolar | Cross Neuropil | Axon Bearing / Columnar | 5 | 3831 | ACH | ME, LA → ME |
| Lamina Tangential | Neuropil Intrinsic | Axon Bearing / Tangential | 1 | 6 | | ME, LO, AME, PLP → LO |
| Lamina Wide Field | Cross Neuropil | Axon Bearing / Columnar | 2 | 320 | ACH | ME → LA |
| Lobula Intrinsic | Neuropil Intrinsic | Non-columnar | 33 | 761 | GABA | LO → LO |
| Lobula Lobula Plate Tangential | Cross Neuropil | Axon Bearing / Tangential | 1 | 36 | GABA | LO, LOP → LO, LOP |
| Lobula Medulla Amacrine | Cross Neuropil | Amacrine | 6 | 134 | GABA | ME, LO → ME, LO |
| Lobula Medulla Tangential | Cross Neuropil | Axon Bearing / Tangential | 4 | 88 | GLUT | LO, ME → LO, ME |
| Lobula Plate Intrinsic | Neuropil Intrinsic | Non-columnar | 15 | 363 | GLUT | LOP → LOP |
| Medulla Intrinsic | Neuropil Intrinsic | Axon Bearing / Columnar | 8 | 3922 | ACH | ME → ME |
| Medulla Lobula Lobula Plate Amacrine | Cross Neuropil | Amacrine | 1 | 1 | GABA | LOP, ME, LO → LOP, LO, ME |
| Medulla Lobula Tangential | Cross Neuropil | Axon Bearing / Tangential | 8 | 295 | ACH | ME, LO → ME, LO |
| Photo Receptors | Neuropil Intrinsic | Axon Bearing / Columnar | 3 | 4751 | | ME, LA → LA, ME |
| Proximal Distal Medulla Tangential | Neuropil Intrinsic | Axon Bearing / Tangential | 1 | 6 | DA | ME → ME |
| Proximal Medulla | Neuropil Intrinsic | Non-columnar | 14 | 599 | GABA | ME → ME |
| Serpentine Medulla | Neuropil Intrinsic | Non-columnar | 43 | 1488 | GABA | ME → ME |
| T1 Neuron | Cross Neuropil | Axon Bearing / Columnar | 1 | 738 | | ME, LA → ME, LA |
| T2 Neuron | Cross Neuropil | Axon Bearing / Columnar | 2 | 1591 | ACH | ME, LO → LO, ME |
| T3 Neuron | Cross Neuropil | Axon Bearing / Columnar | 1 | 823 | ACH | ME, LO → LO, ME |
| T4 Neuron | Cross Neuropil | Axon Bearing / Columnar | 4 | 3104 | ACH | ME, LOP → LOP, ME |
| T5 Neuron | Cross Neuropil | Axon Bearing / Columnar | 4 | 2996 | ACH | LO, LOP → LOP, LO |
| Translobula Plate | Cross Neuropil | Axon Bearing / Columnar | 4 | 172 | GLUT | LOP → LOP, LO |
| Transmedullary | Cross Neuropil | Axon Bearing / Columnar | 26 | 8855 | ACH | ME, LO → ME, LO |
| Transmedullary Y | Cross Neuropil | Axon Bearing / Columnar | 12 | 2598 | ACH | ME, LOP, LO → LO, ME, LOP |
| Y Neuron | Cross Neuropil | Axon Bearing / Columnar | 5 | 631 | GLUT | LOP, ME, LO → LO, LOP, ME |

Families of optic-lobe intrinsic types and the number of their input / output synapses in each of the optic lobe regions.

**Extended Data Table 3 | Cells and cell types by super class**

| Family | Abbrev. | in LA | in ME | in LO | in LOP | out LA | out ME | out LO | out LOP |
|---|---|---|---|---|---|---|---|---|---|
| Centrifugal | C | 2361 | 110557 | 0 | 5 | 25499 | 192287 | 0 | 6 |
| Distal Medulla | Dm | 109 | 493242 | 29 | 0 | 3 | 461042 | 66 | 0 |
| Distal Medulla Dorsal Rim Area | DmDRA | 0 | 5510 | 0 | 0 | 0 | 6164 | 0 | 0 |
| Lamina Intrinsic | Lai | 53890 | 14 | 0 | 0 | 4729 | 0 | 0 | 0 |
| Lamina Monopolar | L | 118627 | 335509 | 0 | 0 | 4644 | 1018207 | 0 | 0 |
| Lamina Tangential | Lat | 0 | 168 | 138 | 0 | 0 | 0 | 11 | 0 |
| Lamina Wide Field | Lawf | 212 | 76784 | 0 | 0 | 39998 | 583 | 0 | 0 |
| Lobula Intrinsic | Li | 0 | 165 | 465695 | 242 | 0 | 5 | 311758 | 25 |
| Lobula Lobula Plate Tangential | LLPt | 0 | 0 | 18515 | 1542 | 0 | 0 | 13572 | 6752 |
| Lobula Medulla Amacrine | LMa | 0 | 222413 | 118357 | 2571 | 0 | 139822 | 115942 | 585 |
| Lobula Medulla Tangential | LMt | 0 | 6226 | 42542 | 72 | 0 | 22192 | 26123 | 25 |
| Lobula Plate Intrinsic | LPi | 0 | 1 | 3 | 320696 | 0 | 36 | 13 | 222563 |
| Medulla Intrinsic | Mi | 318 | 530040 | 2 | 0 | 1 | 1075969 | 16 | 0 |
| Medulla Lobula Lobula Plate Amacrine | Am | 0 | 3201 | 2183 | 27387 | 0 | 3449 | 3762 | 12089 |
| Medulla Lobula Tangential | MLt | 0 | 33346 | 2088 | 10 | 0 | 28384 | 10823 | 1 |
| Photo Receptors | R | 1356 | 21647 | 0 | 0 | 108499 | 27109 | 0 | 0 |
| Proximal Distal Medulla Tangential | PDt | 0 | 2978 | 0 | 0 | 0 | 1186 | 0 | 0 |
| Proximal Medulla | Pm | 0 | 966964 | 3 | 350 | 0 | 444691 | 1 | 265 |
| Serpentine Medulla | Sm | 0 | 240804 | 10 | 0 | 0 | 239769 | 25 | 0 |
| T1 Neuron | T1 | 6513 | 100560 | 0 | 0 | 127 | 2629 | 0 | 0 |
| T2 Neuron | T2 | 0 | 207836 | 14421 | 229 | 0 | 46074 | 176480 | 85 |
| T3 Neuron | T3 | 0 | 117157 | 7188 | 18 | 0 | 24435 | 81576 | 0 |
| T4 Neuron | T4 | 0 | 210989 | 32 | 28749 | 0 | 26568 | 19 | 240567 |
| T5 Neuron | T5 | 0 | 7 | 216802 | 20094 | 0 | 0 | 22862 | 257408 |
| Translobula Plate | Tlp | 0 | 46 | 2767 | 92321 | 0 | 176 | 16095 | 51725 |
| Transmedullary | Tm | 0 | 939034 | 92457 | 920 | 0 | 1025728 | 583491 | 1679 |
| Transmedullary Y | TmY | 0 | 363265 | 111320 | 151107 | 0 | 147000 | 202629 | 65437 |
| Y Neuron | Y | 0 | 82196 | 23732 | 84795 | 0 | 55853 | 78668 | 56706 |

Proofread cell and type stats broken up by super class in the FlyWire connectome dataset as of October 2023.

# Reporting Summary

## Statistics

For all statistical analyses, confirm that the following items are present in the figure legend, table legend, main text, or Methods section.

| n/a | Confirmed | |
|---|---|---|
| ☒ | ☐ | The exact sample size (*n*) for each experimental group/condition, given as a discrete number and unit of measurement |
| ☒ | ☐ | A statement on whether measurements were taken from distinct samples or whether the same sample was measured repeatedly |
| ☒ | ☐ | The statistical test(s) used AND whether they are one- or two-sided *Only common tests should be described solely by name; describe more complex techniques in the Methods section.* |
| ☒ | ☐ | A description of all covariates tested |
| ☒ | ☐ | A description of any assumptions or corrections, such as tests of normality and adjustment for multiple comparisons |
| ☒ | ☐ | A full description of the statistical parameters including central tendency (e.g. means) or other basic estimates (e.g. regression coefficient) AND variation (e.g. standard deviation) or associated estimates of uncertainty (e.g. confidence intervals) |
| ☒ | ☐ | For null hypothesis testing, the test statistic (e.g. *F*, *t*, *r*) with confidence intervals, effect sizes, degrees of freedom and *P* value noted *Give P values as exact values whenever suitable.* |
| ☒ | ☐ | For Bayesian analysis, information on the choice of priors and Markov chain Monte Carlo settings |
| ☒ | ☐ | For hierarchical and complex designs, identification of the appropriate level for tests and full reporting of outcomes |
| ☒ | ☐ | Estimates of effect sizes (e.g. Cohen's *d*, Pearson's *r*), indicating how they were calculated |

*Our web collection on statistics for biologists contains articles on many of the points above.*

## Software and code

Policy information about availability of computer code

| Data collection | This study is based on the FlyWire fly brain connectome (flywire.ai). List of all data sources used for making this resource is available here: https://codex.flywire.ai/about_flywire (along with publication rules and TOS). |
|---|---|
| Data analysis | Python, Julia programming languages were used for analysis. Some of the tools from FlyWire Codex: https://github.com/murthylab/codex |

For manuscripts utilizing custom algorithms or software that are central to the research but not yet described in published literature, software must be made available to editors and reviewers. We strongly encourage code deposition in a community repository (e.g. GitHub). See the Nature Portfolio guidelines for submitting code & software for further information.

## Data

Policy information about availability of data

All manuscripts must include a data availability statement. This statement should provide the following information, where applicable:
- Accession codes, unique identifiers, or web links for publicly available datasets
- A description of any restrictions on data availability
- For clinical datasets or third party data, please ensure that the statement adheres to our policy

Data produced by this study is available at: https://github.com/murthylab/flywire-visual-neuron-types

# Research involving human participants, their data, or biological material

Policy information about studies with human participants or human data. See also policy information about sex, gender (identity/presentation), and sexual orientation and race, ethnicity and racism.

| | |
|---|---|
| Reporting on sex and gender | N/A |
| Reporting on race, ethnicity, or other socially relevant groupings | N/A |
| Population characteristics | N/A |
| Recruitment | N/A |
| Ethics oversight | N/A |

Note that full information on the approval of the study protocol must also be provided in the manuscript.

# Field-specific reporting

Please select the one below that is the best fit for your research. If you are not sure, read the appropriate sections before making your selection.

☒ Life sciences      ☐ Behavioural & social sciences      ☐ Ecological, evolutionary & environmental sciences

For a reference copy of the document with all sections, see nature.com/documents/nr-reporting-summary-flat.pdf

# Life sciences study design

All studies must disclose on these points even when the disclosure is negative.

| | |
|---|---|
| Sample size | N/A |
| Data exclusions | N/A |
| Replication | N/A |
| Randomization | N/A |
| Blinding | N/A |

# Reporting for specific materials, systems and methods

We require information from authors about some types of materials, experimental systems and methods used in many studies. Here, indicate whether each material, system or method listed is relevant to your study. If you are not sure if a list item applies to your research, read the appropriate section before selecting a response.

## Materials & experimental systems

| n/a | Involved in the study |
|---|---|
| ☒ ☐ | Antibodies |
| ☒ ☐ | Eukaryotic cell lines |
| ☒ ☐ | Palaeontology and archaeology |
| ☒ ☐ | Animals and other organisms |
| ☒ ☐ | Clinical data |
| ☒ ☐ | Dual use research of concern |
| ☒ ☐ | Plants |

## Methods

| n/a | Involved in the study |
|---|---|
| ☒ ☐ | ChIP-seq |
| ☒ ☐ | Flow cytometry |
| ☒ ☐ | MRI-based neuroimaging |

## Plants

Seed stocks

N/A

Novel plant genotypes

N/A

Authentication

N/A

