## [Peer Review File · Nature]

Manuscript Title: Neuronal “parts list” and wiring diagram for a visual system

Reviewer Comments & Author Rebuttals

Reviewer Reports on the Initial Version:

Referees' comments:

Referee #1 (Remarks to the Author):

The manuscript by Matsliah et al. describes the connectome of *Drosophila* optic lobes. The base data, which covers the entire optic lobes, was obtained through a combination of crowdsourced annotation and experts' labeling and proofreading. Contrast to the previous study of seven medulla columns (Takemura et al., 2015) which focuses on the detailed synaptic circuits of columnar neurons, the large spatial coverage of this study provides exciting possibilities of new discoveries, including new cell types, spatial organization of cells, and cross-neuropil connectivity. These were explored in various degrees using computational analyses of the connectome. In particular, cell types can be automatically assigned by cell-cell distances in the connectivity-based feature space. Cell-type connectivity patterns derived from connectome further generalize the wiring diagram and provide interesting insights into their roles in various visual functions.

Major issues:

(1) New cell types, based on connectome, are arguably the most exciting aspect of this study. While the authors explained various validation methods (such as check for tiling), I am wondering if partially traced cells could mistaken as new cell types since missing neurites affects both connectivity and morphology assignment. For example, could Tm5e or Tm5f be partially traced Tm5c? The quality of tracing neurites and synaptic assignment is difficult to assess in this study given the complex procedures used and the lack of clear assessment. Could the authors compare connectivity and synapse numbers of the 20 modular neurons with those described in the seven-column study?

(2) The spatial organization and unit structures could be better explored. How many medulla columns are included in the data set? The numbers of most known modular neurons (present in every column) are around 780 or less— consistent with the expected column number. However, the number of T2a is considerably higher. Are T2a tilted and/or is not modular? Are any the cells implicated in color vision (Figure 4b) p- or y-column specific (except the known y-specific Tm5a)? Lobula organization has been disputed. Are there lobula columns (and the number) and modular lobula neurons revealed in this study?

Minor issues:

(1) The Sm class might be the M6-LN cells identified previously (Chin et al., J. Comp. Neurol. 2014).

(2) R7 and R8 connectivity should be included in Figure 4b.

Referee #2 (Remarks to the Author):

In this paper, Matsliah et. al. constructed a “part list” (a catalog of neuron types) of the fly optic lobe based on anatomy and connectivity. They greatly expanded the list of cell types in existing cell families and identified new families containing various cell types among multiple neuropils. Cell typing in this study is based on neuroanatomy and connectivity, using the FlyWire connectome data. The great achievements of this study have been made possible by the combined efforts of researchers and the community (citizen scientists), and a tremendous amount of work has been done. The preliminary clustering of over 37,000 labeled and proofread neurons was refined into 224 cell types with a computational method based on their connectivity. Errors in type assignments were auto-corrected by re-assigning cell types and manual inspection. Morphological variations and special coverage were carefully considered during cell type assignment and validation. This study also generated a wiring diagram of cell types, and contributed to cell typing in aspects of methodology and implications on mammals. Overall, this is a novel study that generated a comprehensive connection map in a complex visual system.

The methods used in this study are clearly described and make sense to experimental neuroscientists with limited expertise in computational clustering and cell typing. Using connectivity achieves more functionally feasible cell typing than using anatomy alone and provides a more reliable reference for future functional study. The enormous amount of data involved in this study is amazing; the community efforts in cell labeling and proofreading are irreplaceable, bringing science into civilians and making use of their great potential in supporting scientific research. The way that the data are presented is mostly readable and highlights major conclusions without wasting the main figure space on too many details. The conclusions of this paper are mostly valid and reliable regarding the neurons involved in the study. The cell type assignment seems to be unbiased; extreme conditions such as variability in connectivity or morphology have been carefully examined and rightfully treated. Additionally, the authors used spatial coverage to further validate some of the classifications, which makes the conclusions more reliable. However, as the authors mentioned, there are still non-proofread cells in the connectome and some cell typing results, especially the types with a small number of cells, might be affected by this fact; this might also affect the implications of their functions. With the release of new FlyWire versions, the authors should re-visit the data to assign cell types for the newly proofread cells and validate the less numerous cell types.

This paper is well written with the proper amount of context for general neuroscientists to understand. It might not be suitable for a more general audience with a limited background in neuroscience or a very basic knowledge of machine learning and clustering. This paper has summarized the past and current stages of research on connectome, optic system, and cell typing, crediting key findings in the field and properly stating the contribution of this research. Overall, this is an innovative study that provides a

foundation for future experimental (functional) study and insight into other subdivisions of nervous systems or other animals.

The inclusion of some light microscopy data to be used as a ground truth to validate the reconstruction will be a significant addition to this study. There are many split-Gal4 lines available that target sparse subsets of neurons in *Drosophila* Visual circuits. I wonder if the authors have attempted to identify split-Gal4 lines that target the new cell types identified in this connectome. Adding the expression pattern and list of such split-Gal4 lines will be an invaluable starting point for experimental neuroscientists interested in the development (formation) and/or function of *Drosophila* visual circuits.

In addition, adding a detailed manual (text or video guide as supplementary data) on how to access and navigate the EM dataset (navigating the Codex website and browsing data) will be another useful addition. Some labs do amazing work on circuits but have little or no experience in working with large-scale connectome data. A beginner-level manual will make this highly comprehensive connectome data less intimidating for such laboratories.

Referee #3 (Remarks to the Author):

Key results, relevance, and originality of the work

When functionally exploring unknown neuronal territory, knowledge on cell types and synaptic connectivity has proven extremely helpful or even a game-changer. Maps on cells and cell-connectivity do not explain how neuronal circuits process information. However, the information included in such maps enables the design of efficient experiments and aids in interpreting obtained data. This is even more the case in the fruit fly that has emerged as an important genetic model system for interrogating the neuronal underpinning of computations or algorithms and ultimately behavior.

In their submitted manuscript 'Neuronal "parts list" and wiring diagram for a visual system', Matsliah et al., the team around Sebastian Seung and the FlyWire Consortium, provide us with a highly detailed, breath-taking continuation of the analysis of the connectome of a large part of the optic lobe of the fruit fly *Drosophila*. Their analysis is based on a very large EM-dataset on a part of the brain of a single *Drosophila* adult female fly, which was reconstructed and released by the FlyWire Consortium (Dorkenwald, Matsliah, et al., 2023; Schlegel et al., 2023). The authors impressively demonstrate that reconstruction of a large part of the fly visual system – a neuronal system of high synaptic complexity and small to medium size – from serial electron microscopic images with synaptic resolution has now turned into real.

This raises great hope that similar analysis and data might 'soon become available' for the entire brain in the head of the fly and the thoracic ganglion (the entire central nervous system) of *Drosophila*. For visual neuroscience research in the fly, the study represents an important advance because it fills white spots in previously published work and in recently released studies on the neuronal wiring diagram of the *Drosophila* brain and fly visual system, including studies of the same team that explore the same dataset

(Zheng et al., 2018; Kind et al., (2021);..... Dorkenwald, Matsliah, et al., 2023; Schlegel et al., 2023).

The study of Matsliah et al. adds highly valuable information to existing knowledge on the number of cells and cell types in the optic lobe and they include detailed information on chemical synaptic connectivity and its strength. By their connectome-based approach, the authors identify new cell types that were so far difficult or impossible to tell apart based on cell anatomy, branching and co-stratification. Matsliah and coworkers accomplish this task by combining traditional methods for cell identification with computer-/ algorithm-based data analysis, in particular for the analysis of chemical synaptic connectivity; they subsequently validate the results by statistical methods.

Next to the information on functional connectivity, the 'unbiased computational' approach allows the authors to identify new cell types that are on average members of less numerous types, including 'few-cell' and 'single-cell' types that so far notoriously escaped analysis. This includes the introduction of entire new and large families, like the serpentine medulla intrinsic cells. In total, the study doubles the number of known cell types but this does not result in an explosion of complexity. Rather their elegant approach enables a reduction of functional complexity. A feature vector for each neuron type is defined based on connectivity with cell types, and the authors observe that a lower dimensionality feature vector with only optic lobe intrinsic types (covering 80% of the synapses) is sufficient for defining cell types. The suggested cell types are then validated in several statistical ways. Ultimately, the combined use of cell anatomy and synaptic connectivity enables the assignment of in total 37.500 intrinsic neurons of a single right optic lobe to 224 intrinsic cell types. This strong reduction in complexity, together with a type-to-type connection matrix is extremely helpful for research on the neuronal circuits of vision in *Drosophila* and fosters a comparative perspective on other visual systems, in particular the vertebrate visual system.

Any more detailed description of the findings of the study would quickly turn into a 'discussion among experts on the neuronal circuits of fly vision'. The provided data will certainly be of strong influence for future studies on visually guided behavior (I am not fully convinced that perception of visual features like color, motion, and form can indeed be accessed in the fly, as mentioned by the authors) and research on the underlying neuronal computations. This data will definitely accelerate research on the identification and characterization of visual circuits. I expect this study to be highly relevant for future research on fly vision.

Data, validity, and methodology

To my knowledge, the underlying data set and analysis represents the current state of the art. Future data sets might enable higher optical resolution and include electric synapses (innexin junctions), which would represent a dramatic further advance. However, at present, data on electric connectivity is only available on much smaller datasets.

The methodology for data analysis seems to be carefully chosen and validated. However, frankly speaking, I was never directly involved in the analysis of this data or development of algorithms. This reduced the weight of my judgement. From the perspective of an outsider of connectome data analysis, the methods seemingly represent the culmination of a long thought process within the community of

researchers pushing connectomics. Sebastian Seung and his team are leading scientists in this field. At my level of understanding, the reporting, methodology, and treatment of data is robust, reliable, and support the conclusions. An in-depth analysis / review of the methodology and appropriateness of statistical testing I have to leave to experts in the field.

General

The manuscript is well written and data is presented very well. I cannot identify any inflammatory material. Abstract and introduction are easily accessible. The manuscript contains a very large amount of supplementary data, in particular the 'ocean of online data' that is impossible to review and that is where much of the music plays.

The manuscript clearly is of high interest to readers with interest in fly vision and represents a conceptual step forward, given that photoreceptors become included.

Suggestions to strengthen the study

The about 6000 photoreceptors of each ommatidium are pivotal for vision and knowledge on the feedforward and feedback connectivity of photoreceptors and their synaptic partner neurons is pivotal for the rigorous functional analysis of most visual circuits. From my perspective, it is a major weakness of the present study that photoreceptors have been actively excluded from the present dataset and that the authors plan to submit this information separately.

The lack of data on photoreceptors and their connectivity to identified postsynaptic cell types in the lamina and medulla contradicts the central promise made by the authors to provide a part list on the fly visual system. At present, the submitted work is an 'Incomplete neuronal "parts list" and wiring diagram for the visual system.' that might not fully meet the high standard of publication in Nature.

However, this could be changed by including photoreceptors and connectivity into the study, which would result in a state-of-the-art part list and wiring diagram of the fly visual system of sufficient general interest to the broad readership of Nature.

See manuscript page 24, Future Releases:

>>One major change will be the addition of the remaining photoreceptors, which by now have been completely proofread and typed, and are already accessible via CAVEclient (Dorkenwald, Matsliah, et al. 2023; Dorkenwald, Schneider-Mizell, et al. 2023)<<

Minor

♣ Although I want to leave it to the authors how this could be changed: large parts of the results section are lengthy and have the character of an encyclopedia written for experts. This also results in large volume of text. A brief example:

>>Of the types with partial coverage that contain multiple cells, $Sm \leftarrow CB0566 \leftarrow MeTu2$, $Sm \leftarrow MC65 \Rightarrow TmY5a$, $Sm \leftarrow MeTu3 \Rightarrow MeTu2$, $Sm \leftarrow MeTu4 \leftarrow MeTu1$, $Sm \leftarrow MeTu4 \Rightarrow CB0566$, $Sm \leftarrow Mi15 \Rightarrow MLt5$, $Sm \leftarrow Mi9 \Rightarrow Tm32$, and $Sm \Rightarrow TmY3$ are dorsal-only. Ventral-only types include $Sm \leftarrow Mi9 \Rightarrow CB0165$, $Sm \leftarrow TmY5a \Rightarrow Tm7$, and $Sm \leftarrow Y12 \Rightarrow TmY10$. Notably, $Sm \leftarrow Mi9 \Rightarrow Tm32$ resembles

similar stratification to Mi3 in upper and lower boundaries of the M7 layer. However, $Sm \leftarrow Mi9 \Rightarrow Tm32$ exhibits an additional stratification within the M4 layer....<<

Could some of this information / similar paragraphs be partially moved to the supplement? This would also enable more visible highlighting of results with broader relevance.

♣ Can information be added on expected changes for an optic lobe of a male fly?

In same context:

Page 20, end of paragraph 4

>>(Note that our wiring diagram is derived from a female brain, but is likely to be the same or similar for a male brain, as there are few sexual dimorphisms within the optic lobe.) <<

This statement deserves a citation.

♣ page 4, paragraph 3 >>To refine these coarse class distinctions,.....<<

Do the authors really know whether axons are indeed axons (as defined on page 12)? Or rather mixed processes?...

Wouldn't it be better to call them main neurites or processes, unless there is clear evidence for axon or dendrite? Didi I miss that there is clear evidence in all cases where axon are mentioned?

♣ page 8, Type-to-type connectivity

Dot size. Dot size is used to display the strength of synaptic connectivity, the encoding of numbers in size is non-linear. Although difficult to display, in its present form the display is not really helpful as there is little variation between dots that are all very small.....

Could this be solved by in addition providing the number of synapses in a further table / tables in the supplement? I might have missed that such tables are available online?

♣ page 8, Type-to-type connectivity, paragraph 2

>>The threshold depends on the scientific question to be addressed.<<

Does this mean that analysis of circuitries of motion vision involves other thresholds than for instance detection of form or color? Can the authors please explain this cryptic statement?

♣ page 8, Type-to-type connectivity, last paragraph

How many false positives are expected?

♣ page 10, paragraph 2 >>Almost 400 Dm8 cells have been proofread and identified in v630...<<

Why does this para on Dm cells appear here? I suggest moving the para to page 13/14 where results on Dm cell are presented.

♣ page 10, paragraph 4 <<But overall Tm8a and Tm8b differ markedly in connectivity. $Li \Rightarrow LC12 \Rightarrow Tm8a$ is the strongest output of Tm8a, but weak or nonexistent as an output of Tm8b...<<

The devised discriminative names are very helpful in general. However, I cannot understand the name in

the example above and few other places. On page 3 of the manuscript: For example, $Li \leftarrow TmY4 \Rightarrow LT79$ is the Lobula intrinsic (Li) neuron that receives input from TmY4 and sends output to LT79
How can this logic be applied to the example above such that output from Tm8a to Li.....becomes visible? Did I miss something?

♣ page 14, paragraph 2

In the section on Dm8p and Dm8y the work of Li et al., 2021 should be cited as they were the first and only ones to reveal that p and y types of Dm8 are indeed functionally different.

♣ page 16, LPi Lobula plate intrinsic, paragraph 2

>>H2 is a lobula plate tangential cell (LPTC) that prefers back-to-front motion (Wei et al. 2020).<<
This citation is not suitable as Wei et al did not perform any experiments on the directional tuning / receptive field layout of LPTCs.

♣ page 19, Discussion

>>Early advocates of connectomics predicted that deriving cell types from connectivity would be transformative (H. Sebastian Seung 2012). Now that the connectomic era has finally dawned, we can begin to see how well this prediction holds up.<<

This notion does not enable any new insight or perspective on the presented data and should therefore become deleted.

♣ page 20, Discussion

>>Color preference in *Drosophila* varies over the day (Schnaitmann et al. 2018).....<<

This citation is wrong / does not support the given statement as Schnaitmann et al. 2018 did not at all investigated color preference behavior.

A study suggesting changing color preference during the course of the day has been published by Lazopulo et al. in Oct. 2019 in Nature. 2019 Oct;574(7776):108-111. doi: 10.1038/s41586-019-1571-y. Epub 2019 Sep 18.

♣ page 21, Discussion

>>Single cell transcriptomics is providing detailed information about the molecules in fly visual neurons (Kurmangaliyev et al. 2020; Özel et al. 2021; Konstantinides et al. 2022).<<

Do the authors really mean 'single cell transcriptomics' or rather 'single cell type transcriptomics'? Based on what are cells / cell types defined in these studies?

♣ Discussion:

I might have missed this: has the complete lack of information on electric coupling be discussed at all? Including electric coupling might fundamentally change current functional interpretations of chemical connectivity (...mind the gap (junctions)!).

Author Rebuttals to Initial Comments:

Overview of major changes to the manuscript

We thank the referees for their informative comments, and for reading the manuscript so carefully. A major revision was prompted by Referee #3, who commented that “large parts of the results section are lengthy and have the character of an encyclopedia written for experts.” We reframed the narrative by utilizing the dendrogram of cell types, which was previously buried in the Supplementary Figures. The dendrogram was thresholded to yield clusters of cell types, and proposed interpretations of the clusters in terms of visual functions like motion, object, and color vision (Fig. 2). The revised manuscript is now organized around the functional subsystems, which makes the paper accessible and interesting to a broader audience. The previous narrative, a litany of neuropil-defined families, has been moved to the Methods, and is still valuable for specialists.

As Referee #1 suggested, comparisons with the HHMI Janelia seven-column medulla reconstruction have been performed.

As requested by Referee #2, the revised manuscript has been upgraded to v783 of the proofreading. This includes the photoreceptors requested by Referee #3.

The discriminative names for interneuron types were intended to be temporary, and have now been discarded in favor of short names.

Fig. 1 now defines a hierarchy of class, family, and type.

Many Figures now contain wiring diagrams of connectivity between cell types. For simplicity, only the top input and output connection for each cell type is shown. We find that this visualization technique is a happy medium, informative but not overwhelming.

Referees' comments:

Referee #1 (Remarks to the Author):

The manuscript by Matsliah et al. describes the connectome of *Drosophila* optic lobes. The base data, which covers the entire optic lobes, was obtained through a combination of crowdsourced annotation and experts' labeling and proofreading. Contrast to the previous study of seven medulla columns (Takemura et al., 2015) which focuses on the detailed synaptic circuits of columnar neurons, the large spatial coverage of this study provides exciting possibilities of new discoveries, including new cell types, spatial organization of cells, and cross-neuropil connectivity. These were explored in various degrees using computational analyses of the connectome. In particular, cell types can be automatically assigned by cell-cell distances in the connectivity-based feature space. Cell-type connectivity patterns derived from connectome further generalize the wiring diagram and provide interesting insights into their roles in various visual functions.

Major issues:

(1) New cell types, based on connectome, are arguably the most exciting aspect of this study. While the authors explained various validation methods (such as check for tiling), I am wondering if partially traced cells could be mistaken as new cell types since missing neurites affects both connectivity and morphology assignment.

Proofreading errors can indeed result in misclassification of individual cells. However, it is unlikely to result in a new cell type. Small cells are numerous, and a few bad cells does not change the clustering. For a large cell, large proofreading errors are easy to spot, and small proofreading errors have little quantitative effect on the connectivity and morphology.

For example, could Tm5e or Tm5f be partially traced Tm5c?

This is a great example that illustrates the power of connectivity. Indeed Tm5e and Tm5f look similar to Tm5c based on morphology. But the synaptic partners of these types are very different. Such marked differences could not arise from inadequate proofreading.

The quality of tracing neurites and synaptic assignment is difficult to assess in this study given the complex procedures used and the lack of clear assessment.

These issues are addressed by the Methods section of the “flagship” paper, which is available in preprint form (Dorkenwald et al. 2023). They were also addressed in a previous methods paper (Dorkenwald et al. 2022).

Could the authors compare connectivity and synapse numbers of the 20 modular neurons with those described in the seven-column study?

Thanks for this helpful suggestion. The revised manuscript includes a supplementary figure with such a comparison. There is good agreement, which provides validation that is specific to the optic lobe. This complements the assessments mentioned above.

(2) The spatial organization and unit structures could be better explored.

How many medulla columns are included in the data set?

We have proofread and identified 796 Mi1 cells in the right optic lobe. This is a lower bound on the number of columns, and we think it is very close to the true number. So we are using the round number 800 as our estimate of the number of columns. L1 (793 cells) and L2 (792 cells) are also approaching the 800 number. This is clarified in the section on “numerous” types.

The numbers of most known modular neurons (present in every column) are around 780 or less—consistent with the expected column number. However, the number of T2a is considerably higher. Are T2a tilted and/or is not modular?

Tm3, T2a, T3, and T4c are all represented by more than 800 cells each, more than the number of columns. Like the reviewer, we were surprised by this finding, and the issue deserves further exploration. Assigning every T2a to a column will result in some collisions. We have preliminary evidence that it is more appropriate to assign each T2a cell to an elementary triangle in the hexagonal lattice of columns, because a “typical” T2a cell receives strong connections from three Mi1/Tm1 cells in such a triangle. A hexagonal lattice contains two kinds of elementary triangle with mirrored orientations. According to this idea, T2a cells are not in one-to-one correspondence with columns, nor with triangles; they are just similar in number to the columns.

So to answer the reviewer’s question, the terms “modular” or “uni-columnar” need to be defined more precisely through a careful geometric analysis including “spatial organization” and “unit structures,” which will be the subject of future work.

Are any the cells implicated in color vision (Figure 4b) p- or y-column specific (except the known y-specific Tm5a)?

This is a great question but the task of creating a definitive p/y annotation of the columns is nontrivial and has been left for future work. The proposed methods of doing this (Kind et al. 2021) are somewhat noisy, so more investment of time and effort is needed to figure out the right method. We believe that accurate photoreceptor synapses are the key, and once the next version of synapses becomes available we will tackle the pale/yellow issue.

Lobula organization has been disputed. Are there lobula columns (and the number) and modular lobula neurons revealed in this study?

If we understand correctly, the reviewer hypothesizes that (1) the lobula has some number N_{LO} of columns and (2) there is a set of lobula modular types with cardinality N_{LO} . What we can say at this point is that the cardinalities of Tm and TmY types vary greatly, and the same is true of LC types. So it is unclear whether lobula organization should be conceptualized in terms of columns. The lobula might consist of many overlapping mosaics with different spacing, more like ganglion cells of the mammalian retina. This are just speculations, however, and a careful analysis of the geometric relationships between the columnar types of the lobula is warranted. This requires serious study, and will be the subject of future work.

Minor issues:

(1) The Sm class might be the M6-LN cells identified previously (Chin et al., J. Comp. Neurol. 2014).

We thank the reviewer for bringing this paper to our attention. M6-LN initially sounds completely different from Sm, which primarily stratifies in M7. But the detailed descriptions in the paper indicate that most R7 neurons do not intersect with M6-LNs. Therefore M6-LN cells might be at the M6/M7 border, and could potentially correspond to those Sm types that stratify at this border. However, establishing correspondences between M6-LN and Sm types proves challenging, for the same reason that our Sm type definitions rely on connectivity and spatial coverage rather than morphology. We were unable to find convincing matches that take into account cell body position, stratification, arbor

diameter, and neurotransmitter. Therefore we have cited the paper in the Methods of the revised manuscript, but do not provide correspondences.

(2) R7 and R8 connectivity should be included in Figure 4b.

There is now a new figure on the color system that includes R7 and R8.

Referee #2 (Remarks to the Author):

In this paper, Matsliah et. al. constructed a "part list" (a catalog of neuron types) of the fly optic lobe based on anatomy and connectivity. They greatly expanded the list of cell types in existing cell families and identified new families containing various cell types among multiple neuropils. Cell typing in this study is based on neuroanatomy and connectivity, using the FlyWire connectome data. The great achievements of this study have been made possible by the combined efforts of researchers and the community (citizen scientists), and a tremendous amount of work has been done. The preliminary clustering of over 37,000 labeled and proofread neurons was refined into 224 cell types with a computational method based on their connectivity. Errors in type assignments were auto-corrected by re-assigning cell types and manual inspection. Morphological variations and special coverage were carefully considered during cell type assignment and validation. This study also generated a wiring diagram of cell types, and contributed to cell typing in aspects of methodology and implications on mammals. Overall, this is a novel study that generated a comprehensive connection map in a complex visual system.

The methods used in this study are clearly described and make sense to experimental neuroscientists with limited expertise in computational clustering and cell typing. Using connectivity achieves more functionally feasible cell typing than using anatomy alone and provides a more reliable reference for future functional study. The enormous amount of data involved in this study is amazing; the community efforts in cell labeling and proofreading are irreplaceable, bringing science into civilians and making use of their great potential in supporting scientific research. The way that the data are presented is mostly readable and highlights major conclusions without wasting the main figure space on too many details.

The conclusions of this paper are mostly valid and reliable regarding the neurons involved in the study. The cell type assignment seems to be unbiased; extreme conditions such as variability in connectivity or morphology have been carefully examined and rightfully treated. Additionally, the authors used spatial coverage to further validate some of the classifications, which makes the conclusions more reliable. However, as the authors mentioned, there are still non-proofread cells in the connectome and some cell typing results, especially the types with a small number of cells, might be affected by this fact; this might also affect the

implications of their functions. With the release of new FlyWire versions, the authors should re-visit the data to assign cell types for the newly proofread cells and validate the less numerous cell types.

The paper has been “upgraded” to the v783 release. A changelog is provided in the Supplementary Information. Two Li types were inadvertently omitted from the original submission and have been restored. (One of these types was mentioned in the text but not in the figures.) A third Li type was formerly a single cell “weirdo” but has since been discovered on the left hand side and hence promoted to a type. Two Sm types were merged into a single type consisting of three cells, after comparison with the left hand side. Overall, the number of intrinsic types has changed from 224 to 226.

Types with a small number of cells are not less certain, because we are insisting on spatial coverage. None of our types consist of a few small cells scattered over the visual field. There are a few groupings like this, but they appear to be developmental abnormalities rather than types. Some sentences about these have been added to the section on Morphological Variation.

This paper is well written with the proper amount of context for general neuroscientists to understand. It might not be suitable for a more general audience with a limited background in neuroscience or a very basic knowledge of machine learning and clustering. This paper has summarized the past and current stages of research on connectome, optic system, and cell typing, crediting key findings in the field and properly stating the contribution of this research. Overall, this is an innovative study that provides a foundation for future experimental (functional) study and insight into other subdivisions of nervous systems or other animals.

The inclusion of some light microscopy data to be used as a ground truth to validate the reconstruction will be a significant addition to this study. There are many split-Gal4 lines available that target sparse subsets of neurons in Drosophila Visual circuits. I wonder if the authors have attempted to identify split-Gal4 lines that target the new cell types identified in this connectome. Adding the expression pattern and list of such split-Gal4 lines will be an invaluable starting point for experimental neuroscientists interested in the development (formation) and/or function of Drosophila visual circuits.

This is an important but highly challenging task. Because the new cell types are often difficult or impossible to distinguish based on morphology alone, establishing definitive correspondences with split-Gal4 lines will often require characterizing the connectivity of the cells in the lines, a time-consuming endeavor. Therefore we are leaving it to the research community to establish correspondences on a case by case basis.

In addition, adding a detailed manual (text or video guide as supplementary data) on how to access and navigate the EM dataset (navigating the Codex website and browsing data) will be another useful addition. Some labs do amazing work on circuits but have little or no experience in working with

large-scale connectome data. A beginner-level manual will make this highly comprehensive connectome data less intimidating for such laboratories.

We completely agree, and such resources are described in the flagship paper.

Referee #3 (Remarks to the Author):

Key results, relevance, and originality of the work

When functionally exploring unknown neuronal territory, knowledge on cell types and synaptic connectivity has proven extremely helpful or even a game-changer. Maps on cells and cell-connectivity do not explain how neuronal circuits process information. However, the information included in such maps enables the design of efficient experiments and aids in interpreting obtained data. This is even more the case in the fruit fly that has emerged as an important genetic model system for interrogating the neuronal underpinning of computations or algorithms and ultimately behavior.

In their submitted manuscript 'Neuronal "parts list" and wiring diagram for a visual system', Matsliah et al., the team around Sebastian Seung and the FlyWire Consortium, provide us with a highly detailed, breath-taking continuation of the analysis of the connectome of a large part of the optic lobe of the fruit fly *Drosophila*. Their analysis is based on a very large EM-dataset on a part of the brain of a single *Drosophila* adult female fly, which was reconstructed and released by the FlyWire Consortium (Dorkenwald, Matsliah, et al., 2023; Schlegel et al., 2023). The authors impressively demonstrate that reconstruction of a large part of the fly visual system - a neuronal system of high synaptic complexity and small to medium size - from serial electron microscopic images with synaptic resolution has now turned into real.

This raises great hope that similar analysis and data might 'soon become available' for the entire brain in the head of the fly and the thoracic ganglion (the entire central nervous system) of *Drosophila*. For visual neuroscience research in the fly, the study represents an important advance because it fills white spots in previously published work and in recently released studies on the neuronal wiring diagram of the *Drosophila* brain and fly visual system, including studies of the same team that explore the same dataset (Zheng et al., 2018; Kind et al., (2021);..... Dorkenwald, Matsliah, et al., 2023; Schlegel et al., 2023).

As summarized in our Fig. S1c, the FlyWire Codex contains cell type annotations for central brain neurons (Schlegel et al. 2023; Dorkenwald et al. 2023). Furthermore, 500 types of boundary neurons (straddling the optic lobe and central brain) are annotated in the Codex (Schlegel et al. 2023). We have added some text to the Introduction to explain this.

The study of Matsliah et al. adds highly valuable information to existing knowledge on the number of cells and cell types in the optic lobe and they include detailed information on chemical synaptic connectivity and its strength. By their connectome-based approach, the authors identify new cell types that were so far difficult or impossible to tell apart based on cell anatomy, branching and co-stratification. Matsliah and coworkers accomplish this task by combining traditional methods for cell identification with computer-/ algorithm-based data analysis, in particular for the analysis of chemical synaptic connectivity; they subsequently validate the results by statistical methods.

Next to the information on functional connectivity, the 'unbiased computational' approach allows the authors to identify new cell types that are on average members of less numerous types, including 'few-cell' and 'single-cell' types that so far notoriously escaped analysis. This includes the introduction of entire new and large families, like the serpentine medulla intrinsic cells. In total, the study doubles the number of known cell types but this does not result in an explosion of complexity. Rather their elegant approach enables a reduction of functional complexity. A feature vector for each neuron type is defined based on connectivity with cell types, and the authors observe that a lower dimensionality feature vector with only optic lobe intrinsic types (covering 80% of the synapses) is sufficient for defining cell types. The suggested cell types are then validated in several statistical ways. Ultimately, the combined use of cell anatomy and synaptic connectivity enables the assignment of in total 37.500 intrinsic neurons of a single right optic lobe to 224 intrinsic cell types. This strong reduction in complexity, together with a type-to-type connection matrix is extremely helpful for research on the neuronal circuits of vision in *Drosophila* and fosters a comparative perspective on other visual systems, in particular the vertebrate visual system.

Any more detailed description of the findings of the study would quickly turn into a 'discussion among experts on the neuronal circuits of fly vision'. The provided data will certainly be of strong influence for future studies on visually guided behavior (I am not fully convinced that perception of visual features like color, motion, and form can indeed be accessed in the fly, as mentioned by the authors) and research on the underlying neuronal computations. This data will definitely accelerate research on the identification and characterization of visual circuits. I expect this study to be highly relevant for future research on fly vision.

Data, validity, and methodology

To my knowledge, the underlying data set and analysis represents the current state of the art. Future data sets might enable higher optical resolution and include electric synapses (innexin junctions), which would represent a dramatic further advance. However, at present, data on electric connectivity is only available on much smaller datasets.

The methodology for data analysis seems to be carefully chosen and validated. However, frankly speaking, I was never directly involved in the analysis of this data or development of algorithms. This reduced the weight of my judgement. From the perspective of an outsider of connectome data analysis, the methods seemingly represent the culmination of a long thought process within the community of researchers pushing connectomics. Sebastian Seung and his team are leading scientists in this field. At my level of understanding, the reporting, methodology, and treatment of data is robust, reliable, and support the conclusions. An in-depth analysis / review of the methodology and appropriateness of statistical testing I have to leave to experts in the field.

General

The manuscript is well written and data is presented very well. I cannot identify any inflammatory material. Abstract and introduction are easily accessible. The manuscript contains a very large amount of supplementary data, in particular the 'ocean of online data' that is impossible to review and that is where much of the music plays.

The manuscript clearly is of high interest to readers with interest in fly vision and represents a conceptual step forward, given that photoreceptors become included.

Suggestions to strengthen the study

The about 6000 photoreceptors of each ommatidium are pivotal for vision and knowledge on the feedforward and feedback connectivity of photoreceptors and their synaptic partner neurons is pivotal for the rigorous functional analysis of most visual circuits. From my perspective, it is a major weakness of the present study that photoreceptors have been actively excluded from the present dataset and that the authors plan to submit this information separately.

The lack of data on photoreceptors and their connectivity to identified postsynaptic cell types in the lamina and medulla contradicts the central promise made by the authors to provide a part list on the fly visual system. At present, the submitted work is an 'Incomplete neuronal "parts list" and wiring diagram for the visual system.' that might not fully meet the high standard of publication in Nature.

However, this could be changed by including photoreceptors and connectivity into the study, which would result in a state-of-the-art part list and wiring diagram of the fly visual system of sufficient general interest to the broad readership of Nature.

We agree that the photoreceptors are essential. The revised manuscript has been upgraded to v783, which includes 3436 R1-6, 651 R7, and 640 R8 in the right optic lobe.

See manuscript page 24, Future Releases:

>>One major change will be the addition of the remaining photoreceptors, which by now have been completely proofread and typed, and are already

accessible via CAVEclient (Dorckenwald, Matsliah, et al. 2023; Dorckenwald, Schneider-Mizell, et al. 2023)<<

Minor

Although I want to leave it to the authors how this could be changed: large parts of the results section are lengthy and have the character of an encyclopedia written for experts. This also results in large volume of text. A brief example:

>>Of the types with partial coverage that contain multiple cells, Sm←CB0566←MeTu2, Sm←MC65⇒TmY5a, Sm←MeTu3⇒MeTu2, Sm←MeTu4←MeTu1, Sm←MeTu4⇒CB0566, Sm←Mi15⇒MLt5, Sm←Mi9⇒Tm32, and Sm⇒TmY3 are dorsal-only . Ventral-only types include Sm←Mi9⇒CB0165, Sm←TmY5a⇒Tm7, and Sm←Y12⇒TmY10. Notably, Sm←Mi9⇒Tm32 resembles similar stratification to Mi3 in upper and lower boundaries of the M7 layer. However, Sm←Mi9⇒Tm32 exhibits an additional stratification within the M4 layer...<<

Could some of this information / similar paragraphs be partially moved to the supplement? This would also enable more visible highlighting of results with broader relevance.

The narrative in the main text is now organized around the motion, object, color etc subsystems that are defined by cutting the dendrogram of Fig. 2b into clusters. This organization by function should be more interesting to a broader audience.

The narrative in the old text was organized by neuropil-defined family, and has been moved to the Methods and Supplementary Information.

Can information be added on expected changes for an optic lobe of a male fly?

HHMI Janelia has a parallel effort to reconstruct an optic lobe of an adult male fly. It is surely better to wait for their paper than to speculate.

In same context:

Page 20, end of paragraph 4

>>(Note that our wiring diagram is derived from a female brain, but is likely to be the same or similar for a male brain, as there are few sexual dimorphisms within the optic lobe.) <<

This statement deserves a citation.

We have removed this statement. It seems unnecessary and any citation of past work would only apply to a subset of types at best. For the reviewer's benefit, we can say here that we were thinking of the Fru and Dsx neurons (Yu et al. 2010; Rideout et al. 2010). Some of these neurons are visual (LC14), and a companion manuscript is characterizing their sexual dimorphism.

page 4, paragraph 3 >>To refine these coarse class distinctions,...<<
Do the authors really know whether axons are indeed axons (as defined on page 12)? Or rather mixed processes?...
Wouldn't it be better to call them main neurites or processes, unless there is clear evidence for axon or dendrite? Didi I miss that there is clear evidence in all cases where axon are mentioned?

We have added text to the Methods about the axon-dendrite distinction. The reviewer is correct that there are subtleties to the definition, and another manuscript about the axon-dendrite distinction is in preparation. To some extent, axon can be distinguished from dendrite by morphology. The axon has swellings that are synaptic boutons, while the dendrite has numerous fine twigs that are postsynaptic. These morphological differences are presumably what (Fischbach and Dittrich 1989) could see in their light microscope. However, these authors admit to some uncertainty in their axon-dendrite judgments. We use electron microscopic images, which allow more definitive judgments based on the proportion of presynapses and postsynapses. The reviewer is correct, however, that axon-dendrite is not a black-and white distinction in the fly brain. An arbor can have a mixture of presynapses and postsynapses. Sometimes the axon-dendrite distinction is relative within a neuron, meaning that the axon is defined as having a much higher fraction of presynapses than the dendrite, and this may not be fraction that is high in an absolute sense. Codex provides two aids to make the distinction. First, every neuron's "Cell Info" page shows presynapse and postsynapse numbers by neuropil. Second, the 3D view of a neuron now shows its presynapses and postsynapses as blue and yellow dots superimposed on the 3D reconstruction of the cell.

page 8, Type-to-type connectivity
Dot size. Dot size is used to display the strength of synaptic connectivity, the encoding of numbers in size is non-linear. Although difficult to display, in its present form the display is not really helpful as there is little variation between dots that are all very small....
Could this be solved by in addition providing the number of synapses in a further table / tables in the supplement? I might have missed that such tables are available online?

Thanks for the suggestion. The numbers are now provided in the accompanying github repository: https://github.com/murthylab/flywire-visual-neuron-types/blob/main/data/783/right/type_to_type_synapse_counts.csv. The numbers are also provided in a spreadsheet as Supplementary Information.

page 8, Type-to-type connectivity, paragraph 2
>>The threshold depends on the scientific question to be addressed.<<
Does this mean that analysis of circuitries of motion vision involves other thresholds than for instance detection of form or color? Can the authors please explain this cryptic statement?

We apologize for being cryptic. In our original manuscript, we did not expand on this statement because it is impossible to anticipate every kind of scientific question that will be addressed using this data. The best we can do is explain retrospectively how and why we have varied the threshold in our studies so far. In the central brain, most cell types have cardinality 2 (cell and its mirror twin in the opposite hemisphere, Fig. S1e). In the hemibrain, the cardinality is typically reduced to one. Therefore if you would like to know whether there is a connection between cell type A and cell type B, you must decide based on only two or three examples of the ordered pair (A, B) in all the connectomic data that is so far available. This is a small sample size, and the hemibrain paper imposed a threshold to reduce false positives that could arise because of noisy synapse detection or biological variability.

But in the optic lobe, there are often many examples of the ordered pair (A, B), because so many cell types have high cardinality. Therefore, if a connection is consistently found from type A to type B, one can have more confidence even if the average number of synapses in the connection is not so high. That is why we set the threshold lower in the optic lobe than in the central brain. In particular, we have found that certain inhibitory types consistently make connections that involve relatively few synapses, and these connections seem real.

The Methods section of the revised manuscript explains these issues.

page 8, Type-to-type connectivity, last paragraph
How many false positives are expected?

False positive rates for individual synapses are quantified in the flagship paper. This is now noted in the Methods.

page 10, paragraph 2 >>Almost 400 Dm8 cells have been proofread and identified in v630...<<

Why does this para on Dm cells appear here? I suggest moving the para to page 13/14 where results on Dm cell are presented.

Done

page 10, paragraph 4 <<But overall Tm8a and Tm8b differ markedly in connectivity. Li \Rightarrow LC12 \Rightarrow Tm8a is the strongest output of Tm8a, but weak or nonexistent as an output of Tm8b...<<

We have dropped the discriminative names, though we still provide the 2D discriminators in Data S3. To explain the now defunct discriminative name, Tm8a is indeed an output of Li \Rightarrow LC12 \Rightarrow Tm8a. It turns out that Tm8a is also one of the top inputs to Li \Rightarrow LC12 \Rightarrow Tm8a, but that fact is not indicated by the name. Adding all strong inputs and outputs to a discriminative name would make it impractically long.

The devised discriminative names are very helpful in general. However, I cannot understand the name in the example above and few other places. On page 3 of the manuscript: For example, Li \Leftarrow TmY4 \Rightarrow LT79 is the Lobula intrinsic (Li) neuron that receives input from TmY4 and sends output to LT79
How can this logic be applied to the example above such that output from Tm8a to Li.....becomes visible? Did I miss something?

Discriminative names were already unwieldy, and would become even more unwieldy if they incorporated further information about inputs and outputs. In the revised manuscript, we have replaced the discriminative names with short names. Discriminating features are still provided for reference in Data S3.

page 14, paragraph 2

In the section on Dm8p and Dm8y the work of Li et al., 2021 should be cited as they were the first and only ones to reveal that p and y types of Dm8 are indeed functionally different.

Thanks for pointing us to that reference, which we have now cited.

page 16, LPi Lobula plate intrinsic, paragraph 2

>>H2 is a lobula plate tangential cell (LPTC) that prefers back-to-front motion (Wei et al. 2020).<<

This citation is not suitable as Wei et al did not perform any experiments on the directional tuning / receptive field layout of LPTCs.

The sentence was eliminated in the course of our effort to remove “encyclopedia”-like text.

page 19, Discussion

>>Early advocates of connectomics predicted that deriving cell types from connectivity would be transformative (H. Sebastian Seung 2012). Now that the connectomic era has finally dawned, we can begin to see how well this prediction holds up.<<

This notion does not enable any new insight or perspective on the presented data and should therefore become deleted.

Deleted.

page 20, Discussion

>>Color preference in Drosophila varies over the day (Schnaitmann et al. 2018).....<<

This citation is wrong / does not support the given statement as Schnaitmann et al. 2018 did not at all investigated color preference behavior.

A study suggesting changing color preference during the course of the day has been published by Lazopulo et al. in Oct. 2019 in Nature. 2019

Oct;574(7776):108-111. doi: 10.1038/s41586-019-1571-y. Epub 2019 Sep 18.

The paragraph has been deleted in the revised manuscript.

page 21, Discussion

>>Single cell transcriptomics is providing detailed information about the molecules in fly visual neurons (Kurmangaliyev et al. 2020; Özel et al. 2021; Konstantinides et al. 2022).<<

Do the authors really mean 'single cell transcriptomics' or rather 'single cell type transcriptomics'? Based on what are cells / cell types defined in these studies?

As far as we understand, the data are transcriptomes from single cells, which can be clustered to define cell types. But the reviewer is correct that it is nontrivial to establish correspondences between transcriptomic clusters and morphological or molecular-morphological cell types. Transgenic lines are used for this purpose, and that might be called single cell type transcriptomics.

Discussion:

I might have missed this: has the complete lack of information on electric coupling be discussed at all? Including electric coupling might fundamentally change current functional interpretations of chemical connectivity (...mind the gap (junctions)!).

This issue is discussed in the flagship paper.

Reviewer Reports on the First Revision:

Referees' comments:

Referee #1 (Remarks to the Author):

I am satisfied by the responses to the questions raised. The addition of the comparison with HHMI's seven-column medulla data is reassuring.

However, the manuscript has been extensively reorganized around the functional subsystems. The authors assigned functions for individual clusters by extrapolating from few cells of known functions in the clusters. This approach has a number of potential problems, many of which were not acknowledged clearly. First, the thresholding of 0.91 for clustering seems arbitrary. Second, spatial arrangement and local patterns of connectivity were not considered. Third, a small number of cells of known functions were used to argue specific functions for a much large number of cells. As such, there are already a number of examples that are somewhat inconsistent with known functions of specific cells. For example, Tm9 and Tm1/2/4 are main inputs to T5 but they are separated into the motion and the OFF subsystems. DmDRA1 and DmDRA2 (known for polarization) are included in the clusters 5 (polarization) and 6 (color), respectively. The assignment of the color subsystem is perhaps the most problematic. Over 100 cells were included using only a handful of cells of known functions (that appear to be in a separate cluster of their own). It is likely that the cluster1 would be broken into several clusters with a different threshold and lumping all of them into the color subsystem seems premature. Thus, it is impressive (but perhaps not too surprising) that a good number of cells of a single cluster share similar functions. But ascribing that function to the entire cluster would be a stretch.

There are some inconsistent color codes (Figures 2-7). Also I am puzzled by regarding R7/R8/Dm8/Dm9 as "late color" (Figure 3).

Referee #2 (Remarks to the Author):

This revised manuscript categorized and clustered 38,500 neurons in the *Drosophila* vision system based on their connectivity, and described the wiring diagram of different functional subsystems (ON/OFF and luminance, motion, and color). Clustering is based on the input and output connectivity of neurons, is not biased by the neurotransmitter identity of neurons, and can be used to speculate on their function in the visual system. The switch in narrative (from describing individual families to discussing clusters/families/cell types based on their functions) makes the story more amusing to read and interesting to a more general audience. The revised manuscript has been upgraded to v783 of the FlyWire database of proofread neurons and includes new cell types. Although the revised manuscript still lacks cross-validation with light microscopy, it provides a great foundation for future functional studies.

Remarks to the author

Overall, the flow of the paper has been significantly improved. Describing the wiring diagram within subsystems puts more sense on cell typing and clustering, and is interesting to experimentalists who study the physiology and function of the circuits. This also validates the clustering methods and results, which can be used to make speculations or hypotheses regarding the functions of poorly studied cell types. It is worth mentioning that a more concise cell-type naming is used in this revision to improve readability.

I appreciate the fact that the authors re-ran the algorithm using the newer v783 of the database. In addition, the authors linked it to the flagship paper for an introduction to the FlyWire interface wherever possible. It is critical that the information is up-to-date, accessible, and easy to use.

The authors stated the challenges of using light microscopy to validate reconstruction with split-Gal4 lines targeting new cell types. It is regrettable not to see at least a few examples of new cell types in another brain; however, I understand that this is a challenging task. This study can be used as a great reference to explore or hypothesize about the functions of unknown neurons.

Minor points:

The wiring diagrams in some figures (such as in Figure 3) have poor resolution. Text and arrow/circle tips are not always easy to read when zoomed-in. Is it possible to replace these graphs with a vector picture so that they will not lose resolution if the reader chooses to inspect the details?

In addition, does the line width/thickness indicate the number of synaptic connections in these wiring diagrams?

Cluster 13 (a small cluster) is not featured in the text where different subsystems are discussed. Are there any speculations regarding their function?

Remarks on code availability

The algorithm is well described and makes sense to experimental neuroscientists; however, there is no available source code. The source data for the annotations of the cell types were provided for downloading.

Referee #3 (Remarks to the Author):

Reviewer 3 - response to the authors / to rebuttal letter and revised manuscript

The authors addressed the major concerns and request. The manuscript is much improved. There is a list of remaining questions and required changes. Much of it is linked to missing clarity, missing information in the legends and probably a lack of understanding on my side. However, this might include the possibility that other readers will have problems too....

Major:

- Narrative & names: The authors have successfully reframed the narrative. Cell types are now described according to their proposed functional affiliation with motion-, color-, object-,.... detection 'clusters/circuits'. The previous arrangement into neuropil communities is still available outside of the main text and same for the previously used 'discriminative names'.

Beginning with the end of page 2 (first submission), the manuscript has been changed substantially. Single-cell anatomy has been replaced by class, family and type.....following the new narrative. Information on the techniques used has seemingly been removed?

The provided resolution / quality of the figures is generally very poor.

- New Fig.1 fits the new narrative.

- Old Fig.2 has been replaced by the new Figure 2 to fit the new narrative.

- Old Fig 3 removed / moved to supplement. I am somehow missing old Fig3 (wiring diagram) as it nicely summarized the complexity of the matrix. However, the old info is still available elsewhere. Don't know whether this change was required.

- New Fig.4: would it be possible to use a 'figure-to-figure' consistent color code for functional groups? Only TmY11 and Mi4 are depicted in purple: does this mean that only these two cell types are members of the motion subsystem (see new Fig.3) when it comes to the quantification of main in and output?

Much of the figure is not / hardly accessible.

What is the importance / meaning of line-color and thickness (see other figures as well)?

Dm15 and 17 receive 'hidden arrows' - what does that mean?

Li26 receives 'curved arrows'?

- New Fig.5. motion subsystem. In Fig 3, the motion subsystem is depicted in purple. In this new Figure none of the cells is purple. Why?

- New Fig 6: same type of questions.

-

- New Fig. 7: When comparing the depicted connections with the strength/ number of synapses (GitHub TableS1) I found discrepancies. Am I missing something? To me it is very irritating that table and figure might not match well. For instance: Sm20 is the strongest input to Dm8a with 6500 synapses, but Sm20 is not at all included in the figure? Am I wrong? Is the figure wrong? Is the table right?

It appears I am lost here.

And: also here much of the legend is missing....

- Upgrade & photoreceptors: The authors implemented an upgrade to v783 of the proofread reconstruction of FlyWire/FAFB which lead to an increase in the number of proofread cells from 37,000 to 38,500, and from 224 intrinsic cell types to 226.

The authors now include information on the connectivity of 4700 photoreceptor cells (outer R1-R6 and inner R7, R8) to their postsynaptic partner cells in the lamina and medulla.

I thank the authors for their readiness to add this important information.

- n synapses. The study now includes a 'type-to-type' connectivity matrix (Fig.S4) where dot size corresponds to the number of synapses AND the number of synapses is now provided in a separate table S1.

(from the rebuttal letter: >> Thanks for the suggestion. The numbers are now provided in the accompanying github repository: https://github.com/murthylab/flywire-visual-neuron-types/blob/main/data/783/right/type_to_type_synapse_counts.csv.

The numbers are also provided in a spreadsheet as Supplementary Information.<<).

Question / comment to the authors:

Matrix and table together represent extremely valuable information for scientists working on the function of circuits. I still find it hard to pull out precise information on the number of synapses between identified pairs of cells using Table S1. Is it just my lack of understanding?

The github-repository is fantastic.

- I realized only now that all information on LPTCs is missing. Why have these important cells been excluded? Because many / some of them appear only once? Because some but not all of them project out of the optic lobe? If so, are they / some of them (but certainly not all) hidden in the group of so called 'boundary neurons'?

- abstract: >>Connectivity with "boundary types" that straddle the optic lobe and central brain is also quantified.<< but not LPTCs?

- p2 >>We additionally provide rules of connectivity between intrinsic types and types of boundary neurons, defined as those that connect the optic lobe with regions in the central brain.<< well, LPTC should be included then?

Again, I am probably missing something. If not: the authors should openly declare what cell types are excluded and why. Any selection should be justified by technical reasons.

In the current manuscript, LPTC are mentioned twice – meaning you are (of course!) aware of these cells:

- p43 – programmatic tools: >> Another tool created from LPTCs (e.g. HS, VS, H1) aided definition of layers in the lobula plate.<<

- p53: >>The tangential types connect neuropils within one optic lobe and do not leave the optic lobe. Our usage of the term “tangential” focuses on axonal orientation only. It should not be misunderstood to imply a wide-field neuron that projects out of the optic lobe, which is the case for the well-known lobula plate tangential cells (LPTCs) or lobula tangential cells (LTs). The term “tangential” presupposes that we can identify an axonal arbor for the cell (see Axon versus Dendrite).<<

Some of the authors are also involved in an accompanying paper on bioRxiv doi:

<https://doi.org/10.1101/2024.04.16.589741> on the optic lobe of a male fly. Here, LPTCS are captured.

Minor:

- The now included comparison with the 7-column study is much appreciated! Thanks for filling the gap.

- It would be fantastic to include information on p/y columns (reviewer 1), but I understand that the community has to wait until further data analysis has been done.

Please apologize the many questions. I congratulate the authors to the revised manuscript and hope the authors can clarify the above questions.

Best,

Dierk Reiff

Author Rebuttals to First Revision:

Response to reviews of ‘Neuronal “parts list” and wiring diagram for a visual system’

We are grateful to the Referees for taking the time to review such a lengthy manuscript, and provide thoughtful suggestions. Thanks to the feedback, we have been able to improve the manuscript further. In addition to a clean PDF, we are forwarding an MS Word document with Track Changes on.

Summary of major changes

- Caveats about clusters, subsystems, and their functional interpretations are more prominent.
- The wiring diagrams (Figs. 3-7) are now more complex, in response to the requests of Referee 3. As requested, many more boundary types are now displayed. Previously many cell types were suppressed by various thresholds to make the diagrams simpler.
- Wiring diagrams have been “upgraded” from 90 dpi to 150 dpi, and vector graphics will be provided for the final publication.
- Wiring diagrams are covered in the Methods.
- The colors of cell types are now consistent across the wiring diagrams and the dendrogram (Fig. 2).
- Symbols in the wiring diagrams have been simplified and made consistent, and are explained in the legends.
- A new Fig. S10 provides alternative colorings of the dendrogram indicating flat clusterings that are refinements of the flat clustering shown in Fig. 2c. This responds to the criticism of Referee 1 that the threshold in Fig. 2c seems arbitrary.
- Connectivity between clusters has been added to Fig. 2.

Referee #1 (Remarks to the Author):

I am satisfied by the responses to the questions raised. The addition of the comparison with HHMI’s seven-column medulla data is reassuring.

However, the manuscript has been extensively reorganized around the functional subsystems. The authors assigned functions for individual clusters by extrapolating from few cells of known functions in the clusters. This approach has a number of potential problems, many of which were not acknowledged clearly.

The Referee’s point is well-taken: caveats about the clustering should have been more prominent. They were in the Methods of the previous submission. Now the main text has been revised to be more cautious.

First, the thresholding of 0.91 for clustering seems arbitrary.

We have now clarified that you are free to choose whatever threshold you want, which is an important advantage of a hierarchical clustering. As examples, the new Fig. S10 provides alternative colorings of the same dendrogram with 26 and 39 clusters, respectively. Different choices of the threshold will be appropriate for different biological questions. You can effectively adjust the threshold by eye if you inspect the dendrogram.

Second, spatial arrangement and local patterns of connectivity were not considered.

The Referee is perspicacious to notice that the present work ignores the spatial organization of connectivity. We were surprised that we could get so far without it, and indeed that could be regarded as a finding of the paper. It turns out that working in the high dimensional feature space of type connectivity is very powerful.

Of course, adding back the information about spatial organization should make structural analysis even more powerful. But we think that spatial information will often elaborate on the concepts presented in this paper, rather than overturning them. For example, a companion paper uses spatial analysis to predict that the six cell types of Cluster2 are local orientation detectors analogous to neurons in primary visual cortex (Seung 2023). This analysis ascribes a particular function to each of the six cell types. But the refined analysis does not contradict the coarser idea of lumping these types together in a single subsystem for form vision. In fact, the refined analysis elaborates on the coarser idea.

Third, a small number of cells of known functions were used to argue specific functions for a much large number of cells. As such, there are already a number of examples that are somewhat inconsistent with known functions of specific cells. For example, Tm9 and Tm1/2/4 are main inputs to T5 but they are separated into the motion and the OFF subsystems.

The idea that Tm1/2/4 are dedicated to motion could be regarded as an accident of history. The present work reveals that their targets are diverse (Figs. 4, 5, 6). Other strong targets include T2, T2a, and T3, which detect small objects (Fig. 6) (Keleş et al. 2020). In addition, Tm1 is a prominent input to the putative form subsystem (Cluster2), as detailed in a companion paper (Seung 2023). T5 is merely the target that was discovered first. Therefore it makes sense for Tm1/2/4 to be regarded as part of a general purpose OFF channel rather than dedicated to motion only.

Tm9 is a different story. Relative to Tm1/2/4, Tm9 dedicates a larger fraction of its output synapses to T4/T5. Tm9 does not target the putative object subsystem. Tm9 receives strong input from L3. Based on these properties, it seems reasonable to separate Tm9 from Tm1/2/4 and assign it to the motion subsystem.

DmDRA1 and DmDRA2 (known for polarization) are included in the clusters 5 (polarization) and 6 (color), respectively.

The previous submission already noted the issue with DmDRA2: "DmDRA2 should also be part of the polarization subsystem, but is currently assigned to the color-related Cluster6. That may be because we do not currently distinguish R7 and R8 in the dorsal rim area from other R7 and R8."

The assignment of the color subsystem is perhaps the most problematic. Over 100 cells were included using only a handful of cells of known functions (that appear to be in a separate cluster of their own). It is likely that the cluster1 would be broken into several clusters with a different threshold and lumping all of them into the color subsystem seems premature.

Cluster1 (and other clusters) can certainly be divided more finely by lowering the threshold. The new Fig. S10 provides examples of finer subdivisions. The caveats in Main and Discussion have been expanded to mention this possibility.

At the same time, a finer description does not necessarily contradict a coarser description. One example of this was the hypothetical form subsystem mentioned above. Another example is the motion subsystem, which Fig. 2c separates into Cluster12 (mostly T4/T5) and Cluster11 (mostly LPi). The interneurons in Cluster11 are thought to mediate motion opponency (Ammer et al. 2023) or normalization (Seung 2024) for the motion-detecting neurons in Cluster12 (Discussion). But these fine distinctions do not contradict the coarser description that lumps both clusters into a motion subsystem. We can go even finer by subdividing the motion subsystem into six clusters (Fig. S10b). CT1/Li14/Tm9 splits apart from T4/T5. Each of the other four clusters contain LPi types that are specialized for one of the four motion directions. This even finer clustering makes functional sense, but it does not contradict the coarser concept of lumping all six clusters into a motion subsystem.

Similarly, finer divisions of the hypothetical color subsystem (Fig. S10) may not contradict the coarse idea of a color subsystem. Like the Referee, we were also surprised by the large number of cell types in the hypothetical color subsystem, as noted in the Discussion of the previous submission. But this large number is based on data and algorithms; there is no reason to regard it as less plausible than our prior expectations. Some insects are known to have sophisticated color vision capabilities such as color constancy (Song and Lee 2018), and *Drosophila* might also have such capabilities. Color constancy is known to require complex computations, so a large number of cell types might be necessary. So we hope that the Referee will not be opposed to presenting the hypothetical color subsystem, now that many caveats have been added.

Thus, it is impressive (but perhaps not too surprising) that a good number of cells of a single cluster share similar functions. But ascribing that function to the entire cluster would be a stretch.

The clusters should certainly be regarded as hypotheses about functional groupings, not the gospel truth. We have added caveats to that effect to Main and Discussion. The word "hypothetical" has been inserted into several Figure titles.

There are some inconsistent color codes (Figures 2-7).

We apologize for the inconsistencies. The dendrogram and all wiring diagrams now have consistent color coding of cell types.

Also I am puzzled by regarding R7/R8/Dm8/Dm9 as "late color" (Figure 3).

We apologize that some labels were inadvertently switched. It does not matter now because we have replaced such labels by "Cluster1" through "Cluster13" in the spirit of being more cautious about functional interpretations.

Referee #2 (Remarks to the Author):

This revised manuscript categorized and clustered 38,500 neurons in the *Drosophila* vision system based on their connectivity, and described the wiring diagram of different functional subsystems (ON/OFF and luminance, motion, and color). Clustering is based on the input and output connectivity of neurons, is not biased by the neurotransmitter identity of neurons, and can be used to speculate on their function in the visual system. The switch in narrative (from describing individual families to discussing clusters/families/cell types based on their functions) makes the story more amusing to read and interesting to a more general audience. The revised manuscript has been upgraded to v783 of the FlyWire database of proofread neurons and includes new cell types. Although the revised manuscript still lacks cross-validation with light microscopy, it provides a great foundation for future functional studies.

Remarks to the author

Overall, the flow of the paper has been significantly improved. Describing the wiring diagram within subsystems puts more sense on cell typing and clustering, and is interesting to experimentalists who study the physiology and function of the circuits. This also validates the clustering methods and results, which can be used to make speculations or hypotheses regarding the functions of poorly studied cell types. It is worth mentioning that a more concise cell-type naming is used in this revision to improve readability. I appreciate the fact that the authors re-ran the algorithm using the newer v783 of the database. In addition, the authors linked it to the flagship paper for an introduction to the FlyWire interface wherever possible. It is critical that the information is up-to-date, accessible, and easy to use.

The authors stated the challenges of using light microscopy to validate reconstruction with split-Gal4 lines targeting new cell types. It is regrettable not to see at least a few examples of new cell types in another brain; however, I understand that this is a challenging task. This study can be used as a great reference to explore or hypothesize about the functions of unknown neurons.

Minor points:

The wiring diagrams in some figures (such as in Figure 3) have poor resolution. Text and arrow/circle tips are not always easy to read when zoomed-in. Is it possible to replace these graphs with a vector picture so

that they will not lose resolution if the reader chooses to inspect the details?

The wiring diagrams have been upgraded to 150 dpi for the time being, and vector graphics will be provided for the published version.

In addition, does the line width/thickness indicate the number of synaptic connections in these wiring diagrams?

Good guess. This is now noted in the new Figure 3 legend and Methods.

Cluster 13 (a small cluster) is not featured in the text where different subsystems are discussed. Are there any speculations regarding their function?

Thanks for catching this. Cluster13 consists of R1-6, Lai, and Lawf1, and is now described briefly in the text. It provides input to the ON, OFF, and luminance channels. Understanding the functions of the lateral and top-down interactions mediated by Lai and Lawf1 will require deeper analysis, and is left for future work.

Remarks on code availability

The algorithm is well described and makes sense to experimental neuroscientists; however, there is no available source code. The source data for the annotations of the cell types were provided for downloading.

Some of the code and data is already available at <https://github.com/murthylab/codex> and <https://github.com/murthylab/flywire-visual-neuron-types>.

The remaining code is being refactored and will be made available on or before publication.

Referee #3 (Remarks to the Author):

Reviewer 3 - response to the authors / to rebuttal letter and revised manuscript

The authors addressed the major concerns and request. The manuscript is much improved. There is a list of remaining questions and required changes. Much of it is linked to missing clarity, missing information in the legends and probably a lack of understanding on my side. However, this might include the possibility that other readers will have problems too...

Major:

- Narrative & names: The authors have successfully reframed the narrative. Cell types are now described according to their proposed functional affiliation with motion-, color-, object-,... detection 'clusters/circuits'. The previous arrangement into neuropil communities is still available outside of the main text and same for the previously used 'discriminative names'.

Beginning with the end of page 2 (first submission), the manuscript has been changed substantially. Single-cell anatomy has been replaced by class, family and type...following the new narrative. Information on the techniques used has seemingly been removed?

The technical information is important, and has been preserved by moving it to the Methods.

The provided resolution / quality of the figures is generally very poor.

This has been fixed by upgrading from 90dpi to 150dpi. Vector graphics will be made available in the final published version.

- New Fig.1 fits the new narrative.
- Old Fig.2 has been replaced by the new Figure 2 to fit the new narrative.
- Old Fig 3 removed / moved to supplement. I am somehow missing old Fig3 (wiring diagram) as it nicely summarized the complexity of the matrix. However, the old info is still available elsewhere. Don't know whether this change was required.

We liked the big matrix too, but ended up moving it to Figure S4 because there are so many main figures.

- New Fig.4: would it be possible to use a 'figure-to-figure' consistent color code for functional groups?

Colors of cell types in the dendrogram and wiring diagrams should be consistent now.

Only TmY11 and Mi4 are depicted in purple: does this mean that only these two cell types are members of the motion subsystem (see new Fig.3) when it comes to the quantification of main in and output?

In our hierarchical clustering, Mi4 is not assigned to the motion subsystem. (The purple color was just a coincidence in our inconsistent coloring scheme.) T4 is a target of Mi4, but not an especially strong one.

Much of the figure is not / hardly accessible.

What is the importance / meaning of line-color and thickness (see other figures as well)?

Line thickness is proportional to number of synapses (new Fig. 3 legend)..

Dm15 and 17 receive 'hidden arrows' - what does that mean?

Hidden arrows were an artifact of obstructions / overlaps in the layout of the wiring diagram. In the new versions of these diagrams we made best effort to minimize such overlaps, but it's not always possible since these networks are far from being planar. For additional clarification we added a section in Methods about wiring diagrams.

Li26 receives 'curved arrows'?

Curved arrowheads have been eliminated.

- New Fig.5. motion subsystem. In Fig 3, the motion subsystem is depicted in purple. In this new Figure none of the cells is purple. Why?

Colors should be consistent now.

- New Fig 6: same type of questions.

-

- New Fig. 7: When comparing the depicted connections with the strength/ number of synapses (GitHub TableS1) I found discrepancies. Am I missing something? To me it is very irritating that table and figure might not match well. For instance: Sm20 is the strongest input to Dm8a with 6500 synapses, but Sm20 is not at all included in the figure? Am I wrong? Is the figure wrong? Is the table right?

To simplify Fig. 7, we suppressed cell types containing less than a threshold number of cells. The new Fig. 7 has no such threshold, and is more complex and complete now. Sm20 is now visible near the right border.

It appears I am lost here.

And: also here much of the legend is missing...

- Upgrade & photoreceptors: The authors implemented an upgrade to v783 of the proofread reconstruction of FlyWire/FAFB which lead to an increase in the number of proofread cells from 37,000 to 38,500, and from 224 intrinsic cell types to 226.

The authors now include information on the connectivity of 4700 photoreceptor cells (outer R1-R6 and inner R7, R8) to their postsynaptic partner cells in the lamina and medulla.

I thank the authors for their readiness to add this important information.

- n synapses. The study now includes a 'type-to-type' connectivity matrix (Fig.S4) where dot size corresponds to the number of synapses AND the number of synapses is now provided in a separate table S1.

(from the rebuttal letter: >> Thanks for the suggestion. The numbers are now provided in the accompanying github repository:

https://github.com/murthylab/flywire-visual-neuron-types/blob/main/data/783/right/type_to_type_synapse_counts.csv.

The numbers are also provided in a spreadsheet as Supplementary Information.<<).

Question / comment to the authors:

Matrix and table together represent extremely valuable information for scientists working on the function of circuits. I still find it hard to pull out precise information on the number of synapses between identified pairs of cells using Table S1. Is it just my lack of understanding? The github-repository is fantastic.

Table S1 contains properties of types, not connections. Connections between types are captured in Fig. S4 and the github repo. Connections between individual cells can be examined in the FlyWire Codex (codex.flywire.ai) or by downloading data from <https://codex.flywire.ai/api/download>

- I realized only now that all information on LPTCs is missing. Why have these important cells been excluded? Because many / some of them appear only once? Because some but not all of them project out of the optic lobe? If so, are they / some of them (but certainly not all) hidden in the group of so called 'boundary neurons'?

They appeared in Data S4 of the previous submission, but were largely suppressed by various threshold criteria in the wiring diagrams. The new wiring diagrams have now been allowed to be highly complex, and contain many boundary types including LPTCs.

- abstract: >>Connectivity with "boundary types" that straddle the optic lobe and central brain is also quantified.<< but not LPTCs?

We apologize for "burying" this information. In the Class, Family, and Type section, we wrote that "Connections between intrinsic and boundary types are detailed in Data S4," but we realize now that this sentence is easy to miss.

To make the information more prominent, the wiring diagrams in the main figures have been updated, and now include many boundary types. This comes at the cost of greater complexity, but we realize now that many readers, like this Referee, will enjoy the complexity.

- p2 >>We additionally provide rules of connectivity between intrinsic types and types of boundary neurons, defined as those that connect the optic lobe with regions in the central brain.<< well, LPTC should be included then?

Many LPTCs now appear in the new Figure 5.

Again, I am probably missing something. If not: the authors should openly declare what cell types are excluded and why. Any selection should be justified by technical reasons.

A new subsection about the wiring diagram has been added to the Methods, consolidating information scattered throughout the Figure legends. We hope that this will clarify the situation.

In the current manuscript, LPTC are mentioned twice - meaning you are (of course!) aware of these cells:

- p43 - programmatic tools: >> Another tool created from LPTCs (e.g. HS, VS, H1) aided definition of layers in the lobula plate.<<

- p53: >>The tangential types connect neuropils within one optic lobe and do not leave the optic lobe. Our usage of the term "tangential" focuses on axonal orientation only. It should not be misunderstood to imply a wide-field neuron that projects out of the optic lobe, which is the case for the well-known lobula plate tangential cells (LPTCs) or lobula tangential cells (LTs). The term "tangential" presupposes that we can identify an axonal arbor for the cell (see Axon versus Dendrite).<<

Some of the authors are also involved in an accompanying paper on bioRxiv doi: <https://doi.org/10.1101/2024.04.16.589741> on the optic lobe of a male fly. Here, LPTCS are captured.

All of the information in our resource can be interactively accessed through the FlyWire Codex and programmatically accessed using Codex downloads. But there is no way to include all of the information in the Figures of the paper. Therefore we have often struggled to decide what information to show and what information to suppress. The Referee's frustration is valuable to hear, and now we know that we suppressed too much information before. We hope that the new expanded wiring diagrams will be satisfactory. At the same time, even these diagrams leave out so much information. The Referee is invited to look at Data S4 for more information, and even better to consult the Codex.

Minor:

- The now included comparison with the 7-column study is much appreciated! Thanks for filling the gap.

- It would be fantastic to include information on p/y columns (reviewer 1), but I understand that the community has to wait until further data analysis has been done.

Please apologize the many questions. I congratulate the authors to the revised manuscript and hope the authors can clarify the above questions.

Best,

Dierk Reiff

References

- Ammer, Georg, Etienne Serbe-Kamp, Alex S. Mauss, Florian G. Richter, Sandra Fendl, and Alexander Borst. 2023. "Multilevel Visual Motion Opponency in *Drosophila*." *Nature Neuroscience* 26 (11): 1894–1905.
- Keleş, Mehmet F., Ben J. Hardcastle, Carola Städele, Qi Xiao, and Mark A. Frye. 2020. "Inhibitory Interactions and Columnar Inputs to an Object Motion Detector in *Drosophila*." *Cell Reports* 30 (7): 2115–24.e5.

- Seung, H. S. 2023. "Insights into Vision from Interpretation of a Neuronal Wiring Diagram." *bioRxiv*.
<https://doi.org/10.1101/2023.11.15.567126>.
- . 2024. "Interneuron Diversity and Normalization Specificity in a Visual System." *bioRxiv*.
<https://doi.org/10.1101/2024.04.03.587837>.
- Song, Bo-Mi, and Chi-Hon Lee. 2018. "Toward a Mechanistic Understanding of Color Vision in Insects."
Frontiers in Neural Circuits 12 (February): 16.

Reviewer Reports on the Second Revision:

Referees' comments:

Referee #1 (Remarks to the Author):

I appreciate the authors' efforts to acknowledge the potential problems of functional inference from connectivity-based clustering. The history of EMD connectome and subsequent numerous revisions serves as a painful reminder that such warning is warranted.

With respect to the hypothetical color subsystem, indeed the inclusion of a large number of cells is somewhat unexpected. Color-driven behaviors in *Drosophila* (at least for color-vision) are known to be rather labile, as compared to those of other sensory modalities (motion or olfaction) or in other insects (such as bees and butterflies, where high color vision, such as color constancy, has been readily demonstrated). Whether this could be attributed to *Drosophila* "color" subsystem's connectivity (ex. the lack of swift links to the memory center or motor system) or simply investigators' competence is not known but at this point it would be hard pressed to argue for a sophisticated color constancy system in *Drosophila*. Instead, as the authors alluded to (also with examples from vertebrates), the potentially spectral-coding cells might serve non-color-vision functions (such as color edge detection), or non-image-forming. Breaking the clusters (by lowering the threshold) does not appear to provide any obvious functional insights (but rather dividing cells in different neuropils or gross subtypes). And the cluster 1 is not obviously more sensitive to threshold than the other subsystems. This perhaps shows the limitation of such an approach and the authors' efforts allow me to better understand their predicament. Thus, I won't oppose to the presentation of the hypothetical "color" subsystem with appropriate caveats.

Referee #2 (Remarks to the Author):

I had a few minor comments on the revised version. The authors have addressed those points. Also, looking at the third version and the response to the comments of the two other reviewers, I think the updated manuscript has gotten even better.

Referee #3 (Remarks to the Author):

I would like to thank the authors for their additional work. The revised manuscript (2nd revision) is much improved. I suggest the manuscript is (basically) ready for publication. I have no further major concerns, only few minors that the authors will be happy to implement (I assume).

My most important suggestions affects the concept and realization of grouping and proposing functions for cell types with unknown physiology. This approach is, as other reviewers already mentioned, maybe unavoidable but inherently dangerous. Being too strict here can backfire in future....the authors already

added several 'proposed' etc pp. However:

(1) In small-to-medium size brains with high computational power, individual neuron types can be assumed to be integrated in different circuits in parallel. Matsliah et al., provide examples for this. The phenomenon is likely a general feature of neurons and networks but might be reinforced by a smaller number of cells in total. Furthermore, multiplexing and multi-purpose is likely more prominent towards the front end of sensory circuitries.....I am not sure whether this has been discussed in a satisfying manner.

(2) The second issue affects most of the figures: Fig.3 – 7, Fig.S5, S6. The new and consistent color code is greatly appreciated! However, the currently used yellow labeling accidentally lumps all members of the cluster 1, 3, and 6. The cells of cl1 and cl6 cannot be told apart at all. To retain the 'color to proposed system match', cluster 6 cell types might be displayed by using 'yellow outlines of symbols' instead of filled symbols....

(3) Page 11: >>How yDm8 and pDm8 correspond with our Dm8a and Dm8b remains to be settled. Some preliminary clues are described in the Methods, but a definitive answer will require annotating yellow and pale columns using an improved version of photoreceptor synapses that is forthcoming.<< Given that yellow : pale equals roughly 3 : 1, the authors should be able to provide a guess on the question whether a = yellow and b = pale.....

(4) Page 14/15 >>Some insects are known to have sophisticated color vision capabilities such as color constancy (Song and Lee 2018). The computations required for color constancy are quite complex, requiring the integration of image information over long ranges (Land and McCann 1971).<< Please add the cited literature of this para to the bibliography.

Congratulation to this very nice & important study!

Best regards, Dierk Reiff

Author Rebuttals to Second Revision:

Response to reviews of v4 ‘Neuronal “parts list” and wiring diagram for a visual system’

Referee #1 (Remarks to the Author):

I appreciate the authors' efforts to acknowledge the potential problems of functional inference from connectivity-based clustering. The history of EMD connectome and subsequent numerous revisions serves as a painful reminder that such warning is warranted.

The history of Takemura et al. 2013 is painful to us, even though we only watched from afar! We thank the referee for reminding us to be circumspect about hypotheses and speculations.

With respect to the hypothetical color subsystem, indeed the inclusion of a large number of cells is somewhat unexpected. Color-driven behaviors in *Drosophila* (at least for color-vision) are known to be rather labile, as compared to those of other sensory modalities (motion or olfaction) or in other insects (such as bees and butterflies, where high color vision, such as color constancy, has been readily demonstrated). Whether this could be attributed to *Drosophila* “color” subsystem’s connectivity (ex. the lack of swift links to the memory center or motor system) or simply investigators’ competence is not known but at this point it would be hard pressed to argue for a sophisticated color constancy system in *Drosophila*. Instead, as the authors alluded to (also with examples from vertebrates), the potentially spectral-coding cells might serve non-color-vision functions (such as color edge detection), or non-image-forming. Breaking the clusters (by lowering the threshold) does not appear to provide any obvious functional insights (but rather dividing cells in different neuropils or gross subtypes). And the cluster 1 is not obviously more sensitive to threshold than the other subsystems. This perhaps shows the limitation of such an approach and the authors’ efforts allow me to better understand their predicament. Thus, I won’t oppose to the presentation of the hypothetical “color” subsystem with appropriate caveats.

Referee #2 (Remarks to the Author):

I had a few minor comments on the revised version. The authors have addressed those points. Also, looking at the third version and the response to the comments of the two other reviewers, I think the updated manuscript has gotten even better.

Referee #3 (Remarks to the Author):

I would like to thank the authors for their additional work. The revised manuscript (2nd revision) is much improved. I suggest the manuscript is (basically) ready for publication. I have no further major concerns, only few minors that the authors will be happy to implement (I assume).

My most important suggestions affects the concept and realization of grouping and proposing functions for cell types with unknown physiology. This approach is, as other reviewers already mentioned, maybe unavoidable but inherently dangerous. Being too strict here can backfire in future...the authors already added several 'proposed' etc pp. However:

We have added a sentence clarifying that these are speculations, and serve as a starting point for hypothesis generation and experimentation.

(1) In small-to-medium size brains with high computational power, individual neuron types can be assumed to be integrated in different circuits in parallel. Matsliah et al., provide examples for this. The phenomenon is likely a general feature of neurons and networks but might be reinforced by a smaller number of cells in total.

Furthermore, multiplexing and multi-purpose is likely more prominent towards the front end of sensory circuitries....I am not sure whether this has been discussed in a satisfying manner.

Thanks for making this really important point. The Discussion now includes cautionary text that cell types can have multiple functions, so the hard cluster boundaries are simplistic.

(2) The second issue affects most of the figures: Fig.3 - 7, Fig.S5, S6. The new and consistent color code is greatly appreciated! However, the currently used yellow labeling accidentally lumps all members of the cluster 1, 3, and 6. The cells of cl1 and cl6 cannot be told apart at all. To retain the 'color to proposed system match', cluster 6 cell types might be displayed by using 'yellow outlines of symbols' instead of filled symbols...

We use similar colors for functionally related clusters. These have been refined to be more distinguishable.

(3) Page 11: >>How yDm8 and pDm8 correspond with our Dm8a and Dm8b remains to be settled. Some preliminary clues are described in the Methods, but a definitive answer will require annotating yellow and pale columns using an improved version of photoreceptor synapses that is forthcoming.<<

Given that yellow : pale equals roughly 3 : 1, the authors should be able to provide a guess on the question whether a = yellow and b = pale.....

The obvious guess is that Dm8a = yellow and Dm8b = pale. Here is the relevant excerpt from the Methods:

Molecular studies previously divided Dm8 cells into two types (yDm8 and pDm8), depending on whether or not they express DIP γ ^{51,53}. Physiological studies demonstrated that yDm8 and pDm8 have differing spectral sensitivities⁸⁸. The main dendrites of yDm8 and pDm8 were found to connect with R7 in yellow and pale columns, respectively. Based on its strong coupling with Tm5a, our Dm8a likely has some correspondence with

yDm8, which is likewise selectively connected with Tm5a^{51,53}. It is not yet clear whether there is a true one-to-one correspondence of yDm8 and pDm8 with Dm8a and Dm8b. It is the case that Dm8a and Dm8b strongly prefer to synapse onto Tm5a and Tm5b, respectively. However, Tm5a and Tm5b are not in one-to-one correspondence with yellow and pale columns. Rather, the main dendritic branch of Tm5a is specific to yellow columns, while the main dendritic branches of Tm5b are found in both yellow and pale columns⁵⁰. Furthermore, Dm8a and Dm8b cells are roughly equal in number, while the yDm8:pDm8 ratio is expected to be substantially greater than one,^{51,53} like the ratio of yellow to pale columns. Therefore the correspondence of Dm8a and Dm8b with yDm8 and pDm8 is still speculative.

The numbers do not match up, which is why the obvious guess is omitted from the main text.

(4) Page 14/15 >>Some insects are known to have sophisticated color vision capabilities such as color constancy (Song and Lee 2018). The computations required for color constancy are quite complex, requiring the integration of image information over long ranges (Land and McCann 1971).<<

Please add the cited literature of this para to the bibliography.

Thanks for catching the omission. It has been added now.

Congratulation to this very nice & important study!

Best regards, Dierk Reiff